# Fibroblast growth factor 18 stimulates the proliferation of hepatic stellate cells, thereby inducing liver fibrosis

Yuichi Tsuchiya[1,2,19], Takao Seki[1,19], Kenta Kobayashi[1,3], Sachiko Komazawa-Sakon [1], Shigeyuki Shichino [4], Takashi Nishina [1], Kyoko Fukuhara[5], Kenichi Ikejima[5], Hidenari Nagai [6], Yoshinori Igarashi[6], Satoshi Ueha [4], Akira Oikawa[7], Shinya Tsurusaki [8,9], Soh Yamazaki[1], Chiharu Nishiyama [3], Tetuo Mikami[10], Hideo Yagita[11], Ko Okumura[12], Taketomo Kido[13], Atsushi Miyajima[13], Kouji Matsushima [4], Mai Imasaka[14], Kimi Araki [15,16], Toru Imamura[17,18], Masaki Ohmuraya[14], Minoru Tanaka [8,9] & Hiroyasu Nakano [1] ✉

Liver fibrosis results from chronic liver injury triggered by factors such as viral infection, excess alcohol intake, and lipid accumulation. However, the mechanisms underlying liver fibrosis are not fully understood. Here, we demonstrate that the expression of *fibroblast growth factor 18* (*Fgf18*) is elevated in mouse livers following the induction of chronic liver fibrosis models. Deletion of *Fgf18* in hepatocytes attenuates liver fibrosis; conversely, overexpression of *Fgf18* promotes liver fibrosis. Single-cell RNA sequencing reveals that overexpression of *Fgf18* in hepatocytes results in an increase in the number of *Lrat*⁺ hepatic stellate cells (HSCs), thereby inducing fibrosis. Mechanistically, FGF18 stimulates the proliferation of HSCs by inducing the expression of *Ccnd1*. Moreover, the expression of *FGF18* is correlated with the expression of profibrotic genes, such as *COL1A1* and *ACTA2*, in human liver biopsy samples. Thus, FGF18 promotes liver fibrosis and could serve as a therapeutic target to treat liver fibrosis.

Liver fibrosis results from chronic liver injury triggered by many factors, including viral infection, excess alcohol intake, lipid accumulation, and autoimmune diseases[1,2]. Liver fibrosis ultimately develops into liver failure and cancer; hence, liver fibrosis is tightly associated with cause-specific mortality in several liver diseases[3,4]. In addition to liver fibrosis caused by chronic viral hepatitis, nonalcoholic steatohepatitis (NASH), which is now renamed metabolic dysfunction-associated steatohepatitis (MASH)[5], is one of the leading causes of liver fibrosis. Simple steatosis and MASH belong to the same disease category characterized by metabolic dysfunction-associated steatotic liver disease (MASLD)[5]. Approximately 10% of patients with MASLD are ultimately diagnosed with MASH, which is characterized by hepatocyte steatosis, inflammation, and hepatocyte cell death[6,7]. Although MASH

is presumably caused by cell death due to lipid-mediated toxicity or oxidative stress, the detailed mechanisms are not fully understood[8,9].

Recent advancements in single-cell RNA sequencing (scRNA-seq) technology have revealed that cell–cell interactions play a crucial role in the development of liver fibrosis[10–12]. Among the various cells involved in liver fibrosis, hepatic stellate cells (HSCs) are central players in promoting liver fibrosis[13,14]. Under physiological conditions, quiescent HSCs mainly localize in the space of Disse and contain large amounts of vitamin A, thereby contributing to vitamin A homeostasis. Quiescent HSCs have retinol-containing lipid droplets and express the characteristic marker genes, *lecithin retinol acyltransferase (Lrat)* and *desmin (Des)*[10]. In response to liver injury, HSCs become activated and activated HSCs may lose the expression of *Lrat* but evolve into

---

α-SMA-expressing myofibroblasts that highly express profibrotic genes[13,14]. Various cytokines have been shown to activate HSCs; for example, Hedgehog, connective tissue growth factor (CTGF), and transforming growth factor beta (TGFβ) activate HSCs[15]. However, it remains unclear which growth factors control the numbers of quiescent and activated HSCs and myofibroblasts during the development of fibrosis.

Fibroblast growth factors (FGFs) comprise 22 structurally related proteins and regulate various biological and cellular responses, including organogenesis, tissue remodeling, angiogenesis, and regulation of metabolism[16,17]. The extracellular FGF family comprises the FGF1, FGF4, FGF7, FGF8, and FGF19 subfamilies, and FGFs transmit signals through cognate receptors, including FGF receptors 1 to 4. The FGF1, FGF4, FGF7, and FGF8 subfamilies exert their activities in a paracrine or autocrine manner, whereas the FGF19 subfamily acts as a hormone[16,17]. Of note, members of the FGF family play crucial roles in regulating liver fibrosis[18]. For example, FGF1 and FGF2 suppress the activation of HSCs[19,20], thereby attenuating liver fibrosis. However, another study reported that deletion of *Fgf1* and *Fgf2* attenuates CCl4-induced liver fibrosis[21], suggesting that FGF1 and FGF2 exhibit a profibrotic role under certain conditions. While hepatic fibrosis is attenuated in *Fgf15*-deficient mice in murine MASH and CCl4-induced models[22], administration of an FGF21 analog attenuates hepatic inflammation and fibrosis in a murine MASH model[23]. Together, these results suggest that members of the FGF family attenuate or exacerbate liver fibrosis depending on the experimental conditions.

Cellular FLICE-inhibitory protein (cFLIP), which is encoded by *Cflar*, is a caspase 8 homolog but does not have protease activity; it binds to caspase 8 and suppresses apoptosis[24]. Germline deletion of *Cflar* results in embryonic lethality due to enhanced apoptosis and necroptosis[25,26], but the in vivo functions of *Cflar* have been extensively investigated by deleting *Cflar* in specific tissues[27]. We and others have reported that hepatocyte-specific *Cflar*-deficient (*Cflar^LKO*) mice have increased susceptibility to death ligand-induced cell death[28,29]. Using *Cflar^LKO* mice, we identified that histone H3 is released from apoptotic hepatocytes and induces endothelial cell injury[30].

In the present study, we utilized *Cflar^LKO* mice to identify factors that promote liver fibrosis. We identified FGF18 as a critical factor promoting liver fibrosis in murine chronic liver injury models. scRNA-seq revealed that overexpression of FGF18 increases the number of *Lrat*⁺ HSCs. In vitro experiments showed that FGF18 induces the proliferation of HSCs by inducing the expression of *Ccnd1* but suppresses TGFβ-induced profibrotic gene expression, thereby providing a microenvironment for the proliferation of HSCs. Moreover, the expression of *FGF18* is correlated with the expression of profibrotic genes, such as *COL1A1* and *ACTA2*, in human liver biopsy samples. Our findings suggest that FGF18 promotes liver fibrosis and could serve as a therapeutic target to treat liver fibrosis.

## Results
### *Cflar^LKO* mice develop mild liver fibrosis
Since liver fibrosis is tightly associated with apoptosis of hepatocytes[31], we crossed *Cflar^flox* mice with *albumin* promoter-driven *Cre recombinase* transgenic (*Alb-Cre*) mice to generate *Cflar^LKO* mice that were utilized to identify factors promoting liver fibrosis. As we previously reported[29], the expression of cFLIP at the mRNA and protein levels diminished, but was not entirely abrogated, in hepatocytes of *Cflar^LKO* mice (Supplementary Fig. 1a, b). Serum alanine aminotransferase (ALT) concentrations were slightly increased in *Cflar^LKO* mice compared to *Cflar^FF* mice (Supplementary Fig. 1c), suggesting that mild liver injury occurred in *Cflar^LKO* mice under homeostatic conditions. Immunohistochemistry (IHC) revealed that the numbers of cleaved caspase 3 (CC3)⁺ and TUNEL⁺ cells were slightly increased in the livers of *Cflar^LKO* mice (Supplementary Fig. 1d–f). Notably, Sirius Red⁺ areas, the hydroxyproline content, and the expression of *Col1a2* and *Col3a1*, which are hallmarks

of fibrosis, were increased in the livers of *Cflar^LKO* mice compared to *Cflar^FF* mice (Supplementary Fig. 1d, g–i). Moreover, CK19⁺ areas and expression of *Krt19*, which encodes CK19, were increased in the livers of *Cflar^LKO* mice (Supplementary Fig. 1d, j, k), suggesting that the ductular reaction was enhanced. These results indicate that *Cflar^LKO* mice spontaneously developed mild liver fibrosis and ductular reaction.

### Liver fibrosis is exacerbated in *Cflar^LKO* mice fed the CDE diet
To identify factors that promote liver fibrosis, we treated or fed *Cflar^LKO* mice various agents or diets. Among them, we chose the choline-deficient ethionine-supplemented (CDE) diet because our preliminary experiments revealed that the CDE diet resulted in drastic exacerbation of liver fibrosis in *Cflar^LKO* mice (Fig. 1). We first determined the experimental conditions for the CDE diet, such as the sex and ages of mice, to efficiently induce liver fibrosis using wild-type mice. We found that 8-week-old female mice fed the CDE diet for 4 weeks were suitable for the liver fibrosis model (Supplementary Fig. 2). Then, we compared the phenotypes of 8-week-old female *Cflar^FF* and *Cflar^LKO* mice fed the normal or CDE diet for 4 weeks. ALT concentrations were significantly elevated in *Cflar^LKO* mice fed the CDE diet compared to *Cflar^FF* mice fed the CDE diet or *Cflar^LKO* mice fed the normal diet (Fig. 1a). The body weights (BWs) of *Cflar^FF* and *Cflar^LKO* mice fed the normal diet gradually increased at 12 weeks after birth (left panel, Fig. 1b). However, the BW of *Cflar^LKO* mice was progressively decreased during the CDE diet (right panel, Fig. 1b). In contrast, BW recovered to pre-diet levels before the CDE diet in *Cflar^FF* mice (right panel, Fig. 1b). As expected, many hepatocytes from *Cflar^FF* and *Cflar^LKO* mice fed the CDE diet were filled with cytoplasmic fat droplets (Fig. 1c). Consistent with an increase in ALT levels in *Cflar^LKO* mice after 4 weeks of the CDE diet, the numbers of both CC3⁺ and TUNEL⁺ cells were increased in the livers of *Cflar^LKO* mice (Fig. 1d, e). Moreover, Sirius Red⁺ and desmin⁺ areas, the hydroxyproline content, and the expression of *Col1a2* and *Col3a1* were increased in the livers of *Cflar^LKO* mice compared to *Cflar^FF* mice fed the CDE diet for 4 weeks (Fig. 1c, f–i). CK19⁺ areas and *Krt19* expression were increased in the livers of *Cflar^LKO* mice (Fig. 1c, j, k). Given that desmin is a universal marker for HSCs[10], these results indicate that *Cflar^LKO* mice rapidly developed severe liver fibrosis along with an increase in the number of HSCs after being fed the CDE diet.

### *Fibroblast growth factor 18* is elevated in the livers of *Cflar^LKO* mice
To identify the gene(s) involved in the exacerbation of liver fibrosis in *Cflar^LKO* mice, we compared the gene expression profiles of *Cflar^FF* and *Cflar^LKO* mice fed the CDE diet for 4 weeks by whole liver RNA sequencing (RNA-seq) (Supplementary Data 1). Gene Ontology (GO) enrichment analysis revealed that genes related to cell adhesion, extracellular matrix organization, and cell migration were enriched in the livers of *Cflar^LKO* mice compared to *Cflar^FF* mice (Fig. 2a, b). As the desmin⁺ and CK19⁺ areas were increased in the livers of *Cflar^LKO* mice (Fig. 1c, g, j), we focused on growth factor-related genes. After investigating the mRNA expression kinetics in the livers of 8-week-old female mice before and after CDE diet feeding for 4 weeks (Fig. 2c), we focused on *Fgf18*, *Tnfsf13b*, and *Wnt5a*. We tested whether overexpression of these genes induced the proliferation of desmin⁺ and CK19⁺ cells in the livers of wild-type mice using the hydrodynamic tail vein injection (HTVi) method[32]. HTVi is an in vivo transfer method that yields high gene expression levels in the liver. We first confirmed that infused genes were overexpressed in the livers using qPCR and IHC for EGFP as a positive control (Supplementary Fig. 3a–c). Unexpectedly, the Ki67⁺ areas and, to a lesser extent, desmin⁺ areas, but not the CK19⁺ areas, were increased in the livers injected with *Fgf18* compared to those injected with *Egfp*, *Tnfsf13b*, or *Wnt5a* 7 days after injection (Fig. 2d, e). Of note, most Ki67⁺ cells did not express CK19, suggesting that these cells were mostly noncholangiocytes (Supplementary Fig. 3d, e). Thus, we focused on FGF18 for subsequent experiments.

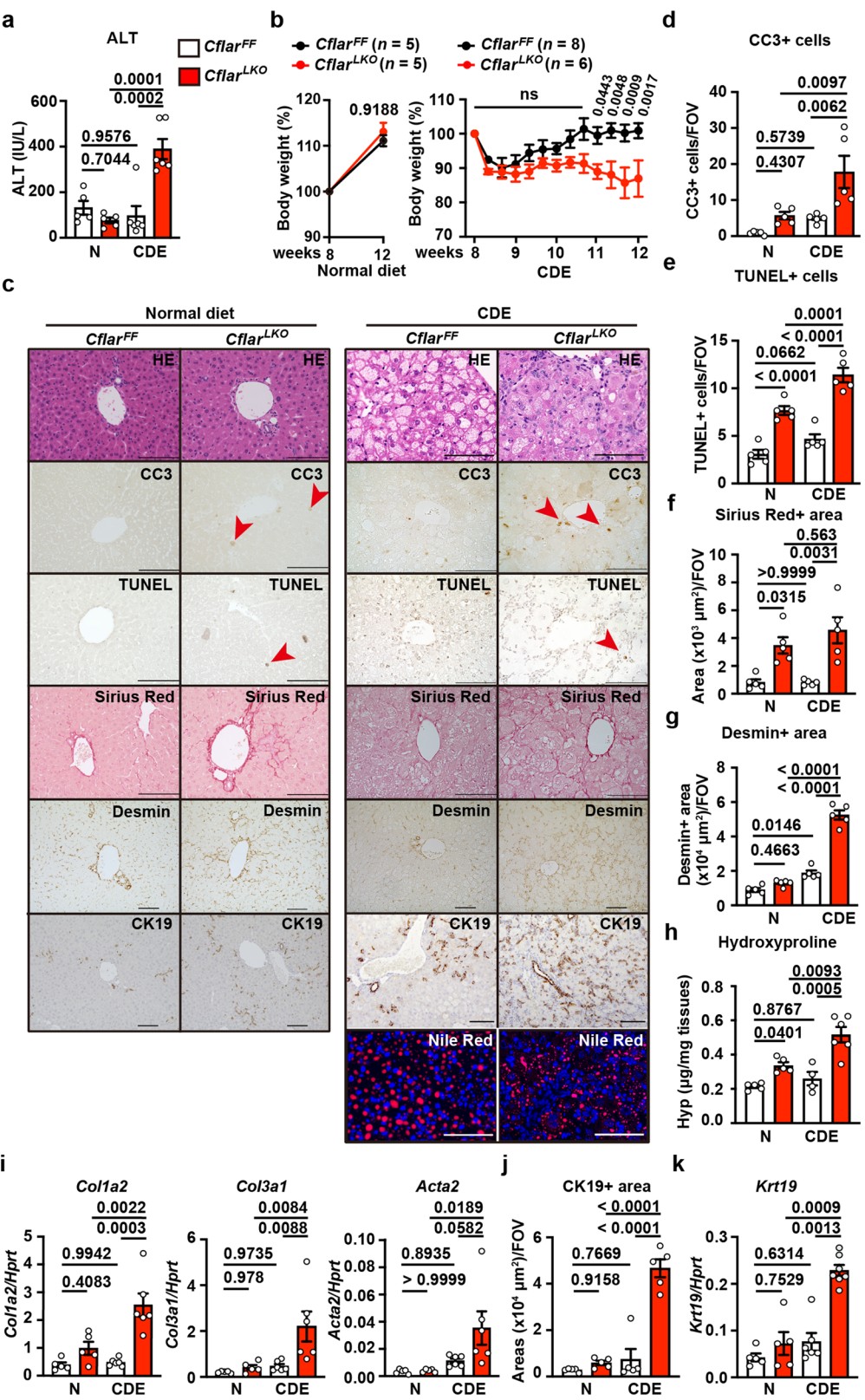

To determine the cellular source of FGF18, we examined the livers by RNAscope, since reliable antibodies against FGF18 for IHC are unavailable as far as we explored. We visualized *Fgf18* mRNA along with *Hnf4a* mRNA (a hallmark of hepatocytes) or *Lrat* mRNA (a hallmark of HSCs) by RNAscope. *Fgf18* mRNA⁺ puncta were very closely located to *Hnf4a*⁺ puncta or *Lrat*⁺ puncta (Fig. 2f), suggesting that hepatocytes and HSCs expressed *Fgf18*. Of note, *Fgf18* mRNA⁺ puncta areas were

increased in the livers of *Cflar^LKO* mice compared to those of *Cflar^FF* mice fed the CDE diet for 4 weeks (Fig. 2f, g). Although *Fgf18* mRNA⁺ puncta were detected in comparable numbers in *Hnf4a*⁺ hepatocytes in *Cflar^FF* and *Cflar^LKO* mice, *Fgf18* mRNA⁺ puncta in nonparenchymal liver cells (NPCs) (presumed *Lrat*⁺ HSCs) were increased in *Cflar^LKO* livers compared to *Cflar^FF* livers (Fig. 2g). Therefore, both HSCs and hepatocytes expressed *Fgf18* under these experimental conditions.

**Fig. 1 | CDE diet-induced liver injury is exacerbated in *Cflar^{LKO}* mice.** Eight-week-old female *Cflar^{FF}* and *Cflar^{LKO}* mice were fed the normal or CDE diet for 4 weeks. **a** Serum ALT concentrations were determined. Results are mean ± SE. *n* = 5 (normal diet) or *n* = 6 (CDE diet) mice. **b** Kinetics of relative body weight (%) changes. Results are mean ± SE. *n* = 5 (*Cflar^{FF}* and *Cflar^{LKO}* fed the normal diet), *n* = 8 (*Cflar^{FF}* fed the CDE diet), or *n* = 6 (*Cflar^{LKO}* fed the CDE diet) mice. **c–g, j** Liver sections from mice with the indicated genotypes fed the normal diet or CDE diet for 4 weeks were stained with H&E, anti-CC3 antibody, TUNEL, Sirius Red, anti-desmin and anti-CK19 antibodies, and Nile Red (*n* = 5 mice) (**c**). Scale bar, 100 μm. Red arrowheads indicate CC3⁺ or TUNEL⁺ cells. The number of CC3⁺ (**d**) and TUNEL⁺ (**e**) cells were counted and expressed as numbers of the field of view (FOV). The Sirius Red⁺ (**f**), desmin⁺ (**g**), and CK19⁺ (**j**) areas were quantified and are expressed as areas of FOV. Results are mean ± SE (*n* = 5 mice). The hydroxyproline content of the livers were determined (**h**). Results are mean ± SE. *n* = 5 (*Cflar^{FF}* and *Cflar^{LKO}* fed the normal diet), *n* = 4 (*Cflar^{FF}* fed the CDE diet), or *n* = 6 (*Cflar^{LKO}* fed the CDE diet) mice. The expression of profibrotic genes (**i**) and *Krt19* (CK19) (**k**) in the livers was determined by qPCR. Results are mean ± SE (*n* = 5). Pooled results from four independent experiments are shown. Significance was determined by two-way ANOVA with Tukey's multiple comparison test (**a**, **d–k**), two-tailed unpaired Student's *t* test (**b**, left), or two-way ANOVA with Sidak's multiple comparison test (**b**, right). Source data are provided as a Source Data file.

Intriguingly, the expression of *Fgf18* was also elevated in choline-deficient, L-amino acid-defined, high-fat diet (CDAHFD)-induced murine MASH or 3,5-diethoxycarbonyl-1,4-dihydrocollidine–supplemented (DDC) diet-induced liver fibrosis models in wild-type mice[33,34] (Supplementary Fig. 4a–h). Reanalysis of GSE99010[35] revealed that the expression of *Fgf18* was correlated with the expression of profibrotic genes, such as *Col1a1* and *Col3a1*, in the livers of mice fed the normal diet or the Western diet combined with $CCl_4$ injections for 24 weeks but not 12 weeks (Supplementary Fig. 4i, Supplementary Table 1). Of note, the expression of *Fgf18* was elevated at relatively late stages compared to that of other profibrotic genes, such as *Acta2, Col1a2, Col3a1*, and *Tgfb1-3* (Supplementary Fig. 4j). These results suggest that increased expression of *Fgf18* is observed in multiple liver fibrosis models, especially in the advanced stages of liver fibrosis.

## Hepatocyte-specific deletion of *Fgf18* attenuates fibrosis in *Cflar^{LKO}* mice

To determine whether FGF18 attenuates or exacerbates liver fibrosis in *Cflar^{LKO}* mice, we generated hepatocyte-specific *Fgf18*-deficient mice (*Fgf18^{LKO}* mice) by crossing *albumin-Cre* mice with *Fgf18^{FF}* mice[36]. Although germline deletion of *Fgf18* results in perinatal death due to a defect in alveolar dilatation[37], *Fgf18^{LKO}* mice were born at the expected Mendelian ratio and grew to adulthood. As expected, *Fgf18* expression was decreased in hepatocytes in *Fgf18^{LKO}* mice (Supplementary Fig. 5a). The expression of the *Cflar* gene was slightly elevated in hepatocytes and NPCs in *Fgf18^{LKO}* mice (Supplementary Fig. 5a, b), but the underlying mechanism is currently unknown. *Fgf18^{LKO}* mice fed the normal diet did not exhibit any liver abnormalities (Supplementary Fig. 5c, d), and compared to *Fgf18^{FF}* mice, *Fgf18^{LKO}* mice fed the CDE diet did not show further exacerbation or attenuation of liver fibrosis (Supplementary Fig. 5e to m).

We generated mice lacking both *Cflar* and *Fgf18* in hepatocytes (*Cflar^{LKO}*;*Fgf18^{LKO}* mice). With the normal diet, the expression of *Fgf18* was diminished in hepatocytes but elevated in NPCs from *Cflar^{LKO}*;*Fgf18^{LKO}* mice compared to *Cflar^{FF}*;*Fgf18^{FF}* mice (Fig. 3a). We also found that *Fgf18* mRNA expression was elevated in NPCs from *Cflar^{LKO}* mice compared to NPCs from *Cflar^{FF}* mice fed the normal diet (Fig. 3b). However, the expression of *Cflar* at the mRNA and protein levels was diminished in hepatocytes, but not NPCs, from *Cflar^{LKO}*;*Fgf18^{LKO}* mice (Fig. 3a, c). These results suggest that the expression of *Fgf18* in NPCs, but not hepatocytes, was elevated in the absence of *Cflar* in mouse hepatocytes, even under normal diet conditions.

We then compared the extent of liver injury and fibrosis among *Fgf18^{LKO}*, *Cflar^{LKO}*, and *Cflar^{LKO}*;*Fgf18^{LKO}* mice following 4 weeks of CDE diet feeding. We utilized the results of *Fgf18^{FF}* and *Fgf18^{LKO}* mice fed the CDE diet in Supplementary Fig. 5. Serum concentrations of ALT and aspartate aminotransferase (AST) were elevated in *Cflar^{LKO}* mice compared to *Fgf18^{LKO}* mice but comparable to those in *Cflar^{LKO}*;*Fgf18^{LKO}* mice. In contrast, serum concentrations of alkaline phosphatase (ALP) did not differ among *Fgf18^{LKO}*, *Cflar^{LKO}*, and *Cflar^{LKO}*;*Fgf18^{LKO}* mice (Fig. 3d). *Fgf18* expression was elevated in the livers of *Cflar^{LKO}* mice compared to *Fgf18^{LKO}* mice, but its expression was reduced in the livers of *Cflar^{LKO}*;*Fgf18^{LKO}* mice (Fig. 3e), confirming that hepatocytes in the livers of *Cflar^{LKO}* mice expressed *Fgf18* to some extent. Intriguingly, the Sirius Red⁺ areas and hydroxyproline content, but not the desmin⁺ areas, were elevated in the livers of *Cflar^{LKO}* mice compared to *Fgf18^{LKO}* mice, but their expression was reduced in the livers of *Cflar^{LKO}*;*Fgf18^{LKO}* mice (Fig. 3f, g–i). Accordingly, the expression of *Col3a1* and *Acta2* was elevated in the livers of *Cflar^{LKO}* mice compared to *Fgf18^{LKO}* mice but attenuated in the livers of *Cflar^{LKO}*;*Fgf18^{LKO}* mice (Fig. 3j). These results suggest that FGF18 derived from hepatocytes contributes, at least in part, to liver fibrosis. The partial attenuating effect of *Fgf18* deletion on liver fibrosis may be caused by residual expression of *Fgf18* in HSCs or compensation by other cytokines. In contrast, CK19⁺ cells and *Krt19* expression were elevated in the livers of *Cflar^{LKO}* mice compared to *Fgf18^{LKO}* mice, but their expression was not decreased in the livers of *Cflar^{LKO}*;*Fgf18^{LKO}* mice (Fig. 3f, k, l).

## Overexpression of *Fgf18* in hepatocytes promotes liver fibrosis

To determine whether the expression of FGF18 alone induces liver fibrosis in vivo, we generated transgenic mice expressing *Fgf18* in hepatocytes. We integrated *Fgf18* cDNA into the *Rosa 26* locus under the control of the *CAG* promoter interrupted by a cassette containing *loxP*-STOP-*loxP* (*LSL*) sites (Fig. 4a), enabling us to express *Fgf18* in a *Cre* recombinase-dependent manner[38]. We crossed *Rosa26-CAG-LSL-Fgf18* Tg mice with *albumin-Cre* Tg mice, generating mice that expressed *Fgf18* in hepatocytes. Hereafter, we refer to *Alb-Cre;Rosa26-CAG-LSL-Fgf18* Tg mice as *Fgf18* Tg mice. *Fgf18* Tg mice appeared normal until adulthood based on their BW and appearance. However, *Fgf18* Tg mice spontaneously developed liver hypertrophy; their liver weights were approximately 1.5-fold greater than those of non-Tg mice (Fig. 4b–d). *Fgf18* mRNA expression was elevated in hepatocytes from *Fgf18* Tg mice compared to those from non-Tg mice (Fig. 4e). We also detected the protein expression of FGF18 in hepatocytes, but not NPCs, from *Fgf18* Tg mice by Western blotting (Fig. 4f). Of note, due to glycosylation, endogenous mFGF18 in hepatocytes migrated slower than recombinant mFGF18 in *E. coli*. Moreover, our in-house enzyme-linked immunosorbent assay (ELISA) for human FGF18 detected mouse FGF18 in the culture supernatants of hepatocytes from *Fgf18* Tg mice[39] (Fig. 4g). In sharp contrast, we could not detect protein expression of FGF18 in either hepatocytes or NPCs from *Cflar^{LKO}*, *Fgf18^{LKO}*, or *Cflar^{LKO}*;*Fgf18^{LKO}* mice due to the lower expression levels of *Fgf18* (Supplementary Fig. 6a–c).

Serum ALT, AST, and ALP concentrations did not differ between non-Tg and *Fgf18* Tg mice (Fig. 4h). H&E-stained liver tissues and IHC revealed that the numbers of mononuclear cells surrounding the large vessels, including Ly6G⁺ neutrophils and CD68⁺ monocytes, were increased in the livers of *Fgf18* Tg mice (Fig. 4i, Supplementary Fig. 7a–d). These cells appeared positive for Ki67 and were considered proliferating (Supplementary Fig. 7e, f). Flow cytometry revealed that the percentages of B cells, monocytes (CD11b⁺Ly6G⁻), and neutrophils (CD11b⁺Ly6G⁺) were increased in the livers of *Fgf18* Tg mice compared to those of non-Tg mice (Supplementary Fig. 7g–i). In several portal areas, the number of small vessels was increased (Fig. 4i), suggesting that aberrant angiogenesis occurred in the livers of *Fgf18* Tg mice.

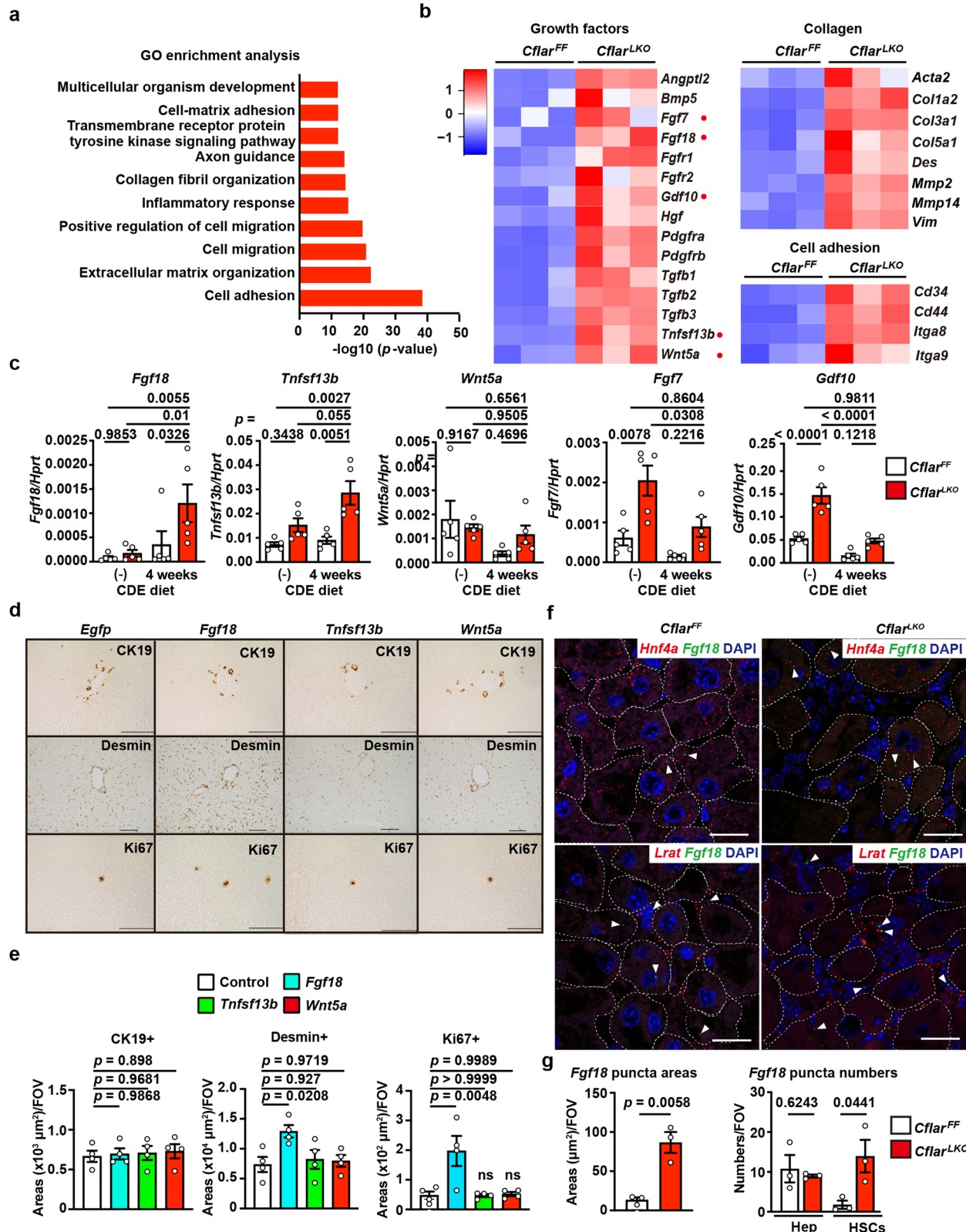

**a** GO enrichment analysis

**b** Growth factors | Collagen | Cell adhesion

**c** Fgf18, Tnfsf13b, Wnt5a, Fgf7, Gdf10 (CDE diet)

**d** Egfp, Fgf18, Tnfsf13b, Wnt5a (CK19, Desmin, Ki67)

**e** CK19+, Desmin+, Ki67+

**f** CflarFF / CflarLKO (Hnf4a Fgf18 DAPI; Lrat Fgf18 DAPI)

**g** Fgf18 puncta areas; Fgf18 puncta numbers

Many cytokines were elevated in the livers of *Cflar^LKO* and *Fgf18* Tg mice compared to the respective control mice (Supplementary Fig. 8a). However, deletion of *Fgf18* only marginally attenuated the expression of several inflammatory cytokines (Supplementary Fig. 8b), suggesting that FGF18 in hepatocytes did not play a dominant role in the induction of these cytokines under *Cflar*-deficient conditions but could upregulate them when it was overexpressed.

Intriguingly, Sirius Red⁺ areas and the hydroxyproline content were elevated in the livers of *Fgf18* Tg mice compared to non-Tg mice (Fig. 4j–l). CK19⁺ areas and *Krt19* expression were increased in the livers

**Fig. 2 | Fibroblast growth factor 18 is elevated in the livers of *Cflar^LKO* mice.** **a**, **b** Eight-week-old female *Cflar^FF* and *Cflar^LKO* mice were fed the CDE diet for 4 weeks, and gene expression in the whole liver was analyzed by RNA-seq. Gene Ontology (GO) enrichment analysis was performed using DAVID 6.8 (**a**). Statistical analysis was determined by one-sided Fisher's exact test. Heatmap showing the Z score scaled expression levels of representative genes in the indicated categories (*n* = 3 mice) (**b**). **c** Kinetics of the expression of the indicated genes in the livers of 8-week-old female mice before (−) and after CDE diet feeding for 4 weeks. Results are means ± SE (*n* = 5 mice). Pooled results from four independent experiments are shown. **d**, **e** Expression of *Fgf18* in the liver by HTVi results in the proliferation of hepatocytes. Eight-week-old female wild-type (WT) mice were injected with the indicated expression vectors by the HTVi method. Liver sections were stained with anti-CK19, anti-desmin, or anti-Ki67 antibodies (**d**) (*n* = 4 mice). Scale bar, 100 μm.

The CK19⁺, desmin⁺, or Ki67⁺ areas were calculated and are expressed as positive areas per FOV (**e**). Results are means ± SE (*n* = 4 mice) and represent two independent experiments. **f**, **g** Mice were treated as in (**a**), and the expression of *Fgf18*, *Hnf4a*, and *Lrat* was determined by RNAscope (*n* = 3 mice) (**f**). Red puncta indicate *Hnf4a* (upper panels) and *Lrat* (lower panels). White arrowheads indicate *Fgf18* mRNA⁺ puncta (green). Nuclei were stained with DAPI (blue). White dotted lines outline the margins of hepatocytes. Scale bar, 100 μm. The total *Fgf18* mRNA⁺ areas and numbers of *Fgf18* mRNA⁺ puncta in *Hnf4a*⁺ hepatocytes and *Lrat*⁺ HSCs were calculated (**g**). Results are mean ± SE (*n* = 3 mice). Statistical significance was determined by the two-way ANOVA with Tukey's multiple comparison test (**c**), one-way ANOVA with Dunnett's multiple comparison test (**e**), or two-tailed unpaired Student's *t* test (**g**). Source data are provided as a Source Data file.

---

of *Fgf18* Tg mice (Fig. 4m–o). These results suggest that over-expression of *Fgf18* alone results in liver fibrosis and ductular reaction.

## FGF18 increases the number of CD31⁻CD34⁺ stromal cells

We performed RNA-seq of the whole liver to elucidate the mechanisms underlying FGF18-induced liver fibrosis (Supplementary Data 2). GO enrichment analysis revealed that genes associated with cell adhesion, angiogenesis, collagen fibril organization, and growth factors were enriched in the livers of *Fgf18* Tg mice compared to non-Tg mice (Fig. 5a, b). We extracted genes elevated more than 2-fold in the livers of two murine models: *Cflar^LKO* mice fed the CDE diet and *Fgf18* Tg mice. A total of 600 genes were upregulated in both models (Fig. 5c; Supplementary Data 3). GO enrichment analysis revealed that these overlapping genes were categorized in cell adhesion, extracellular matrix organization, and collagen fibril organization (Fig. 5d), suggesting that expression of *Fgf18* alone upregulates, at least in part, signature genes involved in CDE-induced liver fibrosis.

As the expression of *Cd34* was elevated in the whole liver RNA-seq of both *Cflar^LKO* mice fed the CDE diet and *Fgf18* Tg mice (Figs. 2b, 5b), we focused on *Cd34*. CD34 is a marker of hematopoietic progenitor cells but is also expressed in vascular endothelial cells, mesenchymal stromal cells, and activated HSCs[40]. We used flow cytometry to analyze and characterize CD34⁺ cells. CD34⁺ cells were composed of CD31⁺CD34⁺ cells and CD31⁻CD34⁺ cells, and the percentage of the latter cells was increased in the livers of *Fgf18* Tg mice (Fig. 5e, f). Most CD31⁻CD34⁺ cells expressed PDGFRα and Sca1, with minor populations expressing Thy1 or podoplanin (Fig. 5e–g). Importantly, sorted CD31⁻CD34⁺ cells expressed *Lrat*, *Hgf*, *Ngfr*, *Acta2*, and *Col1a2* at higher levels than sorted CD31⁺ cells (Supplementary Fig. 9a, Fig. 5h), indicating that CD31⁻CD34⁺ cells indeed contained HSCs. RNAscope and IHC further confirmed that *Lrat*-expressing and desmin⁺ HSCs were increased in the livers of *Fgf18* Tg mice compared to non-Tg mice (Fig. 5i–l).

IHC revealed that the numbers of CD34⁺ cells coexpressing the p75 nerve growth factor receptor (p75NTR) encoded by *Ngfr*, a marker of HSCs[41], and vimentin were increased in the livers of *Fgf18* Tg mice (Supplementary Fig. 9b, c). The expression of *Cd34* and the numbers of CD34⁺ cells were increased in the livers of *Cflar^LKO* mice fed the CDE diet and wild-type mice fed the CDAHFD diet (Supplementary Fig. 9d–i), suggesting that the expansion of CD34⁺ cells is not specific to *Fgf18* Tg mice but a more generalized phenomenon in liver fibrosis.

## Characterization of CD31⁻CD34⁺ stromal cells and cell–cell communication by scRNA-seq

To further characterize CD31⁻CD34⁺ cells, we isolated lineage marker-negative (CD31⁻ CD45.2⁻ CD146⁻ Epcam⁻ Ter119⁻) NPCs from the livers of non-Tg and *Fgf18* Tg mice (Supplementary Fig. 10a) and characterized these cells by scRNA-seq. We identified 22 clusters in NPCs, hepatocytes, and myeloid cells isolated from non-Tg and *Fgf18* Tg mice (Fig. 6a, Supplementary Fig. 10b, Supplementary Table 2). *Lrat*⁺ HSC clusters comprised clusters 1, 3, and 8 (Fig. 6a), whereas *Acta2*⁺

myofibroblast and *Thy1*⁺ portal fibroblast clusters included clusters 16 and 12, respectively (Fig. 6a). Cell composition analysis of each cluster from non-Tg and *Fgf18* Tg mice revealed that the percentages of HSCs were increased in *Fgf18* Tg mice (Fig. 6b). Violin plots from pooled data from non-Tg and *Fgf18* Tg mice showed that several genes, including *Adamtsl2*, *Des*, *Fgfr2*, *Hgf*, and *Ngfr*, were specifically expressed in HSCs. In contrast, other genes, including *Tgfbr3*, *Tgfbi*, and *Tgfb1i1*, were ubiquitously expressed in both HSCs and fibroblasts (Fig. 6c, Supplementary Data 4).

Previous studies reported signature genes of quiescent (qHSCs) and activated HSCs (aHSCs) by scRNA-seq[10,11]. Notably, all *Lrat*⁺ HSC clusters expressed signature genes of qHSCs, including *Ank3*, *Colec11*, *Ecm1*, *Fcna*, *Gucy1b1*, *Reln*, and *Vipr1*, and signature genes of aHSCs, including *Col1a2*, *Col3a1*, *Lgals1*, *Thbs1*, and *Vim* (Supplementary Fig. 10c, Supplementary Data 4). In contrast, *Acta2*⁺ myofibroblast and *Thy1*⁺ portal fibroblast clusters expressed signature genes of aHSCs, including *Col1a2*, *Col3a1*, *Dpt*, *Igf1*, *Lgals1*, *S100a6*, and *Vim* (Supplementary Fig. 10c, Supplementary Data 4). These results suggest that FGF18 mainly expanded cell populations expressing qHSC marker genes.

To integrate the pathways triggered by FGF18 with the phenotypes of the livers in *Fgf18* Tg mice, we analyzed them using the cell–cell communication tool CellChat[42]. Of note, *Fgfr2* was expressed in HSCs, whereas *Fgfr1* was expressed in fibroblasts and minor populations of HSCs (Fig. 6c). Although read-through transcripts of *Fgf18* were expressed in various tissues from *Fgf18* Tg mice, FGF18 signals converged in HSCs and fibroblasts through Fgfr2 and Fgfr1, respectively (Fig. 6d, e). Similarly, the TGFβ1 signal merged in HSCs and portal fibroblasts (Fig. 6d, e). Moreover, several chemokines mainly produced by HSCs, including Ccl19, Cxcl12, and Cx3cl1, may induce signals in neutrophils, monocytes, dendritic cells, and B cells (Fig. 6d, e).

As our scRNA-seq did not contain liver sinusoidal endothelial cell (LSEC) and cholangiocyte, we integrated previously published scRNA-seq datasets from unstimulated liver NPCs with our scRNA-seq datasets[11]. The combined UMAP contained LSEC and cholangiocyte (Supplementary Fig. 11a). CellChat revealed that FGF18 may stimulate cholangiocyte through Fgfr1-3 (Supplementary Fig. 11b, c) and that *Vegfa* from HSC and cholangiocyte may induce signals in LSEC and HSC (Supplementary Fig. 11b, c). Together, these data may explain the mechanisms underlying the proliferation of CK19⁺ cholangiocytes and angiogenesis.

Regarding liver hypertrophy, FGF18 did not activate hepatocytes (Supplementary Fig. 12a, b). The expression of *Hgf* was elevated in the livers of *Fgf18* Tg mice, and *Lrat*⁺ HSCs highly expressed *Hgf* (Figs. 5b, 6c). Given that FGF18 did not upregulate the expression of *Hgf* in HSCs (Supplementary Fig. 12c), FGF18 may induce the proliferation of hepatocytes by HGF released from expanded HSCs that highly expressed *Hgf*. HGF induced ERK phosphorylation and *Jun*, *Ccnd1*, and *Myc* upregulation (Supplementary Fig. 12a, d). Therefore, these results suggest that FGF18 stimulates the proliferation of HSCs that provide an

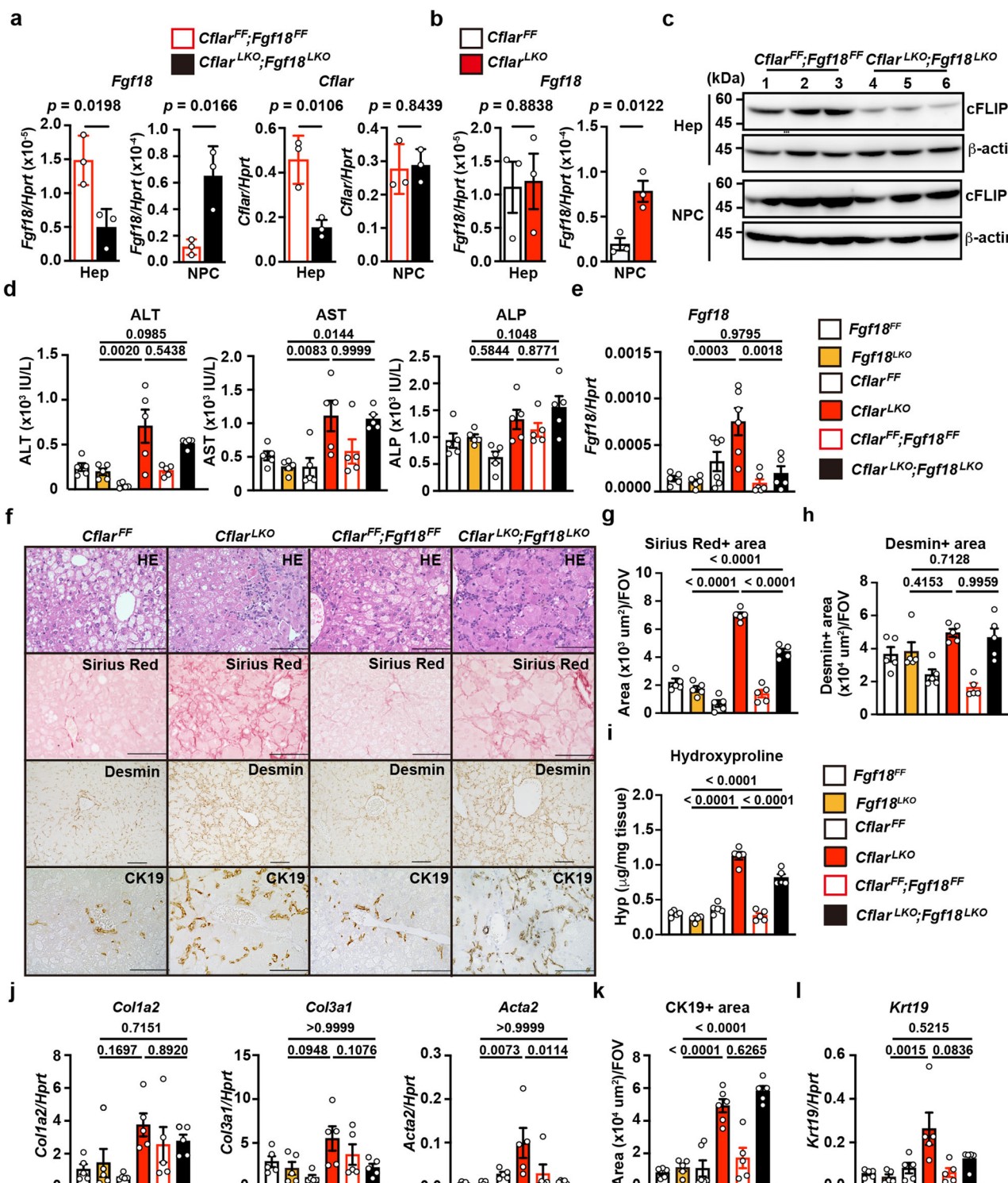

**Fig. 3 | Hepatocyte-specific deletion of *Fgf18* attenuates fibrosis in the livers of *Cflar^LKO* mice fed the CDE diet. a–c** Hepatocytes and NPCs were isolated from 6- to 8-week-old female *Cflar^FF;Fgf18^FF*, *Cflar^LKO;Fgf18^LKO*, *Cflar^FF*, and *Cflar^LKO* mice, and the expression of *Fgf18* and *Cflar* was determined by qPCR (**a, b**). Results are mean ± SE (*n* = 3 mice). The expression of cFLIP was determined by Western blotting with anti-cFLIP antibody (*n* = 3 mice) (**c**). Numbers indicate an individual mouse. Results are representative of two independent experiments. **d–l**, Eight-week-old female *Fgf18^FF*, *Fgf18^LKO*, *Cflar^FF*, *Cflar^LKO*, *Cflar^FF;Fgf18^FF*, and *Cflar^LKO;Fgf18^LKO* mice were fed the CDE diet for 4 weeks, and the serum ALT, AST, and ALP concentrations were determined (**d**). Results are mean ± SE (*n* = 5 mice). The expression of *Fgf18* in the livers was determined by qPCR (**e**). Results are mean ± SE. *n* = 5 (*Fgf18^FF*, *Fgf18^LKO*, *Cflar^FF*, *Cflar^LKO*, and *Cflar^FF;Fgf18^FF*) or *n* = 6 (*Cflar^LKO;Fgf18^LKO*) mice. Liver sections were

stained with H&E, Sirius Red, anti-desmin, or anti-CK19 antibodies (*n* = 5 mice) (**f**). Scale bars, 100 μm. The Sirius Red⁺ (**g**) and desmin⁺ (**h**) areas were quantified and expressed as in Fig. 1f. Results are mean ± SE (*n* = 5 mice). The hydroxyproline content of the liver was determined (**i**). Results are mean ± SE (*n* = 5 mice). Expression of the indicated genes in the liver was determined by qPCR (**j, l**). Results are mean ± SE (*n* = 5 mice). The CK19⁺ areas (**k**) were quantified and expressed as in Fig. 1f. Results are mean ± SE. *n* = 6 (*Cflar^FF* and *Cflar^LKO*,) or *n* = 5 (*Fgf18^FF*, *Fgf18^LKO*, *Cflar^FF;Fgf18^FF*, and *Cflar^LKO;Fgf18^LKO*) mice. Pooled results from six independent experiments are shown (**d, e, g–l**). Statistical significance was determined by the two-tailed unpaired Student's *t* test (**a, b**) or one-way ANOVA with Tukey's multiple comparisons (**d, e, g–l**). Source data are provided as a Source Data file.

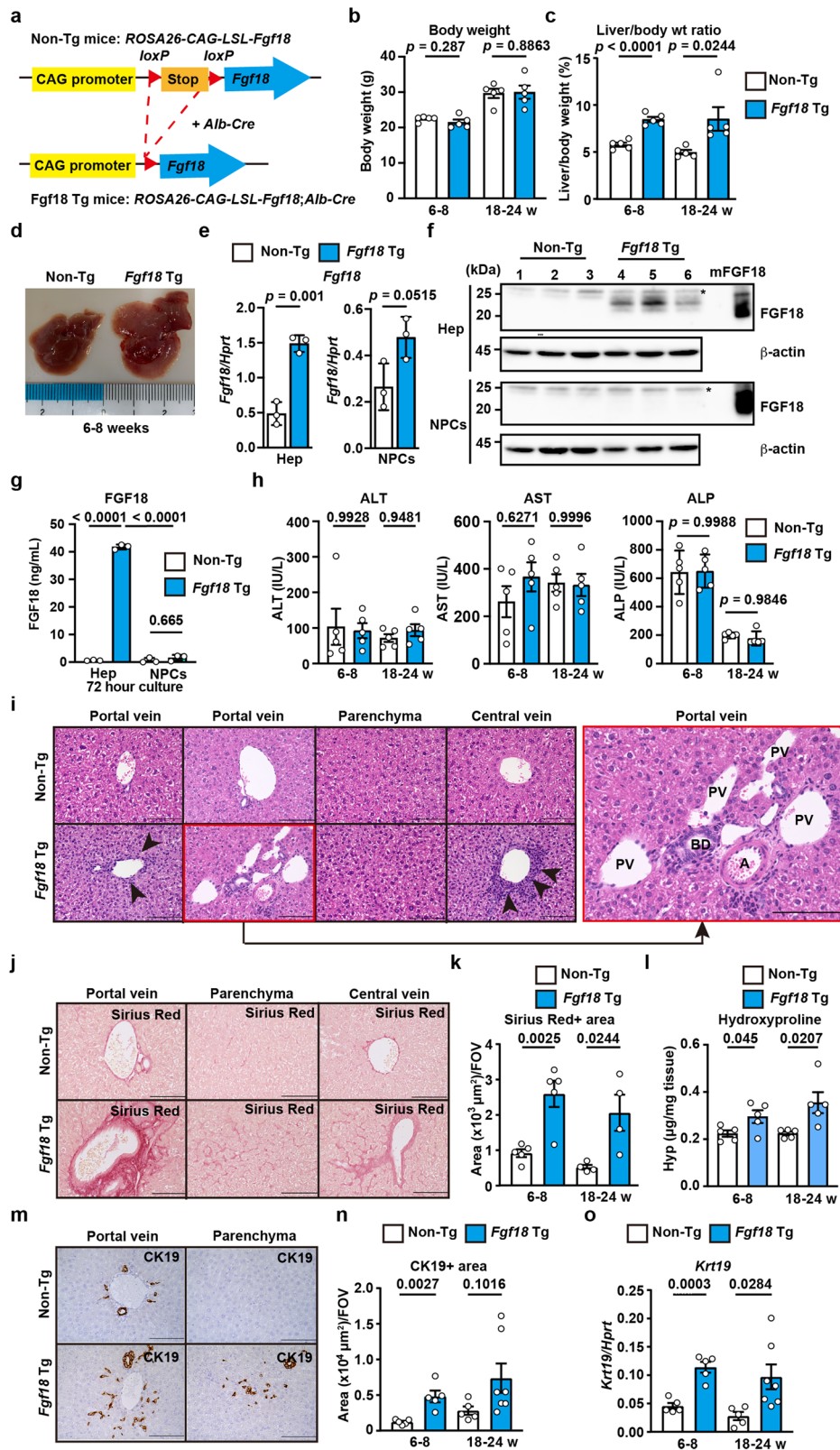

HGF-enriched microenvironment to stimulate hepatocyte proliferation, thereby causing liver hypertrophy.

## FGF18 stimulates the proliferation of HSCs but suppresses TGFβ-induced profibrotic gene expression

To further investigate the mechanisms by which FGF18 promotes liver fibrosis, we isolated primary HSCs by the Nycodenz density gradient centrifugation method[43]. Based on the determination of UV-positive or desmin-positive cells, the percentages of HSCs were approximately 80 to 90% (Supplementary Fig. 13a–d). We stimulated HSCs with FGF18 or TGFβ1 as a positive control to induce the upregulation of profibrotic genes[44]. FGF18 did not upregulate the expression of *Col1a2, Col3a1*, or *Acta2*, but TGFβ1 did (Fig. 7a). Likewise, based on the anti-α-SMA staining, TGFβ1 induced strong morphological changes, such as loss of

**Fig. 4 | Overexpression of *Fgf18* in hepatocytes promotes liver fibrosis.**
**a** Strategy for liver-specific expression of *Fgf18*. Deletion of the stop codon by *albumin-Cre* recombinase results in the expression of *Fgf18* in hepatocytes. **b**–**n** Non-Tg and *Fgf18* Tg mice were analyzed at the indicated times after birth. Body weight (**b**) and calculated liver/body weight ratios (%) of mice (**c**) are shown. Results are mean ± SE (*n* = 5 mice). Representative photos of the macroscopic appearance of the livers of 6- to 8-week-old mice (*n* = 10 mice) are shown (**d**). Hepatocytes and NPCs were isolated from 4-week-old non-Tg and *Fgf18* Tg mice, and the expression of *Fgf18* at both the mRNA (**e**) and protein levels (**f**) was analyzed by qPCR and Western blotting, respectively. Results are mean ± SE (*n* = 3 mice) (**e**). Each number indicates an individual mouse. mFGF18 indicates recombinant murine FGF18. Hepatocytes and NPCs were isolated as in (**e**) and cultured for 72 h (**g**). The concentrations of FGF18 were determined by the in-house ELISA. Results are mean ± SD of triplicate samples. Results are representative of two or three independent experiments (**e**–**g**). The ALT, AST, and ALP concentrations in the sera of mice (**h**). Results are mean ± SE (*n* = 5 mice). Liver sections were stained with H&E (*n* = 5 mice) (**i**). The right panel is an enlarged image of the left red box. A, artery; BD, bile duct; PV, portal vein. Liver sections were stained with Sirius Red (**j**), and the Sirius Red⁺ areas were quantified and expressed as in Fig. 1f (**k**). Results are mean ± SE. *n* = 5 (6–8 weeks) or *n* = 4 (18–24 weeks) mice. The hydroxyproline content was determined (**l**). Results are mean ± SE (*n* = 5 mice). Liver sections were stained with anti-CK19 (**m**) antibody, and the CK19⁺ areas were quantified and expressed as in Fig. 1f (**n**). The expression of *Krt19* was determined by qPCR (**o**). Results are mean ± SE. *n* = 5 (non-Tg and *Fgf18* Tg at 6–8 weeks and non-Tg at 18–24 weeks) or *n* = 7 (*Fgf18* Tg at 18–24 weeks) mice for **m**, **n**, **o**. Pooled results from two to three independent experiments (**b**, **c**, **h**, **k**, **l**, **n**, **o**). Statistical significance was determined by the two-tailed unpaired Student's *t* test (**b**, **c**, **e**, **h**, **k**, **l**, **n**, **o**) or two-way ANOVA with Tukey's multiple comparison (**g**). Source data are provided as a Source Data file.

a star-shaped appearance and elongated and spindle shaped morphology. In sharp contrast, FGF18-stimulated HSCs retained a star-like appearance, which was observed in untreated HSCs (Supplementary Fig. 14). Surprisingly, FGF18 stimulated the proliferation of HSCs and upregulated the expression of *Ccnd1*, encoding cyclin D1, which is involved in cell cycle progression[45] (Fig. 7b, c). Moreover, MEK and Akt inhibitors completely and moderately blocked FGF18-induced *Ccnd1* expression (Fig. 7d), respectively, suggesting that the MEK/ERK pathway mainly contributes to FGF18-induced *Ccnd1* expression. Since *Fgfr2* was selectively expressed in HSCs (Fig. 6c) and its expression was higher than that of *Fgfr3* or *Fgfr4* (Fig. 7e), FGF18 may transmit signals through FGFR2.

A previous study reported that bile acids increase the expression of another FGF protein family member, FGF15/19[46]. The serum concentrations of various bile acids were elevated in the sera of *Cflar^LKO* mice fed the CDE diet (Supplementary Fig. 15a). Nevertheless, bile acids did not upregulate *Fgf18* expression in primary hepatocytes (Supplementary Fig. 15b). In contrast, we found that TGFβ1 induced the expression of *Fgf18* in primary hepatocytes and HSCs (Fig. 7f, g), whereas FGF18 suppressed TGFβ1-induced profibrotic gene expression as well as the expression of *Tgfb2* and *Tgfb3* in HSCs (Fig. 7h, i). Thus, the expression levels of *Fgf18* and *Tgfbs* and their target genes were intimately regulated by each other. As expected, FGF18 did not upregulate the expression of profibrotic genes in hepatocytes, whereas TGFβ did activate them, although the induction levels were very low (Fig. 7j, k). Given that the basal expression levels of profibrotic genes in HSCs were higher than those in hepatocytes (Fig. 7a vs. 7j, 7k), the expansion of HSCs by FGF18, even at the relatively quiescent stages, may contribute to liver fibrosis.

### Expression of *FGF18* is correlated with the expression of *COL1A1* and *ACTA2* in human liver biopsy samples

To determine whether our data obtained from murine experiments are relevant to human liver fibrosis, we examined the expression of *FGF18* in liver biopsy samples to diagnose various human liver diseases (Supplementary Table 3). Although the expression levels of *FGF18* in the livers varied among diseases, the expression of FGF18 was correlated with that of *COL1A1* and *ACTA2* (Fig. 8a). These results suggest that FGF18 contributes to the development of liver fibrosis in humans as well as mice.

We propose the following model of how FGF18 promotes liver fibrosis. In response to various injuries, TGFβ is released from different types of cells, such as macrophages engulfing apoptotic hepatocytes. Then, TGFβ induces the production of FGF18 in HSCs and hepatocytes. FGF18 then stimulates the proliferation of HSCs. Proliferating HSCs further respond to stimuli derived from immune cells, such as scar-associated macrophages, and produce collagens and extracellular matrix, culminating in the development of liver fibrosis (Fig. 8b).

## Discussion

In the present study, we found that the expression of *Fgf18* was elevated in various murine liver fibrosis models. Deletion of *Fgf18* in hepatocytes attenuated CDE-induced fibrosis in *Cflar^LKO* mice; conversely, overexpression of *Fgf18* promoted liver fibrosis by increasing the number of HSCs. Mechanistically, FGF18 stimulated the proliferation of HSCs. Furthermore, the expression of *FGF18* was correlated with the expression of *COL1A1* and *ACTA2* in human liver biopsy samples. Thus, FGF18 is a critical growth factor that promotes liver fibrosis and may be a therapeutic target for treating liver fibrosis.

Few apoptotic hepatocytes were detected in the livers of *Cflar^LKO* mice under homeostatic conditions. *Cflar*-deficient hepatocytes are susceptible to death ligand-induced apoptosis[28,30], suggesting that death ligands expressed under homeostatic conditions trigger hepatocyte apoptosis. Notably, *Cflar^LKO* mice developed mild fibrosis, with an increase in CK19⁺ cells in the liver. Although apoptosis promotes liver fibrosis under various conditions[31], slight and persistent apoptosis of hepatocytes may be sufficient to induce mild fibrosis. Since macrophages release TGFβ upon engulfing apoptotic cells[47], the TGFβ released from macrophages stimulates the upregulation of *Fgf18* expression in HSCs. Indeed, the expression of *Fgf18* was higher in isolated NPCs (presumed HSCs) from *Cflar^LKO* mice, even when fed the normal diet, compared to those from *Cflar^FF* mice. This suggests that the slightly increased levels of *Fgf18* in NPCs (presumed HSCs) may contribute to, at least in part, fibrosis in *Cflar^LKO* mice under normal dietary conditions. Moreover, preexisting mild fibrosis and increased susceptibility to death ligand-induced apoptosis of hepatocytes may drastically exacerbate the development of CDE-induced liver fibrosis in *Cflar^LKO* mice.

Although previous studies have demonstrated an intimate relationship between FGF18 and tumors[48–50], the link between FGF18 and fibrosis remains unknown. Given that HSCs play a central role in liver fibrosis[2], we can surmise that FGF18 activates HSCs, thereby promoting liver fibrosis. Although hematopoietic progenitor cells and CD31⁺ endothelial cells express CD34 under normal conditions, HSCs express CD34 in CCl₄-induced liver injury[51]. The scRNA-seq analysis revealed that CD31⁻ CD34⁺ cells are composed of *Lrat*⁺ HSCs, *Acta2*⁺ myofibroblasts, and *Thy1*⁺ fibroblasts, suggesting that CD34 is a common surface marker of HSCs and fibroblasts, at least in the liver. While *Lrat*⁺ HSCs mostly expressed *Fgfr2*, *Acta2*⁺ and *Thy1*⁺ fibroblasts expressed *Fgfr1*, some minor populations of *Lrat*⁺ HSCs expressed *Fgfr1*. Thus, FGF18 likely induces signals in HSCs through FGFR2 in vitro and in vivo. Regarding the contribution of FGF18 in hepatocytes, deletion of *Fgf18* in hepatocytes moderately attenuated CDE-induced liver fibrosis. However, the effect was not drastic, suggesting that FGF18 produced by HSCs may compensate for the lack of FGF18 in hepatocytes. Hence, it would be interesting to test whether liver fibrosis is attenuated in mice lacking *Fgf18* in HSCs by crossing *Fgf18^flox/flox* mice with *Lrat-Cre* Tg mice.

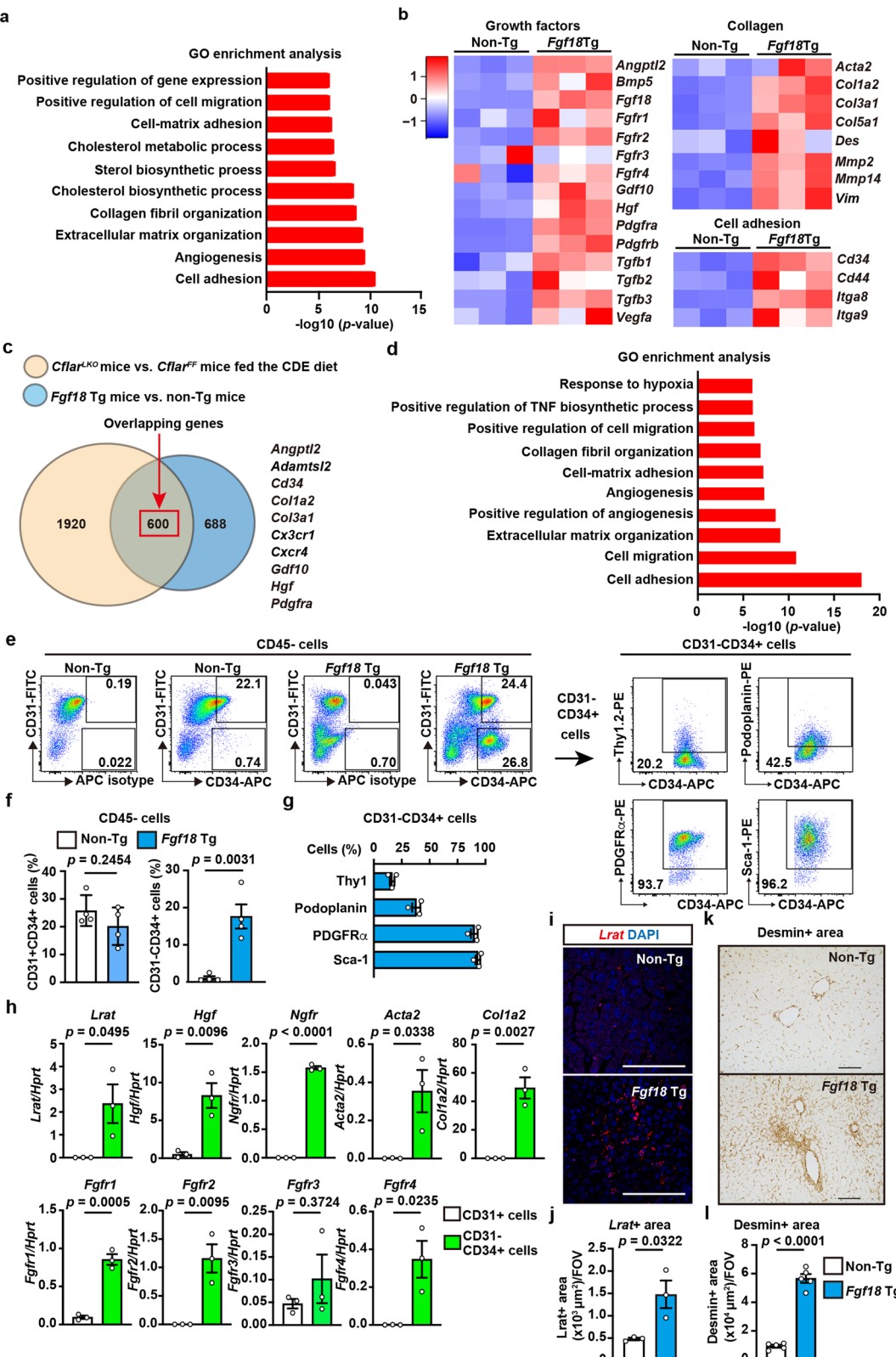

The scRNA-seq analysis revealed that the numbers of *Lrat*+ HSCs were increased in *Fgf18* Tg mice and that *Lrat*+ HSCs mostly expressed quiescent HSC marker genes. *Lrat*+ HSCs expressed several activated HSC markers, including *Col1a2, Col3a1, Igf1,* and *Lgals1*, although their expression levels were relatively low compared to those of *Acta2*+ or *Thy1*+ fibroblasts. Moreover, *Lrat*+ HSCs did not express *Timp1* and *Tnc*, which were elevated in HSCs from CCl₄-treated livers[10,42]. Thus, FGF18

mainly induces the expansion of weakly activated or quiescent *Lrat*+ HSCs and does not fully promote terminal differentiation toward *Acta2*+ myofibroblasts.

In addition to fibrosis, the transgenic expression of *Fgf18* in hepatocytes resulted in liver hypertrophy, consistent with a previous study[52]. The expression of *Fgf18* is elevated in various human cancers, and its expression correlates with poor prognosis[49,50]. FGF18 released

**Fig. 5 | FGF18 increases the number of CD31⁻CD34⁺ stromal cells. a** GO enrichment analysis of the RNA-seq results of the whole livers of 8-week-old non-Tg and *Fgf18* Tg mice (*n* = 3 mice). **b** Heatmap showing Z score scaled expression levels of representative genes in the indicated categories. **c, d** Genes upregulated more than 2-fold in the livers of *Cflar^LKO* mice vs. *Cflar^FF* mice fed the CDE diet for 4 weeks and *Fgf18* Tg mice vs. non-Tg mice were extracted and analyzed by the COUNTIF function. A Venn diagram of the upregulated genes in the livers of mice and representative overlapping genes (**c**). GO enrichment analysis of overlapping genes upregulated in both groups is shown (**d**). **e–h** Characterization of CD31⁻CD34⁺ cells. Liver nonparenchymal cells were prepared and analyzed by flow cytometry gated on CD45⁻ cells as in Supplementary Fig. 7g (**e**, left panel) and the percentage of each cell population among CD45- cells was calculated (**f**). Results are mean ± SE (*n* = 4 mice). The expression (**e**, right panel) and the percentages (**g**) of each cell

population among CD31⁻CD34⁺ cells were calculated and are shown. Results are mean ± SE (*n* = 3 mice). Gene expression in CD31⁻CD34⁺ cells vs. CD31⁺ cells (**h**). CD31⁺ and CD31⁻CD34⁺ cells were sorted as in Supplementary Fig. 8a. Expression of the indicated genes in the sorted cells was analyzed by qPCR. Results are mean ± SE (*n* = 3 mice). **i–l** *Lrat* and desmin⁺ cells are increased in the livers of *Fgf18* Tg mice. Liver tissue sections from 8-week-old non-Tg and *Fgf18* Tg mice were analyzed by RNAscope (**i, j**) or IHC (**k, l**). Scale bar, 100 μm. The *Lrat*⁺ or desmin⁺ areas were calculated and are expressed as *Lrat*⁺ or desmin⁺ areas per FOV. Results are mean ± SE (**j**, *n* = 3 mice; **l**, *n* = 5 mice). Statistical significance was determined by one-sided Fisher's exact test (**a, d**), or the two-tailed unpaired Student's *t* test. Representative results of four (**e**, left panel), three (**e**, right panel), and two (**h**) independent experiments. Source data are provided as a Source Data file.

from tumor cells induces their proliferation in an autocrine or paracrine manner[48]. Under the experimental conditions of the present study, FGF18 did not activate primary hepatocytes. The expression of *Hgf*, which is a critical cytokine responsible for hepatocyte proliferation, was elevated in the livers of both *Cflar^LKO* mice fed the CDE diet and *Fgf18* Tg mice, but FGF18 did not induce the upregulation of *Hgf* in HSCs. Given that the expression levels of *Hgf* were relatively high in HSCs under homeostatic conditions, FGF18 induced the proliferation of hepatocytes, not through the upregulation of *Hgf* but by stimulating the proliferation of HSCs. Moreover, FGF18 is involved in the proliferation and tube formation of endothelial cells[48]. Consistent with previous studies, aberrant angiogenesis occurred in the livers of *Fgf18* Tg mice. This finding was partly because *Vegfa* from cholangiocytes and HSCs may stimulate LSECs.

Notably, TGFβ upregulates the expression of profibrotic genes in HSCs but does not trigger the proliferation of HSCs. Activation of HSCs, rather than their strong expansion, seems sufficient to induce mild liver fibrosis, as observed in wild-type mice fed a CDE diet for 4 weeks, or mice fed the normal diet or the Western diet treated with CCl₄ for 12 weeks. In contrast, as livers progress to advanced fibrosis, it is reasonable to speculate that the expansion of HSCs may become necessary. When the strength of the stimuli exceeds a certain threshold, the expression of *Fgf18* is strongly induced. This situation was observed in the livers of *Cflar^LKO* mice fed the CDE diet for 4 weeks, mice on the normal diet or the Western diet treated with CCl₄ for 24 weeks. Assuming that expansion of HSCs requires some space, it is reasonable to speculate that FGF18 suppresses the production of extracellular matrix to make a space for HSC proliferation. Moreover, it is worth noting that the expression levels of profibrotic genes in HSCs were relatively higher than those in hepatocytes, even if its expression levels were declined in the presence of FGF18. This finding indicates that the increased HSCs by FGF18 stimulation play an important role in the accumulation of collagens and extracellular matrix. Moreover, previous studies reported that scar-associated macrophages differentiate from circulating monocytes, are recruited to the fibrotic niche, and interact with PDGFRα⁺ collagen-producing mesenchymal cells, resulting in the activation of several profibrogenic pathways[53,54]. Indeed, FGF18 induced the expression of inflammatory cytokines and chemokines, thereby recruiting immune cells, including scar-associated macrophages. Thus, proliferating HSCs induced by FGF18 further respond to profibrotic stimuli derived from scar-associated macrophages, such as platelet-derived growth factor (PDGF) and IL-1β[53], ultimately contributing to the development of liver fibrosis.

When focusing on a role of FGF18 in suppressing the expression of profibrotic genes, it appears that FGF18 may mitigate liver fibrosis under specific conditions, such as the transient expression of *Fgf18* as noted by Tong et al.[55]. However, in the long term, FGF18 seems to stimulate HSC proliferation, and subsequently, the proliferating HSCs become responsive to subsequent profibrotic stimuli as described above, ultimately leading to liver fibrosis. From a therapeutic perspective, it would be intriguing to investigate the effect of shRNA-

mediated deletion of *Fgf18* in adult mice with liver fibrosis. Further study will be required to address this issue.

We and others reported that FGF18 suppressed the expression of *Tgfb2* and *Tgfb3* and TGFβ-induced expression of profibrotic genes[55]. In contrast, TGFβ induced upregulation of *Fgf18*. A previous study showed that TGFβ inhibits the phosphorylation of YAP, thereby preventing degradation of YAP and subsequently leading to fibrosis[55]. Notably, FGF18 counteracts TGFβ-mediated YAP dephosphorylation, facilitating the degradation of phosphorylated YAP and preventing liver fibrosis[55]. However, the specific mechanisms underlying the FGF18-induced phosphorylation of YAP were not elucidated in that study. In contrast, we reported that FGF18 induced the upregulation of *Ccnd1*, which was completely abolished in the presence of a MEK inhibitor. This finding suggests that the MEK/ERK pathway is crucial in this process. It remains unclear whether the MEK/ERK pathway and the YAP/TAZ pathway finally converge on the induction of proliferation-related or profibrotic genes. RNA-seq analysis using HSCs treated with TGFβ, FGF18, and a combination of the two could serve as a valuable tool to unravel the crosstalk between TGFβ and FGF18.

## Methods

All experiments were performed according to the guidelines approved by the Institutional Animal experiments Committee of Faculty of Medicine, Toho University (21-409 and 21-412) and Faculty of Medicine and Graduate School of Medicine, Juntendo University (250071). Human study was approved by the ethics committee of Faculty of Medicine and Graduate School of Medicine, Juntendo University (E22-0085-H01).

### Reagents

Human FGF18 (100-28, PeproTech), murine HGF (BL771601, BioLegend), insulin-transferrin-selenium (41400045, Gibco), cholic acid (CA) (C0324, Tokyo Chemical Industry), deoxycholic acid (DCA) (10712-12, Nacalai Tesque), glycocholic acid (GCA) (G2878, Sigma), taurocholic acid (TCA) (T4009, Sigma), human TGFβ1 (100-21, Peprotech), U0126 (CAS 109511-58-2, Calbiochem), and LY294002 (440206, Calbiochem) were purchased from the indicated sources. The following antibodies were used in this study and obtained from the indicated sources: anti-AKT (4691, Cell Signaling, 1:1000), anti-phospho-AKT (4060, Cell Signaling, 1:1000), anti-α-SMA (Ab124964, Abcam, 1:500), PE anti-mouse B220 (50-0452-U100, Tonbo Biosciences, 1:200), anti-APC micro-Beads (130-090-855, Miltenyi, 1:5), anti-PE micro-Beads UltraPure (130-105-639, Miltenyi, 1:5), anti-β-actin (sc-47778, Santa Cruz, 1:5000), anti-cleaved caspase 3 (9664, Cell Signaling, 1:1000), FITC anti-mouse CD3ε (35-0031-U100, Tonbo Biosciences, 1:200), PE anti-mouse CD11b (50-0112-U100, Tonbo Biosciences, 1:500), anti-mouse CD16/CD32 (Bio X Cell, BE0307, 1:200), APC anti-mouse CD31 (17-0311-82, Invitrogen, 1:100), FITC anti-mouse CD31 (102405, BioLegend, 1:200), PE anti-mouse CD31 (102407, BioLegend, 1:100), anti-CD34 (Ab81289, Abcam, 1:1000), APC anti-mouse CD34 (128611, BioLegend, 1:200), APC anti-mouse CD45.2 (558702, BD,

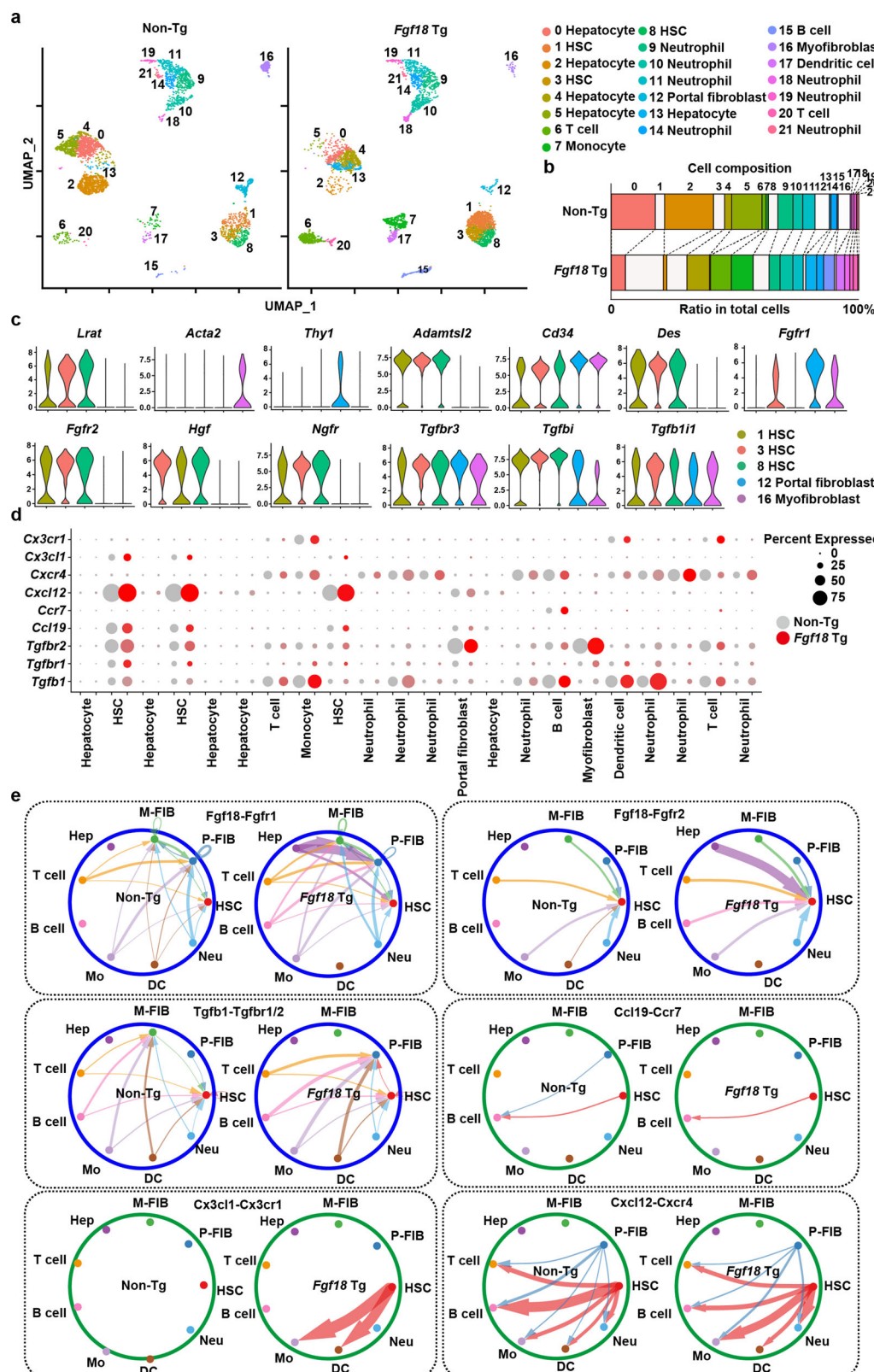

1:100), BV421 anti-mouse CD45.2 (109832, BioLegend, 1:200), PE anti-mouse CD45.2 (109808, BioLegend, 1:100), anti-CD68 (97778, Cell Signaling, 1:200), APC anti-mouse CD146 (134712, BioLegend, 1:100), anti-CK19 (in house, 1:200; MABT913, Merck, 1:1000), anti-desmin (AB32362, Abcam, 1:1000), APC anti-mouse Epcam (563478, BD biosciences, 1:100), anti-ERK (4695, Cell Signaling, 1:1000), anti-phospho-ERK (4370, Cell Signaling, 1:1000), FITC anti-mouse F4/80 (35-4801-

U100, Tonbo Biosciences, 1:500), anti-FGF18 (120525, Wuhan Huamei Biotech, 1:1000; 19S-SE5 and 12G7-9 were made in-house), anti-cFLIP (Dave-2, Adipogen, 1:1000), anti-Ki67 (Ab16667, Abcam, 1:200), anti-Ly-6G (87048, Cell Signaling, 1:200), APC anti-mouse Ly-6G (127614, BioLegend, 1:500), anti-p75NTR (Ngfr/Tnfrsf16) (AF1157, R&D, 1:2000), PE anti-mouse PDGFRα (562776, BD Biosciences, 1:200), PE anti-mouse podoplanin (127407, BioLegend, 1:200), PE anti-mouse Sca1 (561076,

**Fig. 6 | Characterization of CD31-CD34+ stromal cells and cell–cell communication analyzed by scRNA-seq. a** UMAP plot showing the 22 clusters of nonparenchymal liver cells from 6-week-old non-Tg and *Fgf18* Tg mice (*n* = 3 mice per genotype). Each number indicates each cell cluster. **b** Relative percentages of each cluster from non-Tg and *Fgf18* Tg mice are shown. Clusters containing HSC, portal fibroblast, and myofibroblast are indicated by the white boxes. **c** Violin plot showing the expression levels of the indicated genes in HSCs, portal fibroblasts, and myofibroblasts pooled from non-Tg and *Fgf18* Tg mice. **d** Circle plot showing the expression levels and percentages of cells expressing the indicated genes from non-Tg and *Fgf18* Tg mice. Color intensities and circle sizes indicate expression

levels and percentages of cells expressing the indicated genes, respectively. **e** Communication networks between HSCs/fibroblasts and other cells as analyzed by CellChat. Notably, FGF18 signals appeared to originate from various types of cells; this may be caused by read-through transcripts of *Fgf18* expressed in multiple tissues. All significant ligand–receptor pairs that contribute to sending signals from HSC and fibroblast to other cells are shown. The edge width represents the communication probability. scRNA-seq data for endothelial cell and cholangiocyte were not included in the present data. M-FIB myofibroblasts, P-FIB portal fibroblast, HSC hepatic stellate cell, Hep hepatocyte, DC dendritic cell, Mo monocyte, Neu neutrophil.

BD Biosciences, 1:200), APC anti-mouse Ter119 (20-5921-U100, Tonbo Biosciences, 1:100), PE anti-mouse Thy1.2 (105307, BioLegend, 1:200), Alexa 594-conjugated anti-vimentin (677804, BioLegend, 1:400), eBioscience™ Fixable Viability Dye eFluor™ 506 (65-0866-14, eBioscience, 1:5000), and 7-AAD (420404, BioLegend, 1:100). Anti-MHC class I sampletag 1 (1:50), anti-MHC class I sampletag 2 (1:50), anti-MHC class I sampletag 3 (1:50), anti-MHC class I sampletag 4 (1:50), anti-MHC class I sampletag 5 (1:50), and anti-MHC class I sampletag 6 (1:50) were purchased from BD Biosciences. Alexa 647-conjugated donkey anti-rabbit IgG (A31573, 1:500), Alexa 488-conjugated donkey anti-rabbit IgG (A21206, 1:500), and Alexa 488-conjugated donkey anti-goat IgG (A11055, 1:500) antibodies were purchased from Invitrogen. HRP-conjugated donkey anti-rabbit IgG (NA934, 1:5000) and HRP-conjugated sheep anti-mouse IgG (NA931, 1:5000) antibodies were purchased from GE Healthcare. Peroxidase AffiniPure donkey anti-rat IgG (712-035-153, 1:5000) antibody was purchased from Jackson ImmunoResearch. Biotinylated goat anti-rabbit IgG (E0432, 1:200) and streptavidin-HRP (P0397, 1:300) were purchased from DAKO. Streptavidin poly-HRP80 conjugate (65R-S119) was purchased from Stereospecific Detection Technologies GmBH.

## Mice
*Cflar^FF* mice were provided by Y.-W. He and have been described previously[56]. Exon 1 of the *Cflar* gene was flanked by two *loxP* sites and introduced into ES cells, resulting in the generation of *Cflar^FF* mice. *Fgf18^FF* mice (RBRC05691) were obtained from RIKEN Bioresource and were described previously[36]. Exon 3 of the *Fgf18* gene was flanked by two *loxP* sites and introduced into ES cells, resulting in the generation of *Fgf18^FF* mice. *Albumin-Cre* recombinase transgenic (*Alb-Cre*) mice (003574) and C57BL/6 mice were purchased from Jackson Laboratory and CLEA-Japan, respectively. To generate hepatocyte-specific *Cflar*-deficient mice, we crossed *Cflar^FF* mice with *Alb-Cre* mice, generating *Cflar^LKO* mice as described previously[29].

To generate hepatocyte-specific *Fgf18*-deficient mice, we crossed *Fgf18^FF* mice with *Alb-Cre* mice, generating *Fgf18^LKO* mice. To generate hepatocyte-specific *Cflar* and *Fgf18* double-deficient mice, we crossed *Cflar^FF;Alb-Cre* mice with *Fgf18^FF* mice, resulting in the generation of *Cflar^FF;Fgf18^FF;Alb-Cre (Cflar^LKO;Fgf18^LKO)* mice.

Mice were housed in 23 ± 2 °C, a humidity of 55% ± 5%, and a 12 h dark/light cycle. Only female mice were used for the CDE diet experiments, because our preliminary analysis showed that liver fibrosis were more severe in female mice than in male mice. Only male mice were used for the DDC diet experiments because male mice generally give more reproducible results due to the lack of estrous cycle. Feeding experiments began with 8-week-old mice, lasting either for a duration of 4 weeks (for the CDE and DDC diets) or for 12 weeks (for the CDAHFD diet). Mice fed the normal diet were analyzed at 12 weeks. To collect sera and tissue samples, mice were first anesthetized by isoflurane inhalation and then euthanized by cervical dislocation. To isolate liver cells, mice were anesthetized by intraperitoneal injection of a mixture of midazolam, medetomidine, and butorphanol prior to liver perfusion. Both male and female mice were used for other experiments. Non-Tg and *Fgf18*Tg mice were fed the normal diet and analyzed at young (6-8 weeks) and adult (18-24 weeks) ages.

All experiments were performed according to the guidelines approved by the Institutional Animal Experiments Committee of Faculty of Medicine, Toho University School of Medicine (21-409 and 21-412) and Faculty of Medicine and Graduate School of Medicine, Juntendo University (250071).

## Generation of *Rosa 26-LSL-Fgf18* Tg mice
To generate mice expressing *Fgf18* in a tissue-specific manner, *Fgf18* cDNA was inserted into the CTV vector (*CAG* promoter-*loxP*-STOP-*loxP*-*Fgf18* cDNA-polyA) (Addgene #15912)[57]. To induce homologous recombination, we used the *CRISPR/Cas9* system; two sets of sgRNA oligos (5′- caccgTGGGCGGGAGTCTTCTGGGC-3′ and 5′- aaacGCCCAGAAGACTCCCGCCCAc-3′ for pX335-*Rosa26*-3, and 5′- caccGACTGGAGTTGCAGATCACG -3′ and 5′- aaacCGTGATCTGCAACTCCAGTC -3′ for pX335-*Rosa26*-4) were cloned into *BbsI*-digested pX335-U6-Chimeric_BB-CBh-h*SpCas9n* (D10A) plasmid (Addgene #42335). These vectors were electroporated into a feeder-free KTPU8 ES cell line derived from the TT2 ES cell line using a previously described method[58]. Briefly, KTPU8 cells were electroporated using a Gene Pulser set (Bio-Rad). G418 was introduced into the medium after 48 h, and cells were subsequently cultured for an additional 7 days. After G418 selection, several ES clones in which a single transgene was integrated into the *Rosa26* locus were obtained. Chimeric mice were produced by aggregation of ES cells with eight-cell embryos of ICR mice. Chimeras were crossed with C57BL/6N mice, resulting in three lines of *Fgf18* Tg mice. *CAG* promoter-driven *Fgf18* expression was interrupted by the insertion of the *neomycin* gene flanked by the *loxP* sites in the reverse orientation. Thus, *Cre* recombinase-dependent deletion of the *neomycin* gene results in tissue-specific expression of *Fgf18*. To express *Fgf18* in hepatocytes, we crossed *Rosa 26-LSL-Fgf18* Tg mice with *albumin-Cre* Tg mice, resulting in the generation of mice with hepatocyte-specific expression of *Fgf18* in mice, which were referred to as *Fgf18* Tg mice.

## Induction of hepatitis
For the CDE diet, eight-week-old female mice were housed individually and fed the choline-deficient diet (02960034, MP Biomedicals) and drinking water containing 0.15% DL-ethionine (E5139, Sigma) for 4 weeks. For the DDC diet, 8-week-old male mice were fed the diet containing 0.1% 3,5-diethoxycarbonyl-1,4-dihydrocollidine (DDC) (CLEA Japan) for 4 weeks. For CDAHFD, eight-week-old mice were fed the CDAHFD diet (A06071302, Research Diets) for 12 weeks.

## Measurement of serum alanine aminotransferase (ALT), aspartate aminotransferase (AST), and alkaline phosphatase (ALP) concentrations
Serum ALT, AST, and ALP concentrations were determined by Oriental Yeast Co. Ltd.

## Histological, immunohistochemical, and immunofluorescence analyses
The livers were fixed in 10% formalin or 4% paraformaldehyde and embedded in paraffin blocks. Paraffin-embedded liver sections were used for H&E staining and immunohistochemical and

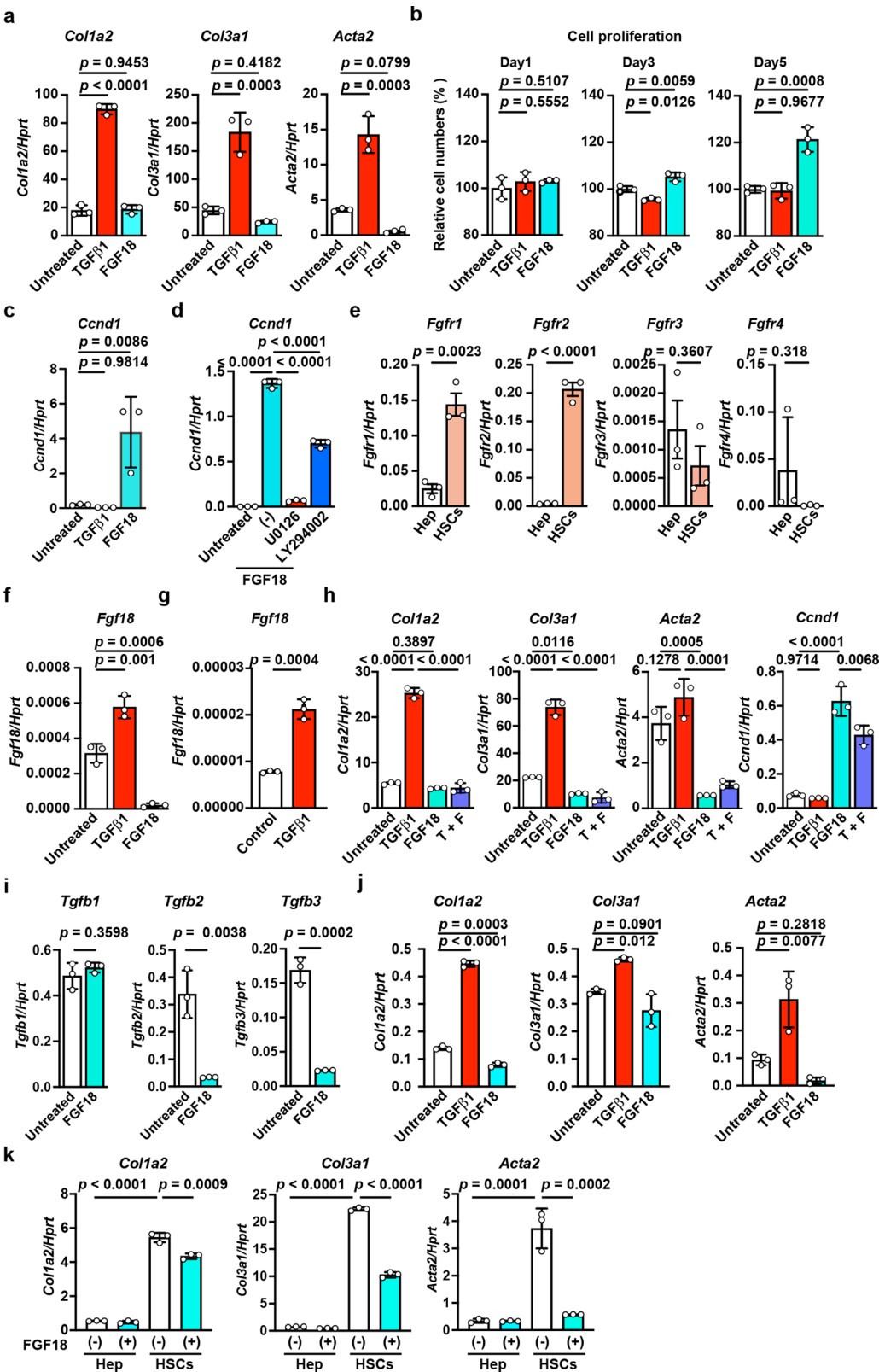

immunofluorescence analyses. Paraffin-embedded liver sections were stained with H&E, anti-CC3, anti-Ki67, anti-CK19, anti-desmin, anti-Ly-6G, anti-CD68, and anti-CD34 antibodies, Sirius Red, or the TUNEL method. The TUNEL method was performed using an In Situ Cell Death Detection Kit (Sigma–Aldrich) according to the manufacturer's instructions. Pictures were obtained by an All-in-One

microscope (BZ-X700, KEYENCE). The numbers of CC3[+] and TUNEL[+] cells were counted manually. The Sirius Red[+], CD34[+], desmin[+], CD68[+], Ki67[+], and CK19[+] areas were automatically calculated by hybrid cell count (BZ-X700, KEYENCE).

For immunofluorescence analysis, tissue sections were pre-incubated with the MaxBlock™ Autofluorescence Reducing Kit

**Fig. 7 | FGF18 stimulates the proliferation of HSCs but inhibits TGFβ-induced upregulation of profibrotic genes.** HSCs (a–f, h, i, k) and hepatocytes (e, g, j, k) were isolated from wild-type mice as described in the Methods. **a** TGFβ1, but not FGF18, upregulates the expression of profibrotic genes. HSCs were untreated or stimulated with TGFβ1 (1 ng/mL) or FGF18 (100 ng/mL) for 24 h. The expression of the indicated genes was analyzed by qPCR. Results are mean ± SD of triplicate samples. **b** FGF18 stimulates the proliferation of HSCs. HSCs were left untreated or stimulated as in (a) for the indicated times. Cell proliferation was analyzed by the WST assays. **c**, FGF18 upregulates the expression of *Ccnd1*. HSCs were stimulated and analyzed as in (a). **d** MEK and Akt inhibitors completely and moderately abolish the FGF18-induced upregulation of *Ccnd1*, respectively. HSCs were stimulated as in (a) in the absence or presence of U0126 (10 μM) or LY294002 (10 μM) for 24 h and were analyzed as in (a). **e** Expression of *Fgfr1-4* in HSCs and hepatocytes. The expression of *Fgfr*1-4 was determined by qPCR. Results are mean ± SEM (*n* = 3 mice).

**f, g** TGFβ1 induces upregulation of *Fgf18* in HSCs (f) and hepatocytes (g). HSCs and hepatocytes were stimulated and analyzed as in (a). **h** FGF18 suppresses the TGFβ1-induced upregulation of profibrotic genes. HSCs were untreated or stimulated with TGFβ1 (1 ng/mL), FGF18 (100 ng/mL), or both and were analyzed as in (a). **i** FGF18 suppresses the expression of *Tgfb1-3*. HSCs were stimulated and analyzed as in (a). **j** Hepatocytes were stimulated and analyzed as in (a). **k** Hepatocytes and HSCs were untreated or stimulated individually with FGF18 (100 ng/mL) as in (a). Statistical significance was determined by the one-way ANOVA with Dunnett's multiple comparison test (**a, b, c, f, j**) or one-way ANOVA with Tukey's multiple comparison test (**d, h**), two-tailed unpaired Student's *t* test (**e, g, i**), or two-way ANOVA with Tukey's multiple comparison test (**k**). All results are representative of two to three independent experiments. Source data are provided as a Source Data file.

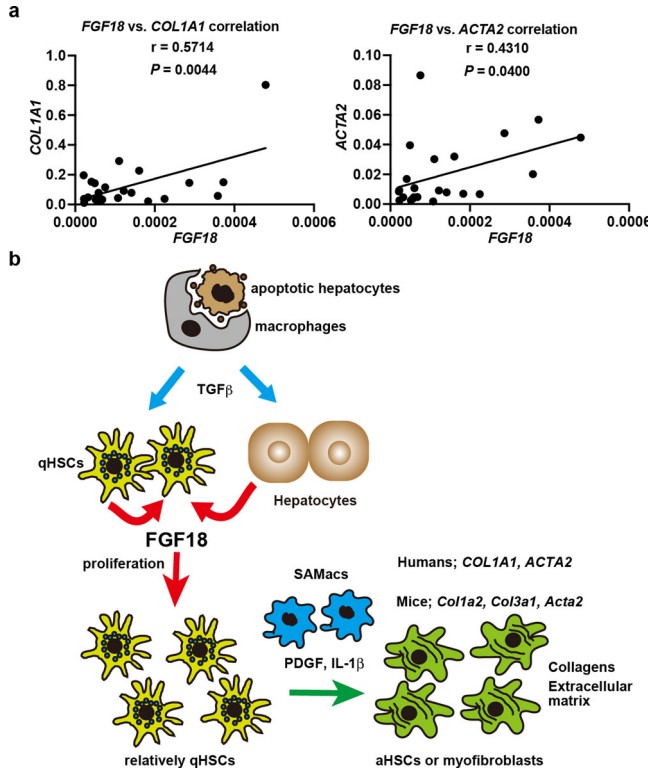

**Fig. 8 | Expression of *FGF18* is correlated with the expression of *COL1A1* and *ACTA2* in human liver biopsy samples. a** RNA was extracted from liver biopsy samples, and the expression of the indicated genes was analyzed by qPCR (*n* = 23). The correlation between *FGF18* and *COL1A1* or *ACTA2* was determined by Pearson correlation coefficient analysis. *P* values were calculated two-sided test. Source data are provided as a Source Data file. **b** A model for FGF18-induced liver fibrosis. In response to various injuries, TGFβ is released from different types of cells, such as macrophages engulfing apoptotic hepatocytes. Then, TGFβ induces the production of FGF18 in HSCs and hepatocytes. FGF18 then stimulates the proliferation of HSCs. Proliferating HSCs further respond to stimuli derived from scar-associated macrophages and produce collagens and extracellular matrix, culminating in the development of liver fibrosis. aHSCs, activated HSCs; PDGF, platelet-derived growth factor; qHSCs, quiescent HSCs; SAMacs, scar-associated macrophages.

(MaxVision Biosciences) according to the manufacturer's instructions. Then, paraffin-embedded sections were treated with Instant Citrate Buffer Solution (RM-102C, LSI Medicine) to retrieve the antigen. Blocking was performed with Blocking One Histo (06349-64, Nacalai Tesque) and 5% donkey serum. Liver tissue sections were stained with the indicated antibodies, followed by visualization with Alexa-conjugated secondary antibodies. Confocal microscopy was performed on an LSM 880 (Zeiss). Images were processed and analyzed using ZEN software (Zeiss).

### Nile red staining
Frozen liver sections were first fixed with 10% formaldehyde neutral buffer solution for 1 h at room temperature, followed by staining with 1 μg/mL Nile Red (144-08811, WAKO) in PBS for 15 min at room temperature. After three washes with PBS, samples were mounted with Vectashield mounting medium with DAPI (H-1200, VECTOR).

### Sirius red staining
Sirius Red/Direct Red 80 (D4132, Tokyo Chemical Industry) was dissolved at 1 g/L in a saturated aqueous picric acid solution. Dewaxed and hydrated paraffin sections were stained in the solution for 1 h and then washed twice with 0.5% acetate in water.

### Quantitative polymerase chain reaction assays
Total RNA was purified from mouse livers and sorted cells using Sepasol-RNA I SuperG (09379-55, Nacalai Tesque) and RNeasy Micro Kit (74004, QIAGEN), respectively. Then, cDNAs were synthesized with the Revertra Ace qPCR RT Kit (FSQ-101, Toyobo). Quantitative polymerase chain reaction (qPCR) analysis was performed with the 7500 Real-Time PCR detection system with the SYBR green method for the target genes and an endogenous control, murine *Hprt*, or human *GAPDH*, with 7500 SDS software (Applied Biosystems). The primers used in this study are included in Supplementary Table 4.

### Measurement of the hydroxyproline content
The concentrations of the hydroxyproline were determined by a Hydroxyproline Assay Kit according to the manufacturer's instructions (STA-675, Cell BIOLABS). Frozen liver tissues (50-100 mg) were acid-hydrolyzed with 10 volumes (w/v) of 6 N HCl at 95 °C for 24 h. After centrifugation, the supernatant was used for the colorimetric assay.

### Hydrodynamic tail vein injection (HTVi)
To express *Egfp* for the negative control, murine *Fgf18, Tnfsf13b*, and *Wnt5a* in hepatocytes, we amplified these cDNAs by RT–PCR and subcloned them into the pLIVE vector (MIR5420, Mirus Bio). The pLIVE vector consists of the *albumin* promoter and *alpha-fetoprotein* enhancer II, thereby expressing the gene specifically in hepatocytes at high levels for a relatively long period in vivo. HTVi was performed according to the standard procedure[32]. Briefly, 20 μg of pLIVE-*Egfp*, *Fgf18, Tnfsf13b*, or *Wnt5a* was diluted in 2 mL of TransIT-EE Hydrodynamic Delivery solution (MIR5340, Mirus Bio) and injected into 8-week-old mice through the tail vein. The expression of these genes in the liver was verified by qPCR, and liver sections were analyzed by immunohistochemistry with anti-CK19, anti-desmin, anti-GFP, or anti-Ki67 antibodies one week after injection.

### RNAscope
Mouse livers were first fixed with buffered 10% formaldehyde neutral buffer at room temperature for 16−32 h and embedded in paraffin. Dewaxed and hydrated liver sections were processed using the

RNAscope Multiplex Fluorescent Reagent Kit v2 (323100, Advanced Cell Diagnostics) and TSA Plus Cyanine 3 and Cyanine 5 kits (NEL744001KT and NEL745001KT, respectively, Akoya Biosciences) according to the manufacturer's instructions. RNAscope target probes for mouse *Fgf18* (495421), *Lrat* (460641-C2), and *Hnf4a* (497651-C3) were purchased from Advanced Cell Diagnostics. Fluorescence images were obtained by either BZ-X700 (Keyence) or LSM880 (Zeiss), and fluorescent areas were calculated by Hybrid Cell Count (Keyence) or Adobe Photoshop software. To identify cells expressing *Fgf18* mRNA, merged images were used to determine whether *Fgf18* mRNA puncta were located close to *Lrat* mRNA puncta (HSC origin) or *Hnf4a* mRNA puncta (hepatocyte origin). The number of *Fgf18* mRNA puncta in HSCs and hepatocytes was manually counted and statistically analyzed.

### Isolation of primary hepatocytes and nonparenchymal liver cells

To purify nonparenchymal liver cells, we performed a modified two-step collagenase perfusion method as described previously[59]. Briefly, the livers from 6- to 8-week-old wild-type, non-Tg, and *Fgf18* Tg mice were perfused with liver perfusion buffer (LPB) 1 (136 mM NaCl, 5.4 mM KCl, 0.5 mM EGTA, 0.5 mM NaH$_2$PO$_3$ 2H$_2$O, 0.42 mM Na$_2$HPO$_3$, 10 mM HEPES pH 7.5, 5 mM glucose and 4.2 mM NaHCO$_3$) at a flow rate of 3 ml/min for 5 min. Then, the livers were perfused with LPB2 (136 mM NaCl, 5.4 mM KCl, 5 mM CaCl$_2$, 0.5 mM NaH$_2$PO$_3$ 2H$_2$O, 0.42 mM Na$_2$HPO$_3$, 10 mM HEPES pH 7.5, 5 mM glucose and 4.2 mM NaHCO$_3$) containing 0.5 mg/ml collagenase type IV (C5138, Sigma–Aldrich) and 0.06 mg/ml DNase I (DN25, Sigma-Aldrich) at a flow rate of 3 ml/min for 8 min. The digested livers were transferred to a glass dish, and fibrous connective tissues were removed from the livers with a pair of tweezers. For *Fgf18* Tg mice, the perfusion of livers with LPB2 alone was insufficient to isolate nonparenchymal cells from fibrotic livers. Thus, we further incubated the livers with LPB2 after perfusion using a stirrer bar at 37 °C for 5 min. Cells were dispersed with tweezers and passed through a 70 μm cell strainer. The supernatant was transferred to a new tube after centrifugation at 50 × *g* for 2 min. The pellet was resuspended in DMEM and centrifuged at 50 × *g* for 2 min; then, the pellet was used as hepatocytes. The new supernatant was combined with the former supernatant and then centrifuged at 100 × *g* for 2 min several times until no cell pellet was visible. The final supernatant was centrifuged at 660 × *g* for 5 min, and the pellets were resuspended in RBC lysis solution (0.17 mM NH$_4$Cl, 0.01 mM EDTA, 0.1 M Tris, pH 7.3) and then washed twice in DMEM containing 10% FCS. Cells were then subjected to flow cytometry or cell sorting. For Western blotting, cells were lysed in lysis buffer as described below.

Hepatocytes were pelleted after centrifugation at 50 × *g* for 2 min, resuspended in DMEM containing 10% FCS, and plated on collagen I-coated 6-well microplates (4810-010, Iwaki Glass). The next day, the medium was changed to remove unattached dead hepatocytes. Cells were used for stimulation or Western blotting by directly adding lysis buffer as described below.

### Flow cytometry and cell sorting

Nonparenchymal liver cells were prepared as described above. Single-cell suspensions were incubated with anti-mouse CD16/CD32 antibody and then stained with the indicated antibodies. Cells were analyzed on an LSRFortessa X-20 (BD Biosciences) or sorted by a FACSAria Fusion (BD Biosciences). Data were processed with CellQuest software or FlowJo (BD Biosciences).

### Western blotting

Cells were lysed with RIPA buffer (50 mM Tris-HCl [pH 8.0], 150 mM NaCl, 1% Nonidet P-40, 0.5% deoxycholate, 0.1% SDS, 25 mM β-glycerophosphate, 1 mM sodium orthovanadate, 1 mM sodium fluoride, 1 mM phenylmethylsulfonyl fluoride, 1 μg/mL aprotinin, 1 μg/mL leupeptin, and 1 μg/mL pepstatin) on ice for 20 min. After centrifugation,

the cell lysates and cultured supernatants were subjected to SDS–PAGE and then transferred onto polyvinylidene difluoride membranes (IPVH 00010, Millipore). The membranes were analyzed by immunoblotting with the indicated antibodies and developed with Super Signal West Dura Extended Duration Substrate (34076, Thermo Scientific). The signals were analyzed with Amersham Imager 600 (GE Healthcare Life Sciences).

To prepare a positive control of mouse FGF18, we generated recombinant mouse FGF18 in *E. coli* as GST-fusion protein as described previously[39]. Briefly, *E. coli* BL21(DE3) was transformed with pGEX-6P-mFGF18. The transformed bacteria were incubated in 2 ml of LB medium containing 50 μg/ml ampicillin at 37°C overnight, followed by inoculation in 250 ml of LB medium, and the culture was continued at 37°C. When the bacteria reached 1.0 OD600, IPTG was added to a final concentration of 0.1 mM and incubated at 20 °C for 8 h to induce protein expression. Then, the bacteria were harvested and resuspended in NETN (20 mM Tris pH 8.0, 100 mM NaCl, 0.5% NP-40, 1 mM EDTA, 1 mM phenylmethylsulfonyl fluoride, 1 μg/ml aprotinin, 1 μg/ml leupeptin, and 1 μg/ml pepstatin), followed by sonication by a Polytron (KINEMATICA). After centrifugation, the supernatants were incubated with 125 μl of glutathione–Sepharose 4B beads (GE Healthcare, 17-0756-01) (1:1 slurry) for 1 h. Then, the beads were washed with NETN 4 times, followed by incubation with PreScission protease (GE Healthcare, 27-0843-01) to release mFGF18 at 4°C overnight. The concentrations of purified proteins were determined by Bradford analysis (Bio-Rad 5000006). The purity of these proteins was evaluated by staining with Coomassie blue.

### Cell culture

Isolated hepatocytes were stimulated with FGF18 (100 ng/mL), HGF (10 ng/mL), or insulin (10 ng/mL) for the indicated times. The phosphorylation of the indicated proteins was analyzed by Western blotting. For bile acid stimulation of primary hepatocytes, the medium was first changed to serum-free DMEM. Three hours later, primary hepatocytes were stimulated with 100 μM CA, GCA, TCA, and DCA for 2 h or with 1 ng/mL TGFβ1 for 24 h. The expression of the indicated genes was determined by qPCR.

### Isolation of HSCs from wild-type mice and in vitro stimulation

To isolate HSCs, we used wild-type female C57BL/6 mice over 24 weeks old with the Nycodenz density gradient centrifugation method[43]. Vitamin A-containing granules are increased in HSCs with aging, enabling us to obtain sufficient numbers of HSCs for qPCR and cell proliferation. Briefly, following the perfusion of livers with 0.3 mg/mL collagenase for 8 min as described above, the livers were perfused with LPB2 containing 0.6 mg/mL pronase E (KA-002, KNF) and 0.06 mg/ml DNase I at a flow rate of 3 ml/min for 8 min. The perfused livers were transferred to a glass dish and dispersed with the help of tweezers in LPB2 containing 0.6 mg/ml of both pronase and collagenase. The cell suspension was transferred to a beaker and incubated at 37 °C for 30 min with careful stirring. Cells were passed through a 70 μm cell strainer. After centrifugation at 600 × *g* for 10 min, the pellet was resuspended in GBSS/B (136 mM NaCl, 5 mM KCl, 1 mM MgCl$_2$ 6H$_2$O, 0.28 mM MgSO$_4$ 7H$_2$O, 0.22 mM KH$_2$PO$_4$, 0.42 mM Na$_2$HPO$_4$ 12H$_2$O, 5.5 mM glucose, 2.7 mM NaHCO$_3$, 1.5 mM CaCl$_2$ 2H$_2$O) containing 0.06 mg/ml DNase I and centrifuged at 600 × *g* for 10 min two times. The pellets were resuspended in GBSS/B containing 8.8% Nycodenz (18003, SEW), and GBSS/B was gently added to the cell suspension. After centrifugation at 1500 × *g* for 22 min without braking, the HSCs were found as a condensed white band in the interphase of both solutions. The HSC fractions were collected with a Pasteur pipette and centrifuged at 600 × *g* for 10 min. We usually obtained 1.0 × 10$^6$ cells from two mice.

To evaluate purity of HSCs isolated from the livers, cells were plated and cultured on poly-L-Lysine-coated 12 mm micro coverslips

(C012001, MATSUNAMI) in 24-well plates for 24 h. After removing nonadherent cells, cells were fixed with 4% paraformaldehyde, permeabilized with 0.1% Triton X-100, and stained with anti-desmin antibody, followed by visualization of Alexa 647-conjugated donkey anti-rabbit antibody. UV-positive or desmin-positive cells were counted manually and expressed as the percentages of total cells. Approximately 80-90% of cells were UV-positive or desmin-positive HSCs by a confocal microscopy (LSM880) (Supplementary Fig. 13).

To test whether FGF18 and TGFβ induce morphological changes of HSCs, HSCs were plated as described above. HSCs were starved with 0.2% FBS containing DMEM for 24 h, and then left untreated or simulated with FGF18 (100 ng/mL) or TGFβ1 (1 ng/mL) for 24 h. Cells were stained following the same procedure as described above, with the exception that anti-α-SMA antibody was used as the primary antibody. Then, cells were visualized with Alexa 647-conjugated donkey anti-rabbit antibody. Pictures were obtained by an All-in-One microscope (BZ-X700, KEYENCE).

HSCs were plated onto 48-well plates ($1 \times 10^5$ cells), starved with DMEM containing 0.2% FBS for 24 h and then stimulated with TGFβ1 (1 ng/mL) or FGF18 (100 ng/mL) in the absence or presence of U0126 (10 μM) or LY294002 (10 μM) for 24 h. The expression of the indicated genes was determined by qPCR.

For the proliferation assay, HSCs were plated onto 96-well plates ($1 \times 10^4$ cells) and then stimulated with TGFβ1 (1 ng/mL) or FGF18 (100 ng/mL) in DMEM containing 0.2% FBS for the indicated times. Cell viability was determined by the WST (water-soluble 2-(4-iodophenyl)-3-(4-nitrophenyl)-5-(2,4-disulfophenyl)[2H] tetrazolium monosodium salt-1) assay using a cell counting kit (343-07623, Dojindo).

## Bulk RNA-sequencing of the whole livers

Transcriptome libraries were constructed using total RNA samples from mouse livers. Briefly, polyA RNA was isolated using Dynabeads M-270 Streptavidin (DB65305, Thermo Fisher Scientific) conjugated with biotin-labeled oligo(dT) primer and reverse transcribed using SuperScript II (18064022, Thermo Fisher Scientific). For amplification of the total cDNA, the beads containing cDNA were first subjected to PCR using KAPA Hifi HS Ready Mix according to the manufacturer's instructions (KK2601, KAPA Biosystems). The first PCR products were purified using an Agencourt AMPure XP kit (A63880, Beckman-Coulter) and were used as templates for the second PCR using KAPA Hifi HS Ready Mix. The second PCR products were purified using the Agencourt AMPure XP kit, and the transcriptome library was subjected to fragmentation/end-repair/polyA-tailing/ligation using the NEBNext Ultra II FS DNA Library Prep Kit for Illumina (E7805, New England Biolabs) according to the manufacturer's instructions. Reaction products were purified using a double-size selection of the Agencourt AMPure XP kit and amplified by PCR using NEBNext Ultra II Q5 (New England Biolabs) and unique dual indexing primers. Reaction products were purified twice using a double-size selection of the Agencourt AMPure XP kit. Final transcriptome libraries, whose lengths were approximately 300 bp, were pooled and sequenced on an Illumina NovaSeq 6000 S4 flow cell (Illumina). Sequences were mapped to the mouse reference genome GRCm39 using Bowtie2 software, and TCC-GUI (https://github.com/swsoyee/TCC-GUI) was used to normalize sequence count data and conduct differential gene expression analysis. DAVID v6.8 (https://david.ncifcrf.gov/summary.jsp) was used for the functional annotations of differentially expressed genes. Heatmaps were generated using R3.6.1 with Genefilter and Gplots libraries. RNA-seq data were deposited in NCBI as GEO accession number GSE188273.

## Enzyme-linked immunosorbent assay (ELISA)

We developed an in-house ELISA for hFGF18 by generating monoclonal antibodies against hFGF18. The generation and characterization of the in-house hFGF18 ELISA system were described previously[39]. Briefly, we used one rat anti-hFGF18 antibody (clone 12G7-9) and one rabbit anti-hFGF18 antibody (clone 19S-SE5) to develop the in-house ELISA. The plate was coated with clone 12G7-9 (30 μg/mL) in 0.5% BSA-PBS at 4 °C overnight and blocked with 1% BSA-PBS-0.05% Tween (PBS-T) for 1 h. After washing with PBS-T three times, samples were added to the plate and incubated at RT for 1 h. After washing with PBS-T three times, the plate was incubated with biotinylated 19S-SE5 antibody (1.0 μg/mL) at RT for 1 h. After washing with PBS-T three times, the plate was incubated with streptavidin poly-HRP80 conjugate (0.2 μg/mL) at RT for 1 h and developed with TMB substrates. The lower detection limits of hFGF18 and mFGF18 were approximately 10 pg/mL and 500 pg/mL, respectively.

## Single-cell RNA sequencing

We isolated nonparenchymal cells from the livers of 6-week-old non-Tg and *Fgf18* Tg mice. To enrich HSCs and fibroblasts, we depleted lineage marker-positive cells using APC-conjugated antibodies that react with the following lineage markers (CD31, CD45.2, CD146, EpCAM, and Ter119), followed by anti-APC microbeads and LS columns (130-042-401, Miltenyi). Following depletion, approximately $5 \times 10^4$ cells were stained with different DNA barcode-conjugated MHC class 1 antibodies (BD™ Mouse Immune Single-Cell Multiplexing Kit [MHC H2 Class I], 626545, BD Biosciences) to identify each mouse. Then, lineage-negative cells were sorted by a BD FACSAria™ III system (BD Biosciences). A mixture of $2.4 \times 10^4$ cells from 3 non-Tg and 3 *Fgf18* Tg mice (approximately $4 \times 10^3$ cells from each mouse) was subjected to scRNA-seq using the BD Rhapsody™ Single-Cell Analysis System (BD Biosciences), and the resulting cDNA was amplified by the TAS-Seq protocol as previously described[60]. Briefly, on-bead cDNA was polyC tailed with stochastic termination conditions by terminal transferase, deoxycytidine, and spiked-in dideoxycytidine. Then, second-strand synthesis and whole-transcriptome amplification were performed by PCR. The size distribution of cDNA and BD Sampletag libraries was analyzed by a MultiNA system (MCE-202, Shimazu). The resulting cDNA library was processed to generate the sequencing library using the NEBNext UltraII FS library prep kit for Illumina (New England Biolabs), and sequencing adapters were added to associated BD Sampletag libraries by PCR. Sequencing was performed by an Illumina NovaSeq 6000 sequencer (Illumina, San Diego, CA, USA) and NovaSeq 6000 S4 Reagent Kit v1.5 (200 cycles). The pooled library concentration was adjusted to 2.0 nM, and 12% PhiX control library v3 (Illumina) was spiked into the library. The sequencing configurations were as follows: read1 67 base-pair [bp], read2 151 bp, index1 8 bp, and index2 8 bp. Adapter trimming, quality filtering, and mapping to the cell barcode and GRCm38-101 reference transcriptome or BD Sampletag reference of fastq files were performed using a pipeline (https://github.com/s-shichino1989/TASSeq) as described previously[60]. Briefly, after the trimming of adapter sequences, cell barcode reads were annotated by Python script provided by BD Biosciences with minor modification. cDNA reads were mapped to reference RNA sequences (build GRCm38 release-101) using bowtie2-2.4.2. Demultiplexing of scRNA-seq data by BD sample tag was performed using a pipeline (https://github.com/s-shichino1989/TASSeq) as described previously[60]. Briefly, cell barcode information of each read was added to the bowtie2-mapped BAM files by the python script and pysam 0.15.4 (https://github.com/pysam-developers/pysam), and read counts of each gene in each cell barcode were counted using mawk. The resulting gene expression count matrix was processed for downstream single-cell analyses (integration of six datasets, UMAP dimension reduction, cell cluster identification, and conserved marker identification) using Seurat version 4.1.1 in R version 4.1.3. Briefly, six scRNA-seq datasets were integrated with the merge functions of Seurat. Cells that contained more than 25% mitochondrial transcripts were filtered out. PCA was performed against 6885 highly-variable genes identified by the FindVariableFeatures (selection.method = mvp, mean.cutoff = c(0.1, Inf), dispersion.cutoff = c(0.5, Inf)) function in Seurat. A total of 1:56 PCs were selected by Jackstraw

analysis and used for clustering analysis. We set the resolution to 1.5. Moreover, a cell expressing two different lineage marker genes was considered a doublet cell and removed from the datasets. Next, marker genes in each cell cluster were defined by the FindAllMarkers function in Seurat (test method = Wilcox, minimum expression in each cluster ≥ 10%). In the subclustering analysis of clusters 1, 3, 8, 12, and 16, we set the resolution to 0.4.

We used CellChat to infer intercellular communication from scRNA-seq data from non-Tg and *Fgf18* Tg mice. We then downsampled cells from *Fgf18* Tg mice to those from non-Tg mice by the SubsetData function of Seurat v4.1.1. We set the resolution to 1.4. The communication probability between cell subsets was analyzed by CellChat (https://github.com/sqjin/CellChat).

We extracted the datasets of LSEC and cholangiocyte clusters from normal mouse liver scRNA-seq data[11] by the SubsetData function of Seurat. Then we integrated these datasets with those of non-Tg and *Fgf18* Tg mice. A total of 1:65 PCs were selected by Jackstraw analysis and used for clustering analysis. We set the resolution to 1.0.

### Bile acid analysis
Twenty microliters of the sera from mice of the indicated genotype fed formal diet or CDE diet for 4 weeks were used to analyze bile acids. One hundred eighty microliters of acetonitrile were added to the sera with 2 µl of 1 mM 2,4-dichlorophenoxyacetic acids as an internal standard. After vortexing for 3 min, the mixtures were centrifuged at $20,400 \times g$ for 3 min. The supernatants were loaded on an ultrafiltration membrane (Nanosep 3 K omega, PALL Corp.) and centrifuged ($14,000 \times g$, 30 min). The filtrate was dried and resuspended in 20 µl of MeOH. The LC–MS conditions were as follows: HPLC, Agilent 1260 infinity series (Agilent Technologies, Inc.), column; Ascentis Express C18 (4.6 ×150 mm, 2.7 µm) (Supelco, Inc.); solvent, (A) 0.1% formic acid aq., (B) 0.1% formic acid in acetonitrile; gradient; 30−75% (B) for 0−30 min, 95% (B) for 30−35 min; flow rate, 0.6 ml/min; column temperature, 40 °C; injection, 10 µl; MS, Agilent 6120 Quadrupole (Agilent Technologies, Inc.); mode, ESI negative, SIM; capillary voltage, 4 kV; dry gas rate, 13.0 l/min (N2); gas temperature, 350 °C; mass range, 150−600 *m/z*.

### Human liver biopsy samples
Human liver biopsy samples were obtained from patients admitted to Juntendo University Hospital to diagnose liver diseases. Half of the biopsy samples were used for histological analysis to determine the diagnosis. RNA was extracted from the remaining liver biopsy samples using an RNeasy Micro Kit (74004, QIAGEN) and analyzed by qPCR. The patients' diagnosis, age, sex, and concentrations of liver enzymes are listed in Supplementary Table 3. Written informed consent was obtained from the patients before liver biopsy. This study was approved by the ethics committee of Faculty of Medicine and Graduate School of Medicine, Juntendo University (approval number E22-0085-H01).

### Statistical analysis
Statistical analysis was performed by two-tailed unpaired Student's *t* test, two-way ANOVA with Sidak's multiple comparison test, two-way ANOVA with Tukey's multiple comparison test, one-way ANOVA with Dunnett's multiple comparison test, one-way ANOVA with Tukey's multiple comparison test, two-sided nonparametric Wilcoxon rank sum test, or Pearson correlation coefficient using GraphPad Prism 9. GO enrichment analyses were performed using DAVID 6.8, and *p* values were calculated based on one-sided Fisher's exact test. *P*-value < 0.05 was considered significant.

### Reporting summary
Further information on research design is available in the Nature Portfolio Reporting Summary linked to this article.

### Data availability
All biological materials, including the *Rosa26-LSL-Fgf18* Tg mice used in this study, are available from the corresponding authors upon request. Obtaining *Cflar^{FF}* and *Fgf18^{FF}* mice requires a material transfer agreement (MTA) with the organization described in the manuscript. The bulk RNA-seq and scRNA-seq datasets were deposited to NCBI under accession numbers GSE188273 and GSE205871, respectively. The authors declare that the data supporting the findings of this study are available within the paper and its supplementary data and supplementary information files. Source data are provided with this paper.

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

## Acknowledgements

The authors thank H. Oshima and M. Oshima for technical advice, Y.-W. He for providing *Cflar^{FF}* mice, and M. Tanaka, C.J. Okumura, and T. Maruyama for generating anti-human FGF18 antibodies. This work was supported in part by Grants-in-Aid for Scientific Research (B) (20H03475 and 23H02707 to Hiroyasu N), Scientific Research (C) (17K08994 and 20K11589 to YT), and Transformative Research Areas (B) (22H05064 to SS) from Japan Society for the Promotion of Science (JSPS); the Japan Agency for Medical Research and Development through AMED-CREST

(22gm1210002 to MT and Hiroyasu N) and AMED-PRIME (21gm6210025 to SS) from the Ministry of Education, Culture, Sports, Science, and Technology, Japan; Toho University Grant for Research Initiative Program (TUGRIP) (to Hiroyasu N); the Science Research Promotion Fund from the Promotion and Mutual Aid Corporation for Private Schools of Japan (to Hiroyasu N and SY); and the Princess Takamatsu Cancer Research Fund (to Hiroyasu N).

## Author contributions

Y.T., M.T., and Hiroyasu N. designed the study and interpreted the results. Y.T., T.S., K.K., S.K-S., T.N., and S.T. performed and analyzed most experiments. T.S., S.S., S.U., and K.M. analyzed the scRNA-seq results. A.O. performed bile acid analysis. H.Y., K.O., M.I., K.A., T.I., and M.O. provided critical reagents. Y.T. constructed the plasmids. K.F. and K.I. collected the patients' characteristics and supplied human liver biopsy samples. Hidenari N., Y.I., S.Y., C.N., T.M., T.K., and A.M. supervised the experiments. Y.T. and Hiroyasu N. wrote the manuscript with constructive input from all authors.

## Competing interests

Y.T. and Hiroyasu N. are inventors on the patent application that includes the ELISA to detect human and murine FGF18 and the generation of transgenic mice that spontaneously developed liver fibrosis used in this study. The other authors declare no conflicts of interest.

## Additional information

[1]Department of Biochemistry, Faculty of Medicine, Toho University, 5-21-16 Omori-Nishi, Ota-ku, Tokyo 143-8540, Japan. [2]Department of Biochemistry, Faculty of Pharmaceutical Sciences, Toho University, 2-2-1 Miyama, Funabashi-shi, Chiba 274-8510, Japan. [3]Department of Biological Science and Technology, Faculty of Advanced Engineering, Tokyo University of Science, 6-3-1 Niijuku, Katsushika-ku, Tokyo 125-8585, Japan. [4]Division of Molecular Regulation of Inflammatory and Immune Diseases, Research Institute for Biomedical Sciences, Tokyo University of Science, 2669 Yamazaki, Noda-shi, Chiba 278-0022, Japan. [5]Department of Gastroenterology, Faculty of Medicine and Graduate School of Medicine, Juntendo University, 2-1-1 Hongo, Bunkyo-Ku, Tokyo 113-8421, Japan. [6]Department of Gastroenterology, Toho University Omori Medical Center, 6-11-1 Omori-Nishi, Ota-ku, Tokyo 143-8541, Japan. [7]Laboratory of Quality Analysis and Assessment, Graduate School of Agriculture, Kyoto University, Gokasyo, Uji-shi, Kyoto 611-0011, Japan. [8]Department of Regenerative Medicine, Research Institute, National Center for Global Health and Medicine, 1-21-1 Toyama, Shinjuku-ku, Tokyo, 162-8655 Tokyo, Japan. [9]Laboratory of Stem Cell Regulation, Institute for Quantitative Biosciences, The University of Tokyo, 1-1-1 Yayoi, Bunkyo-ku, Tokyo, 113-0032 Tokyo, Japan. [10]Department of Pathology, Faculty of Medicine, Toho University, 5-21-16 Omori-Nishi, Ota-ku, Tokyo 143-8540, Japan. [11]Department of Immunology, Faculty of Medicine and Graduate School of Medicine, Juntendo University, 2-1-1 Hongo, Bunkyo-Ku, Tokyo 113-8421, Japan. [12]Atopy Research Center, Faculty of Medicine and Graduate School of Medicine, Juntendo University, 2-1-1 Hongo, Bunkyo-ku, Tokyo 113-8421, Japan. [13]Laboratory of Cell Growth and Differentiation, Institute for Quantitative Biosciences, The University of Tokyo, 1-1-1 Yayoi, Bunkyo-ku, Tokyo 113-0032, Japan. [14]Department of Genetics, Hyogo Medical University, 1-1 Mukogawa-cho, Nishinomiya-shi, Hyogo 663-8501, Japan. [15]Center for Animal Resources and Development, Kumamoto University, 2-2-1 Honjo, Chuo-ku, Kumamoto 860-0811, Japan. [16]Center for Metabolic Regulation of Healthy Aging, Kumamoto University, 1-1-1 Honjo, Chuo-ku, Kumamoto 860-8556, Japan. [17]Hoshi University School of Pharmacy and Pharmaceutical Sciences, 2-4-41 Ebara, Shinagawa-ku, Tokyo 142-8501, Japan. [18]National Institute of Advanced Industrial Science and Technology (AIST), 1-1-1 Umezono, Tsukuba-shi, Ibaraki 305-8560, Japan. [19]These authors contributed equally: Yuichi Tsuchiya, Takao Seki. ✉e-mail: hiroyasu.nakano@med.toho-u.ac.jp

