## [Peer Review File · Nature Communications]

Fibroblast growth factor 18 stimulates the proliferation of hepatic stellate cells, thereby inducing liver fibrosisREVIEWER COMMENTS

Reviewer #1 (Remarks to the Author):

The manuscript NCOMMS-22-25963-T reflects a large body of work demonstrating that FGF18 can activate hepatic stellate cells and then cause liver fibrosis. This is a large set of data as the authors validated their results in several independent in vivo and in vitro models, using several protein knockout/inhibition or additive strategies. The results displayed good reproducibility, a major strength of this work. The concepts and data also do raise some questions, as below.

1. Several FGFs (i.e FGF1, FGF4, FGF21) have been shown to significantly alleviate NAFLD/NASH and also show some improvement on liver fibrosis. More recently, Tong g et al. proved that FGF18 could attenuate liver fibrosis. These evidences are contradictory to the author's findings, and also affect the novelty of the paper. It is suggested that author should provide more discussion and explanation.
2. In the manuscript, different NASH models will cause the upregulation of Fgf18, and this upregulation plays an important role in further inducing liver fibrosis. However, do you know how hepatic FGF18 expression is regulated by Cflar? In addition, there are also deficiencies in the mechanism of how FGF18 promotes the upregulation of TGF β expression and causes liver fibrosis. At least, it is necessary to identify which receptor, since FGFR4 is predominantly expressed in the hepatocyte.
3. The cleaved caspase 3 and TUNEL staining were both increased in the livers of CflarLKO mice upon normal diet, however, only the cleaved caspase 3 was increased in the livers of CflarLKO mice compared to CflarFF mice fed the CDE diet? Could you confirm this cell death in these experiments using downstream markers of liver injury like caspase 3 activation or PARP cleavage by Western blotting?
4. Do there exist differences in the regulatory mechanism of liver fibrosis by normal diet- and CDE diet in CflarLKO mice?

5. In Fig. 2F, we did not find Fgf18 mRNA+ dots with co-localization in Lrat+ HSCs or HNF4a+ hepatocytes. To identify the cellular source of increased liver FGF18, you must get better resolution/rendering of images by immunofluorescence staining.

6. The main bias of this study consist of lacking a ClfarF/F;Fgf18LKO group in Fig.3. Such a group would be important to demonstrate whether FGF18 directly attenuates or exacerbates CDE-induced chronic liver injury.

7. The inflammatory microenvironment triggers the activation of fibrosis-driving hepatic stellate cells. On the other hand, the CDE significantly induced cytokine secretion (such as TNF- α and IL6) and increased the expression of inflammatory genes. It is not so clear to me whether FGF18 deficiency or overexpression would have effects on inflammation.

8. In Fig.2d, Ki67+ areas were increased in the Fgf18-injected livers compared to other group. Fibroblast growth factor (FGF) family is known to have a prominent role in the mitogenic signal transduction and subsequent signal processing. It is not so clear to me whether FGF18 directly activates the proliferation potential of hepatic stellate cells that lead to liver fibrosis. If it is through such a feasible path, the scientific significance of this manuscript will be greatly affected.

Fibroblast growth factor 18 activates hepatic stellate cells that lead to liver fibrosis

- What are the noteworthy results?

In their report, Tsuchiya and colleagues use a hepatocyte-specific Cflar knock-out model to show increased liver (hepatocyte) damage leading to several aspects of chronic liver disease (CLD) including ductular reaction and fibrosis. Upon usage of the CDE diet, these knock-out mice develop a more severe phenotype than their wild-type counterparts. Upon more deep investigation of these models, Fgf18 is identified as a potent (but partial) mediator of the Cflar knock-out phenotype. Hepatocyte-specific knock-out of Fgf18 in hepatocyte-specific Cflar knock-out mice alleviates the phenotype and hepatocyte-specific Fgf18 overexpression mimics the hepatocyte-specific Cflar knock-out mice phenotype. Moreover, hepatocyte-specific Fgf18 overexpression shows clear signs of spontaneous development of CLD: inflammation, ductular reaction, fibrosis, and angiogenesis, bypassing the need for hepatocyte damage (ie ALT levels normal). Next, the Fgf18 function is linked to hepatic stellate cell activation (HSC) activation including insightful single cell RNA Seq analysis.

- Will the work be of significance to the field and related fields? How does it compare to the established literature? If the work is not original, please provide relevant references.

As mentioned by the authors, the findings presented here contrast that of a recent publication by Tong et al. However, the experimental setup is different: overexpression of Fgf18 in this manuscript is genetic whereas Tong et al use a vector-based method. Fgf18 knock-out in this manuscript consists of (combined Cflar-)Fgf18 hepatocyte-specific knock-out whereas Tong et al use an HSC-specific knock-out. Additionally, experimental models differ between manuscripts. More experiments in the future might elucidate these discrepancies.

If anything, both manuscripts underline a role for Fgf18 in CLD which is novel and thus provides a significant contribution to the field.

- Does the work support the conclusions and claims, or is additional evidence needed?

Despite insightful research, this reviewer views several claims as too strong or not backed-up by the provided evidence.

- Are there any flaws in the data analysis, interpretation, and conclusions? - Do these prohibit publication or require revision?

Several flaws concerning the experimental setup can be identified. Some data interpretation and conclusions are not backed up by thorough evidence.

At this stage, this manuscript cannot be published and requires revision.

- Is the methodology sound? Does the work meet the expected standards in your field?

Methodology suffers from less conventional CLD models (ie CDE) when compared to more conventional models (CCl₄, bile duct ligation, or metabolic MAFLD) for CLD. The reviewer highly doubts the quality of HSC culture methods. The HSC culture model does not meet the current standard of the field. At least, thorough data should be provided to show the culture quality of these cells.

The power of the genetic mouse models must be underlined here. They are very strong models to investigate the role of Fgf18 in the liver. However, there is a lack of evidence of successful overexpression or knock-out of indicated genes.

- Is there enough detail provided in the methods for the work to be reproduced?

Yes. Minor remarks can be made for the methodology section, but overall, this is sufficient.

Sex and Gender Equity in Research – SAGER – guidelines

DDC: male mice, other: female mice. The method section might further elaborate on the choice of sex.

Specific comments

- ***Language***

A revision of language should be done throughout the manuscript, including, but not limited to, lines: 54, 77, 102, 104, 137, 314, 533, 678, 719, 953

- ***Title***

The title does not properly represent the presented research and should be altered

- ***NAFLD – MAFLD***

Since 2020 the nomenclature of NAFLD has changed to MAFLD. Since this manuscript draws conclusions for MAFLD pathogenesis, a change in terminology should be addressed. Additionally, next to liver steatosis, evidence for the presence of one of the following must be provided for MAFLD mouse models: significant weight gain, insulin resistance, or any other metabolic abnormality related to metabolic syndrome. Since CDE diet and genetic (non-injured) models are the main models in this manuscript, either evidence for meeting the definition of MAFLD in these models must be shown, or conclusions for MAFLD must be avoided throughout the manuscript.

- ***Abstract***

MAFLD comment: cf supra

Line 60: “FGF18 induces expression of TGFB” address in what cell type this happens

- ***Introduction***

MAFLD comment: cf supra

With a focus on HSCs, the introduction should at least mention their role in chronic liver disease

Line 74: please provide a reference to “many clinical trials”

- ***Results – Line 119 – 148 + Figure 1***

Conclusion: Hepatocyte-specific Cflar knock-out mice spontaneously develop characteristics of CLD. When exposed to the CDE diet, they develop a more severe phenotype.

Experimental setup should be altered: Age-matched mice (both FF and KO) not receiving CDE diet (ie extended data Fig1) should be analyzed together with CDE diet mice (both FF and KO - figure 1); resulting in 4 conditions to be analyzed together.

Data for all conditions must be provided: Gene expression of collagens and Acta2 is lacking in CDE-treated mice. Body weight is lacking in non-injured mice.

The successful knock-out of Cflar has to be shown, both protein and gene expression for all 4 conditions. Since Cre-Alb is used, this has to be shown for hepatocytes compared to other liver cells.

- **Results – Line 149 – 177 + Figure 2**

Conclusion: RNA Seq analysis and overexpression techniques are purposefully used to reveal Fgf18 as a potential mediator of the Cflar knock-out phenotype.

HTVi overexpression: (1) please provide evidence for successful overexpression both for protein and gene expression. (2) Staining of the eGFP control should be shown. (3) Since the albumin promoter – AFP enhancer is used, this has to be shown for hepatocytes compared to other liver cells.

The Cflar knock-out mice express increased levels of Fgf18. However, it seems this increase is not related to the increase of Fgf18 in hepatocytes, but rather in stellate cells. This would implicate that HSCs seem to be the mediator of the Fgf18 phenotype. Later, hepatocyte-specific models (overexpression and knock-out) are used. Please argue this choice.

If mouse MAFLD conclusions are to be made: perhaps the analysis of GSE99010 might be insightful.

- **Results Line 178 – 199 + Figure 3**

Conclusion: Here, evidence shows that hepatocyte-specific knock-out of Fgf18 partially rescues the fibrogenic (but not ductular reaction) phenotype of the Cflar knock-out. This confirms Fgf18 as a partial mediator of the Cflar knock-out phenotype.

The successful knock-out of Fgf18 & Cflar has to be shown, both protein and gene expression for all 4 conditions. Since Cre-Alb is used, this has to be shown for hepatocytes compared to other liver cells.

4 conditions are shown: Cflar^{FF}, Cflar^{KO}, Cflar^{FF}Fgf18^{FF}, Cflar^{KO}Fgf18^{KO}. Please elaborate on the difference between Cflar^{FF} and Cflar^{FF}Fgf18^{FF}. (1) why does this condition have lower Fgf18 levels? (2) wouldn't a Cflar^{FF}Fgf18^{KO} condition be a better control?

- **Results Line 200 – 225 + Figure 4**

Conclusion: Fgf18 overexpression bypasses the need for hepatocyte damage to induce characteristics of CLD: fibrosis, inflammation, and angiogenesis indicating its role in CLD.

The successful overexpression of Fgf18 has to be shown, both protein and gene expression for all conditions. Since Cre-Alb is used, this has to be shown for hepatocytes compared to other liver cells.

Agree with the statement that Fgf18 is not essential for CK19+ cell expansion (but overexpression stimulates ductular reaction)

- **Results Line 226 – 258 + Figure 5**

Conclusion: The RNA Sequencing data nicely shows an overlap of Fgf18 overexpression and the Cflar knock-out further linking the fibrosis phenotype of Cflar knock-out to Fgf18.

A deeper analysis of CD31-CD34+ cells identifies HSCs as one of the affected cells by Fgf18 overexpression.

- **Results Line 259 – 306 + Figure 6**

We have some remarks concerning the experimental setup and methodology:

(1) The main goal of this experiment is to identify intercellular crosstalk. Since the hepatocytes overexpress the Fgf18, why was it chosen to occlude them from single-cell RNA seq analysis? (only non-parenchymal cells were chosen for single cell analysis).

(2) despite the selection of non-parenchymal cells, the single RNA seq results still show plenty of hepatocytes. How can this be explained?

The definition of cell clusters is elaborate and justified.

CellChat analysis nicely shows the impact of Fgf18 and provides evidence for direct hepatocyte-HSC crosstalk.

Line 301: Fig2 refers to the Cflar^{KO} mice, which does not seem relevant to the statement made in this sentence.

- **Results Line 307 – 328 + Figure 7**

Concerning the HSC culture, remarks are to be made. (1) If the method section is understood correctly, lineage marker-negative NPCs, the same cells that were used for single-cell RNA sequencing in figure 6, were plated for 3 days. Thus, this cell culture consists of multiple cell types including hepatocytes (that overexpress Fgf18), stellate cells, myofibroblast, portal fibroblasts, and several immune cells as shown in the single RNA sequencing data. This culture is thus a co-culture of multiple days (even if cells don't attach) which can highly influence HSC activation status. (2) After three days, cells are split and replated for an unclear amount of time (3) It is unclear why HSCs were isolated from Fgf18 Tg mice which, as shown by RNA seq data, influences their activation status. (4) why were these cells culture with Y27632, a ROCK Inhibitor? In conclusion, these experiments are not performed on freshly isolated quiescent HSCs which makes further interpretation of results difficult and unnecessarily complicated.

Either density gradient isolation or ultraviolet sorting methods to culture pure quiescent HSCs from the start are suggested.

In any way, SMA staining and light microscopy are required to show the phenotype of the HSCs. Staining for an HSC marker (e.g. Desmin) with quantification of the amount of HSCs present within the cell culture at the start of the culture should be provided.

Concerning iPSC HSC, light microscopy, SMA staining, and lipid droplet content have to be shown to prove a quiescent to activated-like process.

Based on the results of this section, the reviewer cannot agree with the statement that Fgf18-Tgfb constitutes a feed-forward loop: (1) in mouse HSC: Fgf18 does not induce collagens, Fgf18 nor Tgfb. (2) in iPSC HSC, Tgfb does not induce Fgf18.

- **Discussion**

Line 330: Cf MAFLD comment above

Line 336: Cf Feedforward loop above

Line 368: the way the sentence is written alludes to the fact that HSCs differentiate towards either myofibroblasts or portal fibroblasts. Generally, portal fibroblasts are considered distinct from HSC with their fibrogenic properties.

Line 377: single-cell data in this manuscript clearly shows Hgf expression in HSCs, but not in portal fibroblasts or myofibroblasts (Fig6c). “may produce” should be rephrased. Additionally, this could easily be tested in vitro with correct HSC cultures as explained above

- **Materials & Methods**

Line 441: If Cflar flox/flox mice were provided, please provide text on how hepatocyte-specific Cflar KO mice were generated.

Line 441: Reference should be provided for the RIKEN database and initial publication of the Fgf18^{FF} mice.

A statement on ethical animal use is missing in this section.

Line 536: The reviewer cannot find these qPCR results

- **MAFLD pathogenesis**

If MAFLD pathogenic conclusions are to be made, the reviewer suggests several further analyses: (1) RNA Scope for Fgf18 in healthy vs MAFLD patient livers, (2) analysis of single-cell RNA Seq of human livers (MAFLD vs healthy – eg GSE136103): what cells express Fgf18, perhaps Cellchat for Fgf18?

Since an Fgf18 hepatocyte overexpression model is used, it should be shown that in chronic liver disease, hepatocytes have a higher expression of Fgf18.

Overall, a setup of hepatocyte-specific Fgf18 knock-out vs hepatocyte-specific Fgf18 overexpression vs wild-type mice subjected to a metabolic MAFLD model (e.g. high-fat diet + low dose CCl4 – Tsuchida T et al journal of hepatology 2018 <http://dx.doi.org/10.1016/j.jhep.2018.03.011>) vs standard diet (ie 6 conditions) seems the most obvious choice for drawing conclusions on MAFLD pathogenesis.

Reviewer #3 (Remarks to the Author):

The manuscript by Tsuchiya and colleagues presents an extensive set of data that demonstrates the role of FGF 18 in the stimulation of hepatic fibrosis. The study is well written and performed. However, a few areas of concern exist.

1. The rationale for utilizing the CflarLKO mice is not clearly stated in the manuscript. Also, the choice of CDE hepatotoxic diet rather than a traditional HFD (western diet). This model seems to have an excessive ductular reaction that is not present in NASH while the amount of fibrosis is not that elevated.
2. Figure 2d,e the Ki67 increase in the animals treated with Fgf18 is minimal. This should be discussed. Co-staining should be performed to eliminate Ki67 expression in cholangiocytes.
3. Wild-type fed control diet and CDE should be included at least initially for comparison to observed features in the CflarLKO mic.
4. Only ALT levels are measured. AST and alkaline phosphatase levels should also be included.
5. Rather than use HSCs from iPS cells, the authors should treat primary HSCs or isolated HSCs with FGF18.
6. Staining for HSCs should be performed in all of the animal treatment groups.
7. Interactions between HSCs and hepatocytes stimulated with TGFbeta could be evaluated in co-culture experiments. In addition, Hepatocytes should be stimulated directly with FGF18 and evaluated for factors that are profibrotic.

**REVIEWER COMMENTS**

**Reviewer #1 (Remarks to the Author):**

**The manuscript NCOMMS-22-25963-T reflects a large body of work**
**demonstrating that FGF18 can activate hepatic stellate cells and then**
**cause liver fibrosis. This is a large set of data as the authors validated**
**their results in several independent in vivo and in vitro models, using**
**several protein knockout/inhibition or additive strategies. The results**
**displayed good reproducibility, a major strength of this work. The**
**concepts and data also do raise some questions, as below.**

**1. Several FGFs (i.e FGF1, FGF4, FGF21) have been shown to significantly**
**alleviate NAFLD/NASH and also show some improvement on liver fibrosis.**
**More recently, Tong g et al. proved that FGF18 could attenuate liver**
**fibrosis. These evidences are contradictory to the author's findings, and**
**also affect the novelty of the paper. It is suggested that author should**
**provide more discussion and explanation.**

**RESPONSE:** Thank you for identifying the critical points. Indeed, several members of
the FGF family attenuate liver fibrosis. Although our previous experiments using iPSC-
derived HSC-like cells showed upregulation of fibrotic genes, FGF18 did not upregulate
the expression of profibrotic genes, but suppressed the TGF β -induced upregulation of
profibrotic genes in primary HSCs. These results are consistent with a previous study by
Tong et al. ¹. However, as we mentioned in the initial manuscript, transgenic expression
of *Fgf18* in hepatocytes resulted in liver fibrosis. Moreover, we found that FGF18
induced the proliferation of HSCs along with the upregulation of *Ccnd1*, encoding
cyclin D1, which is involved in cell cycle progression.

How do we reconcile these two inconsistent results? Importantly, the
upregulation of fibrotic genes, including *collagens*, *Acta2*, and *Tgfb3*, was observed
during the early stages following treatment with fibrosis-inducing stimuli. However, the
induction of the *Fgf18* gene occurred at relatively later stages in CCl₄-treated mice fed
the normal diet or a Western diet. Therefore, it is reasonable to speculate that brief
exposure to profibrotic agents leads to the production of TGF β , which activates HSCs
and results in mild fibrosis, but FGF18 is not involved in this process. Prolonged
exposure to profibrotic agents leads to the upregulation of FGF18, which stimulates the

proliferation of quiescent HSCs. Proliferating HSCs eventually undergo full activation
in response to profibrotic stimuli and further exacerbate fibrosis. Hence, the timing and
duration of *Fgf18* expression (during embryonic development versus in adult mice;
short exposure versus sustained exposure) or the methods used for *Fgf18* deletion
(germinal deletion versus shRNA-mediated deletion in adult mice) are likely crucial
factors that determine whether FGF18 could alleviate or worsen liver fibrosis. Further
study will be required to address this issue.

The results from iPSC-derived HSC-like cells were preliminary and may be
affected depending on the maturation stages of iPSC-derived HSC-like cells. Thus, to
keep our results consistent, we have deleted the results in the revised manuscript.

We have included the results using primary HSCs from wild-type mice in Fig.
7 and described these findings in the Results and Discussion sections (Lines 379-407;
478-497). Moreover, we have mentioned the role of other members of the FGF family
in liver fibrosis in the Introduction section (Lines 100-109).

**2. In the manuscript, different NASH models will cause the upregulation of**
**Fgf18, and this upregulation plays an important role in further inducing**
**liver fibrosis. However, do you know how hepatic FGF18 expression is**
**regulated by Cflar? In addition, there are also deficiencies in the**
**mechanism of how FGF18 promotes the upregulation of TGF β expression**
**and causes liver fibrosis. At least, it is necessary to identify which**
**receptor, since FGFR4 is predominantly expressed in the hepatocyte.**

RESPONSE: Thank you for your thoughtful comments. Regarding the relationship
between *Fgf18* and *Cflar*, we found that the expression of *Fgf18* was elevated in the
livers of 8-week-old *Cflar*^{LKO} mice compared to *Cflar*^{FF} mice, even those fed a normal
diet. Of note, *Tgfb*s was elevated in the livers of *Cflar*^{LKO} mice compared to *Cflar*^{FF}
mice (Fig. 2b), and TGF β induced the elevation of FGF18 in HSCs and hepatocytes
(Fig. 7f, g). We hypothesize that *Cflar* deficiency in hepatocytes induces hepatocyte
apoptosis that triggers macrophage-dependent TGF β production through efferocytosis
of apoptotic hepatocytes, thereby upregulating *Fgf18*. We have included the results in
Fig. 3b and mentioned them in the Results and Discussion section (Lines 234-241, 432-
437).

Since FGF18-induced *Tgfb*s elevation was observed only in hiPSC-derived
HSCs, but not primary HSCs, we have deleted those results.

Regarding the contribution of FGF18 to liver fibrosis, we assume that FGF18
derived from hepatocytes and HSCs may stimulate the proliferation of HSCs, but not
the upregulation of profibrotic genes in HSCs (please see the response to comment 1).

Regarding the signaling pathways that induce the proliferation of HSCs, we
first examined the relative expression of *Fgfr1-Fgfr4* in HSCs and hepatocytes by
qPCR. HSCs predominantly expressed *Fgfr1* and *Fgfr2*, but not *Fgfr3* or *Fgfr4* (Fig.
7e). Moreover, a MEK inhibitor, U0126, almost completely blocked the upregulation of
*Ccnd1*. In contrast, an Akt inhibitor moderately suppressed it (Fig. 7d). These results
suggest that FGF18 transmits the signals in HSCs through FGFR2 in a MEK/ERK-
dependent fashion. We have included these results in the indicated figures and
mentioned them in the text (Lines 386-393).

**3. The cleaved caspase 3 and TUNEL staining were both increased in the**
**livers of CflarLKO mice upon normal diet; however, only the cleaved**
**caspase 3 was increased in the livers of CflarLKO mice compared to**
**CflarFF mice fed the CDE diet? Could you confirm this cell death in these**
**experiments using downstream markers of liver injury like caspase 3**
**activation or PARP cleavage by Western blotting?**

RESPONSE: We apologize for causing an incorrect impression. After careful
examination of the liver tissues stained with anti-cleaved caspase 3 antibody and
TUNEL, we concluded that the calculation of the area showing CC3⁺ or TUNEL⁺ was
not appropriate and gave the wrong impression that there were many apoptotic cells in
the livers of *Cflar^{FF}* and *Cflar^{LKO}* mice fed the CDE diet for 4 weeks. To circumvent
this problem, we calculated the numbers of CC3⁺ and TUNEL⁺ cells and presented the
number of CC3⁺ or TUNEL⁺ cells per field (FOV) in a new Fig. 1d and e, and
Supplementary Fig. 1e and f, 2c and d, and 5f and g.

As described above, since the numbers of CC3⁺ cells were relatively low even
in the livers of *Cflar^{LKO}* mice fed the CDE diet, we could not detect the cleaved form of
caspase 3 by Western blotting using liver extracts. Moreover, the uncleaved form of
PARP was slightly decreased in the liver lysates of *Cflar^{LKO}* mice, but we could not
detect the cleaved form of PARP. Please note that bands in the red boxes appear to be
cross-reacted bands of mouse Ig heavy (upper) and light (lower) chains reacted with
antibody against mouse IgGs. Therefore, we have only included the results for CC3 and
PARP in the liver extracts by Western blotting in the rebuttal letter.

**Rebuttal Fig. 1** | Western blotting of caspase 3 and PARP in the liver extracts of mice
 of the indicated genotypes fed the normal or CDE diet. Pro and CC3 indicate the
 proform (Pro) and cleaved form of caspase 3 (CC3), respectively. Full and cleaved
 indicate the full-length and cleaved forms of PARP, respectively. N and P indicate
 negative and positive controls for the activation of caspase 3, respectively. Red boxes
 indicate cross-reacted bands of heavy and light chains of mouse immunoglobulins that
 were detected by anti-mouse IgG antibody used to detect anti-PARP antibody (raised in
 mouse).

**4. Do there exist differences in the regulatory mechanism of liver fibrosis**
 **by normal diet- and CDE diet in *Cflar*^{LKO} mice?**

**RESPONSE:** As shown in our response to comment 2, the expression of *Fgf18* was
 already elevated in NPCs from the livers of *Cflar*^{LKO} mice fed the normal diet, but the
 expression levels were further elevated in *Cflar*^{LKO} mice fed the CDE diet (Fig. 2c and
 Fig. 3b). Thus, we assume that the regulatory mechanisms of liver fibrosis in *Cflar*^{LKO}
 mice fed the normal and CDE diets are, at least in part, mediated by FGF18. Of course,
 signals triggered by inflammatory cytokines produced by immune cells in response to

lipotoxicity other than FGF18 may also be involved in the exacerbation of liver fibrosis
in *Cflar*^{LKO} mice fed the CDE diet. We have mentioned this information in the
Discussion section (Lines 432-439).

**5. In Fig. 2F, we did not find Fgf18 mRNA+ dots with co-localization in**
**Lrat+ HSCs or HNF4a+ hepatocytes. To identify the cellular source of**
**increased liver FGF18, you must get better resolution/rendering of images**
**by immunofluorescence staining.**

RESPONSE: We apologize for the ambiguous presentation of our results. Notably,
those puncta did not appear to colocalize with each other. In this figure, we would like
to stress that *Fgf18* mRNA⁺ puncta were located in close proximity to *Hnf4a* mRNA⁺
puncta or *Lrat* mRNA⁺ puncta. To clarify the margins of hepatocytes, we indicated the
outer margins of hepatocytes with white dotted lines. However, it was relatively
difficult to identify the margins of HSCs, since many immune cells had infiltrated the
livers of *Cflar*^{LKO} mice fed the CDE diet. Given that *Fgf18* mRNA puncta were very
close to *Lrat* mRNA puncta, these results suggest that *Fgf18* mRNA was expressed in
HSCs. We replaced the old Fig. 2f with the new Fig. 2f and mentioned it in the Results
section (Lines 196-207).

**6. The main bias of this study consist of lacking a ClfarF/F;Fgf18LKO**
**group in Fig.3. Such a group would be important to demonstrate whether**
**FGF18 directly attenuates or exacerbates CDE-induced chronic liver**
**injury.**

RESPONSE: It is technically difficult to generate hepatocyte-specific *Fgf18*-deficient
mice under the *Cflar*^{FF} genetic background, because *Alb-Cre* lacks both *flox* alleles of
*Fgf18* and *Cflar*. To circumvent this problem, we have included the results using *Fgf18*^{FF}
and *Fgf18*^{LKO} mice fed the normal and CDE diet that showed that CDE diet-induced liver
fibrosis was not exacerbated or attenuated in the livers of *Fgf18*^{LKO} mice compared to
*Fgf18*^{FF} mice. To include these results, we have made a new Supplementary Fig. 5 and
described it in the Results section (lines 226-232).

**7. The inflammatory microenvironment triggers the activation of fibrosis-**
**driving hepatic stellate cells. On the other hand, the CDE significantly**

**induced cytokine secretion (such as TNF- α and IL6) and increased the**
**expression of inflammatory genes. It is not so clear to me whether FGF18**
**deficiency or overexpression would have effects on inflammation.**

RESPONSE: Thank you for noting this. As suggested, we selected several candidate
genes by bulk RNA-seq analysis of the livers that were elevated in *Fgf18* Tg mice
compared to non-Tg mice. Among the various genes examined, including *Il33*, *Tnf*,
*Tnfsf6* (*Fasl*), and *Tnfsf10* (*Trail*), we found that the expression of *Tnf* was elevated in
the livers of *Clar*^{LKO} mice and that its expression was reduced in the livers of
*Cflar*^{LKO};*Fgf18*^{LKO} mice. Thus, the expression of *Tnf* appeared to be regulated by
FGF18. We have made Supplementary Fig. 8 to include these results and mentioned
them in the Results section (Lines 286-291).

**8. In Fig.2d, Ki67+ areas were increased in the Fgf18-injected livers**
**compared to other group. Fibroblast growth factor (FGF) family is known**
**to have a prominent role in the mitogenic signal transduction and**
**subsequent signal processing. It is not so clear to me whether FGF18**
**directly activates the proliferation potential of hepatic stellate cells that**
**lead to liver fibrosis. If it is through such a feasible path, the scientific**
**significance of this manuscript will be greatly affected.**

RESPONSE: Thank you for the very thoughtful suggestion. As suggested, we isolated
primary HSCs and then stimulated them with FGF18. As shown in Fig. 7b, FGF18
induced the proliferation of HSCs on day 3 and day 5 following stimulation. We also
found that FGF18 induced the expression of *Ccnd1*, which encodes cyclin D1, a protein
that plays an important role in the transition from the G1 to G2/M phases². Given that
FGF18 did not upregulate the expression of profibrotic genes, these results suggest that
FGF18 promotes liver fibrosis by stimulating the proliferation of HSCs. We have
mentioned this in the Results (Lines 386-393).

**Reviewer #2 (Remarks to the Author):**

**Fibroblast growth factor 18 activates hepatic stellate cells that lead to liver**
**fibrosis**

**- What are the noteworthy results?**

**In their report, Tsuchiya and colleagues use a hepatocyte-specific Cflar**
**knock-out model to show increased liver (hepatocyte) damage leading to**
**several aspects of chronic liver disease (CLD) including ductular reaction**
**and fibrosis. Upon usage of the CDE diet, these knock-out mice develop a**
**more severe phenotype than their wild-type counterparts. Upon more deep**
**investigation of these models, Fgf18 is identified as a potent (but partial)**
**mediator of the Cflar knock-out phenotype. Hepatocyte specific knock-out**
**of Fgf18 in hepatocyte-specific Cflar knock-out mice alleviates the**
**phenotype and hepatocyte-specific Fgf18 overexpression mimics the**
**hepatocyte-specific Cflar knock-out mice phenotype. Moreover,**
**hepatocyte-specific Fgf18 overexpression shows clear signs of**
**spontaneous development of CLD: inflammation, ductular reaction, fibrosis,**
**and angiogenesis, bypassing the need for hepatocyte damage (ie ALT**
**levels normal). Next, the Fgf18 function is linked to hepatic stellate cell**
**activation (HSC) activation including insightful single cell RNA Seq**
**analysis.**

**- Will the work be of significance to the field and related fields?**

**How does it compare to the established literature? If the work is not original,**
**please provide relevant references. As mentioned by the authors, the**
**findings presented here contrast that of a recent publication by Tong et al.**
**However, the experimental setup is different: overexpression of Fgf18 in**
**this manuscript is genetic whereas Tong et al use a vector-based method.**
**Fgf18 knock-out in this manuscript consists of (combined Cflar-)Fgf18**
**hepatocyte-specific knock-out whereas Tong et al use an HSC-specific**
**knockout. Additionally, experimental models differ between manuscripts.**
**More experiments in the future might elucidate these discrepancies. If**
**anything, both manuscripts underline a role for Fgf18 in CLD which is novel**
**and thus provides a significant contribution to the field.**

- Does the work support the conclusions and claims, or is additional
evidence needed?

Despite insightful research, this reviewer views several claims as too
strong or not backed-up by the provided evidence.

- Are there any flaws in the data analysis, interpretation, and conclusions?

- Do these prohibit publication or require revision?

Several flaws concerning the experimental setup can be identified. Some
data interpretation and conclusions are not backed up by thorough
evidence. At this stage, this manuscript cannot be published and requires
revision.

- Is the methodology sound? Does the work meet the expected standards
in your field?

Methodology suffers from less conventional CLD models (ie CDE) when
compared to more conventional models(CCI4, bile duct ligation, or
metabolic MAFLD) for CLD. The reviewer highly doubts the quality of HSC
culture methods. The HSC culture model does not meet the current
standard of the field. At least, thorough data should be provided to show
the culture quality of these cells. The power of the genetic mouse models
must be underlined here. They are very strong models to investigate the
role of Fgf18 in the liver. However, there is a lack of evidence of successful
overexpression or knock-out of indicated genes.

- Is there enough detail provided in the methods for the work to be
reproduced?

Yes. Minor remarks can be made for the methodology section, but overall,
this is sufficient. Sex and Gender Equity in Research – SAGER – guidelines
DDC: male mice, other: female mice. The method section might further
elaborate on the choice of sex.

**Specific comments**

· **Language**

A revision of language should be done throughout the manuscript,
including, but not limited to, lines: 54, 77, 102, 104, 137, 314, 533, 678, 719,
953.

RESPONSE: Thank you for your comments on improving the manuscript. We have asked
native English speakers to edit the manuscript again.

• **Title**

**The title does not properly represent the presented research and should be**
**altered**

RESPONSE: As suggested, we changed the title to “Fibroblast growth factor 18 simulates
the proliferation of hepatic stellate cells, thereby inducing liver fibrosis”.

• **NAFLD – MAFLD**

**Since 2020 the nomenclature of NAFLD has changed to MAFLD. Since this**
**manuscript draws conclusions for MAFLD pathogenesis, a change in**
**terminology should be addressed. Additionally, next to liver steatosis,**
**evidence for the presence of one of the following must be provided for**
**MAFLD mouse models: significant weight gain, insulin resistance, or any**
**other metabolic abnormality related to metabolic syndrome. Since CDE diet**
**and genetic (non-injured) models are the main models in this manuscript,**
**either evidence for meeting the definition of MAFLD in these models must**
**be shown, or conclusions for MAFLD must be avoided throughout the**
**manuscript.**

RESPONSE: Thank you for the critical comments on our work. We agree with your
comments that a CDE model is not suitable for investigating the mechanisms underlying
the development of NASH. Thus, we have changed the title of the manuscript and focused
on chronic liver fibrosis, but not NASH. In addition, we used the term “MAFLD” in the
Introduction section (Lines 74-78).

• **Abstract**

**MAFLD comment: cf supra**

**Line 60: “FGF18 induces expression of TGFB” address in what cell type**
**this happens**

RESPONSE: Since we only observed that FGF18 induced the expression of *Tgfb* in
hiPSC-derived HSC-like cells, we have deleted the sentence.

• **Introduction**

MAFLD comment: cf supra

With a focus on HSCs, the introduction should at least mention their role in chronic liver disease

Line 74: please provide a reference to “many clinical trials”

RESPONSE: As suggested, we have discussed the role of HSCs in developing liver fibrosis in the 2nd paragraph of the Introduction (Lines 81-93). Since we did not focus on NASH, we have deleted the sentence “many clinical trials” in the revised manuscript.

• **Results – Line 119 – 148 + Figure 1**

Conclusion: Hepatocyte-specific Cflar knock-out mice spontaneously develop characteristics of CLD. When exposed to the CDE diet, they develop a more severe phenotype.

Experimental setup should be altered: Age-matched mice (both FF and KO) not receiving CDE diet (ie extended data Fig1) should be analyzed together with CDE diet mice (both FF and KO - figure 1); resulting in 4 conditions to be analyzed together. Data for all conditions must be provided: Gene expression of collagens and Acta2 is lacking in CDE-treated mice. Body weight is lacking in non-injured mice.

RESPONSE: As suggested, we included the results from age-matched *Cflar^{FF}* and *Cflar^{LKO}* mice fed the normal or CDE diet in Fig. 1. Moreover, we included the results of the expression of *collagens* and *Acta2* and the body weight of mice fed the normal diet for 4 weeks. To assess the statistical significance between two groups characterized by distinct genotypes and diets, we employed the two-way ANOVA. However, statistical significance was no longer observed between *Cflar^{FF}* and *Cflar^{LKO}* mice fed the normal diet for various parameters, such as CC3⁺ cells and the expression of profibrotic genes (see Fig. 1d, i, j, k vs. Supplementary Fig. 1e, i, j, k). Therefore, we have retained the data in the original Supplementary Fig. 1 as originally presented to demonstrate the statistically significant differences in these parameters between 8-week-old *Cflar^{FF}* and *Cflar^{LKO}* mice fed the normal diet. We have mentioned the findings in the Results section (Lines 131-174).

**The successful knock-out of Cflar has to be shown, both protein and gene**
**expression for all 4 conditions. Since Cre-Alb is used, this has to be shown**
**for hepatocytes compared to other liver cells.**

RESPONSE: As suggested, we isolated hepatocytes and nonparenchymal cells (NPCs)
from 8-week-old female *Cflar^{FF}* and *Cflar^{LKO}* mice and determined the expression levels
of the *Cflar* gene and cFLIP protein by qPCR and Western blotting, respectively. As
shown in our previous paper ³, the deletion of cFLIP in hepatocytes from *Cflar^{LKO}* mice
was partial (Supplementary Fig. 1a, b). Of note, the complete deletion of *Cflar* in
hepatocytes by crossing *Cflar^{fllox}* mice with another Cre line, *Aflp-Cre* Tg mice, results in
perinatal lethality ³. Isolating viable hepatocytes from fibrotic livers is technically
difficult; therefore, we could not obtain the results for *Cflar^{FF}* and *Cflar^{LKO}* mice fed the
CDE diet for 4 weeks. We have mentioned this in the Results section (Lines 131-136).

**· Results – Line 149 – 177 + Figure 2**

**Conclusion: RNA Seq analysis and overexpression techniques are**
**purposefully used to reveal Fgf18 as a potential mediator of the Cflar knock-**
**out phenotype.**

**HTVi overexpression:**

**(1) please provide evidence for successful overexpression both for protein**
**and gene expression.**

**(2) Staining of the eGFP control should be shown.**

RESPONSE: As suggested, we included the results of the expression of transfected genes
at the mRNA level (Supplementary Fig. 3a). We also confirmed the expression of EGFP
at the protein level by immunohistochemistry (Supplementary Fig. 3b).

Our purpose in performing HTVi experiments was to identify the candidate(s)
that potentially induce the proliferation of CK19⁺ cells or desmin⁺ HSCs. Hence, we
assumed that determining the expression of transfected genes by qPCR would be
sufficient for our purpose. Moreover, we could not detect FGF18-positive cells in the
livers by immunohistochemistry using anti-FGF18 antibody (please see the response to
(3)). We have mentioned this in the Results section (Lines 188-195).

**(3) Since the albumin promotor-AFP enhancer is used, this has to be shown**
**for hepatocytes compared to other liver cells. The Cflar knock-out mice**

**express increased levels of Fgf18. However, it seems this increase is not**
**related to the increase of Fgf18 in hepatocytes, but rather in stellate cells.**
**This would implicate that HSCs seem to be the mediator of the Fgf18**
**phenotype. Later, hepatocyte-specific models (overexpression and knock-**
**out) are used. Please argue this choice. If mouse MAFLD conclusions are**
**to be made: perhaps the analysis of GSE99010 might be insightful.**

RESPONSE: Thank you for identifying this critical issue. We first tested which
antibodies were able to detect endogenous FGF18 by Western blotting. Although Tong
et al. used anti-FGF18 antibody from Invitrogen (PA-45495) in the study ¹, we could not
detect the specific band of FGF18 in the liver lysates from *Fgf18* Tg mice (Rebuttal Fig.
2a). After various antibodies against FGF18 were investigated, we found that anti-FGF18
antibody from Cusabio (CSB-PA120525) could detect endogenous FGF18 (Rebuttal Fig.
2a, Fig. 4f). Thus, we used this antibody for subsequent study. Anti-FGF18 antibody
(CSB-PA120525) revealed that a number of hepatocytes appeared to be positive for
FGF18 in the livers of *Cflar*^{LKO} mice fed the CDE diet (Rebuttal Fig. 2b). Therefore, we
crossed *Fgf18*^{fl^{ox}} mice with *Alb-Cre* mice to generate hepatocyte-specific *Fgf18*-deficient
mice and hepatocyte-specific overexpression of *Fgf18* in mice. However, a subsequent
study revealed that anti-FGF18 antibody (CSB-PA120525) also stained hepatocytes in
the livers of CDE-treated *Fgf18*^{LKO}*Cflar*^{LKO} mice (Rebuttal Fig. 2b). These results
indicate that anti-FGF18 antibody (CSB-PA120525) is suitable for Western blotting, but
not for immunohistochemistry. To circumvent this issue, we finally performed in situ
hybridization of *Fgf18* by RNAscope, eventually revealing that *Fgf18* signals were
detected in both hepatocytes and HSCs (Fig. 2f, g). We have mentioned this in the Results
section (Lines 196-207).

Thank you for the thoughtful suggestion to analyze the RNA-seq data deposited
in the previous study ⁴. As suggested, we reanalyzed GSE99010, which was derived from
RNA-seq analysis of the 4-pooled livers from CCl₄-treated mice fed a normal diet or
Western diet for 12 and 24 weeks ⁴. We found that the expression of *Fgf18* was correlated
with the expression of *Colla2* and *Col3a1* (Supplementary Fig. 4i). More importantly,
the expression of *Fgf18* was not detected in the livers of mice under either condition until
12 weeks, but increased in the livers of CCl₄-treated mice fed the normal diet, and further
increased in CCl₄-treated mice fed the western diet at 24 weeks (Supplementary Fig. 4j).
Together, these results suggest that *Fgf18* is also elevated in the livers of a murine model
that mimics human MAFLD. We have mentioned these findings in the Results section
(Lines 211-219).

**Rebuttal Fig. 2 | Determination of the specificity of commercially available anti-**
 **FGF18 antibodies.** **a**, Lysates of hepatocytes from non-Tg and *Fgf18* Tg mice were
 analyzed by immunoblotting with the indicated anti-FGF18 antibodies. mFGF18
 indicates recombinant mouse FGF18 generated in *E. coli* in-house. Please see the
 Materials and Methods section. **b**, Liver tissue sections from mice of the indicated
 genotypes fed a CDE diet for 4 weeks were immunostained with anti-FGF18 antibody
 (CSB-PA120525). Red arrowheads indicate FGF18-positive cells that appear to be
 hepatocytes.

· **Results Line 178 – 199 + Figure 3**

**Conclusion:** Here, evidence shows that hepatocyte-specific knock-out of
 **Fgf18** partially rescues the fibrogenic (but not ductular reaction) phenotype
 of the **Cflar** knock-out. This confirms **Fgf18** as a partial mediator of the **Cflar**
 knock-out phenotype. The successful knock-out of **Fgf18** & **Cflar** has to be
 shown, both protein and gene expression for all 4 conditions. Since **Cre-**
 **Alb** is used, this has to be shown for hepatocytes compared to other liver
 cells. 4 conditions are shown: **Cflar^{FF}**, **Cflar^{KO}**, **Cflar^{FF}Fgf18^{FF}**,
 **Cflar^{KO}Fgf18^{KO}**. Please elaborate on the difference between **Cflar^{FF}** and
 **Cflar^{FF}Fgf18^{FF}**.

**RESPONSE:** Thank you for pointing out this critical issue. Regarding the expression of
 *Cflar* in *Cflar^{FF}* and *Cflar^{LKO}* mice by qPCR and Western blotting, please see
 Supplementary Fig. 1a, b. As we mentioned before, it was very difficult to isolate viable

hepatocytes from the fibrotic livers; thus, we only investigated the expression of *Fgf18*
and *Cflar* in hepatocytes and NPCs in mice of the indicated genotypes before feeding the
CDE diet. As expected, the expression of *Fgf18* was reduced in the hepatocytes of
*Cflar^{LKO};Fgf18^{LKO}* mice compared to *Cflar^{FF};Fgf18^{FF}* mice (Fig. 3a, b). Notably, the
expression of *Fgf18* was elevated in the NPCs of *Cflar^{LKO};Fgf18^{LKO}* and *Cflar^{LKO}* mice
compared to the respective control floxed mice. We showed that the expression of cFLIP
was reduced in hepatocytes, but not NPCs of *Cflar^{LKO};Fgf18^{LKO}* mice compared to
*Cflar^{FF};Fgf18^{FF}* mice at both the mRNA and protein levels (Fig. 3a, b, c). We have
mentioned these results in the Results section (Lines 233-241).

Regarding the expression of FGF18 at the protein level, we detected FGF18
expression in hepatocytes from *Fgf18* Tg mice, but not non-Tg mice (Fig. 4f). While the
expression of *Fgf18* was elevated in NPCs from *Cflar^{LKO}* and *Cflar^{LKO};Fgf18^{LKO}* mice,
we could not detect FGF18 in either of them by Western blotting (Supplementary Fig.
6a-c). We recently generated an in-house human FGF18 ELISA system⁵ (please see the
enclosed in press paper). Using this ELISA system, we detected mFGF18 in the culture
supernatants of hepatocytes from *Fgf18* Tg, but not non-Tg mice (Fig. 4g). We have made
the new Figure 3a-c, Fig. 4e-g, and Supplementary Fig. 6a-c to include these results and
mentioned them in the text (lines 267-275).

We apologize for the confusion regarding the names of *Cflar^{FF}* and
*Cflar^{FF};Fgf18^{FF}*. *Cflar^{FF}* mice harbor *flox* sequences in both alleles of the *Cflar* gene, but
the wild-type allele of the *Fgf18* gene; *Cflar^{FF};Fgf18^{FF}* mice have *flox* sequences in both
alleles of both the *Cflar* and *Fgf18* genes.

**(1) why does this condition have lower Fgf18 levels?**

RESPONSE: Given that the expression of *Fgf18* was detected in both hepatocytes and
HSCs in the livers of mice, deletion of *Fgf18* in hepatocytes lowered the levels of *Fgf18*
mRNA in the livers.

**(2) wouldn't a CflarFFFgf18KO condition be a better control?**

RESPONSE: It is technically difficult to generate hepatocyte-specific *Fgf18*-deficient
mice under the *Cflar^{FF}* genetic background, because *Alb-Cre* lacks both *flox* alleles of
*Fgf18* and *Cflar*. To circumvent this problem, we have included the results using *Fgf18^{FF}*
and *Fgf18^{LKO}* mice fed normal and CDE diets (Supplementary Fig. 5). Feeding of the

CDE diet did not further exacerbate or attenuate liver fibrosis in *Fgf18^{LKO}* mice compared
to *Fgf18^{FF}* mice. We mentioned this in the Results section (lines 226-232).

• **Results Line 200 – 225 + Figure 4**

**Conclusion: *Fgf18* overexpression bypasses the need for hepatocyte**
**damage to induce characteristics of CLD: fibrosis, inflammation, and**
**angiogenesis, indicating its role in CLD. The successful overexpression of**
***Fgf18* has to be shown, both protein and gene expression for all conditions.**
**Since Cre-Alb is used, this has to be demonstrated for hepatocytes**
**compared to other liver cells. Agree with the statement that *Fgf18* is not**
**essential for CK19+ cell expansion (but overexpression stimulates ductular**
**reaction)**

RESPONSE: As suggested, we examined the expression of *Fgf18* in hepatocytes and
NPCs from non-Tg mice (*ROSA26-CAG-LST-Fgf18* mice) and *Fgf18* Tg mice (*ROSA26-*
*CAG-LST-Fgf18;Alb-Cre* mice). As expected, the expression of *Fgf18* was elevated in
hepatocytes from *Fgf18* Tg mice compared to non-Tg mice (Fig. 4e). Of note, the
expression of *Fgf18* in hepatocytes from non-Tg mice was strongly elevated compared to
hepatocytes from *Cflar^{FF}*, *Cflar^{LKO}*, or *Cflar^{FF};Fgf18^{FF}* mice (0.5 versus 1~1.5 x 10⁻⁵).
This may result from the readthrough transcripts beyond the stop codon downstream of
the *CAG* promoter in hepatocytes from non-Tg mice. We detected FGF18 expression in
hepatocytes from *Fgf18* Tg mice, but not non-Tg mice by Western blotting (Fig. 4f,
Supplementary Fig. 6a-c). We also generated an in-house human FGF18 ELISA system
⁵. We were able to detect mFGF18 in the culture supernatants of hepatocytes from *Fgf18*
489 Tg mice but not non-Tg mice. We have made the new Figure 3a-c, Fig. 4e-g, and
490 Supplementary Fig. 6a-c to include these results and described them in the text (Lines
267-275).

• **Results Line 226 – 258 + Figure 5**

**Conclusion: The RNA Sequencing data nicely shows an overlap of *Fgf18***
**overexpression and the *Cflar* knock-out further linking the fibrosis**
**phenotype of *Cflar* knock-out to *Fgf18*. A deeper analysis of CD31-CD34+**
**cells identifies HSCs as one of the affected cells by *Fgf18* overexpression.**

RESPONSE: Thank you for your appreciation of our work.

· Results Line 259 – 306 + Figure 6

We have some remarks concerning the experimental setup and
methodology:

**(1) The main goal of this experiment is to identify intercellular crosstalk.**
**Since the hepatocytes overexpress the Fgf18, why was it chosen to occlude**
**them from single-cell RNA seq analysis? (only non-parenchymal cells were**
**chosen for single cell analysis).**

RESPONSE: We initially planned to combine isolated hepatocytes and nonparenchymal
cells from non-Tg and *Fgf18* Tg mice and then subject the samples to single-cell RNA-
seq analysis. Since we could not prepare viable hepatocytes from the fibrotic livers of
*Fgf18* Tg mice, we only focused on the isolation of nonparenchymal cells. We assumed
that a few hepatocytes must have contaminated the isolated NPCs based on our previous
experience. As expected, contaminated hepatocytes and mononuclear cells were able to
analyze cell–cell communication by CellChat.

**(2) despite the selection of non-parenchymal cells, the single RNA seq**
**results still show plenty of hepatocytes. How can this be explained? The**
**definition of cell clusters is elaborate and justified. CellChat analysis nicely**
**shows the impact of Fgf18 and provides evidence for direct hepatocyte-**
**HSC crosstalk.**

RESPONSE: Please see the response to (1). Thank you for citing a critical issue regarding
how we defined the clusters. We defined each cluster based on a standard method with a
resolution of 0.5. The signature genes for each cluster are as follows. We have included
the list of signature genes for clustering in Supplementary Table 7.

**Rebuttal Table 1.** Signature genes in each cluster.

Clusters in Figure 6			Clusters in Supplementary Figure 9		
Cluster No.	Cell type	Signature genes	Cluster No.	Cell type	Signature genes
0	Hepatocyte	Alb	0	HSC	Lrat
HSC	Lrat	1	Cholangiocyte	Sox9, Igfbp6
Hepatocyte	Alb	2	Hepatocyte	Alb
HSC	Lrat	3	HSC	Lrat
Hepatocyte	Alb	4	Hepatocyte	Alb

Hepatocyte	Alb	5	LSEC	Cdh5, Cldn5
T cell	Lck	6	Hepatocyte	Alb
Monocyte	Cd14, Cx3cr1	7	LSEC	Cdh5, Cldn5
HSC	Lrat	8	Neutrophil	Ly6g, Itgam
Neutrophil	Ly6g, Itgam	9	Neutrophil	Ly6g, Itgam
Neutrophil	Ly6g, Itgam	10	Hepatocyte	Alb
Neutrophil	Ly6g, Itgam	11	Neutrophil	Ly6g, Itgam
Portal fibroblast	Thy1	12	T cell	Lck
Hepatocyte	Alb	13	Neutrophil	Ly6g, Itgam
Neutrophil	Ly6g, Itgam	14	Portal fibroblast	Thy1
B cell	Cd19	15	LSEC	Cdh5, Cldn5
Myofibroblast	Acta2	16	Cholangiocyte	Sox9, Igfbp6
Dendritic cell	Itgax	17	B cell	Cd19
Neutrophil	Ly6g, Itgam	18	Myofibroblast	Acta2
Neutrophil	Ly6g, Itgam	19	Dendritic cell	Itgax
T cell	Lck	20	Neutrophil	Ly6g, Itgam
Neutrophil	Ly6g, Itgam	21	Myofibroblast	Acta2
			Neutrophil	Ly6g, Itgam
			T cell	Lck
			HSC	Lrat
			Myofibroblast	Acta2
			HSC	Lrat

**Line 301: Fig2 refers to the CflarKO mice, which does not seem relevant to**
**the statement made in this sentence.**

RESPONSE: We apologize for citing the wrong figure number. We have changed to Fig.
5b and 6c and mentioned them in the Results section (Lines 369-377).

**· Results Line 307 – 328 + Figure 7**

**Concerning the HSC culture, remarks are to be made. (1) If the method**
**section is understood correctly, lineage marker-negative NPCs, the same**
**cells that were used for single-cell RNA sequencing in figure 6, were plated**
**for 3days. Thus, this cell culture consists of multiple cell types including**
**hepatocytes (that overexpress Fgf18), stellate cells, myofibroblast, portal**

fibroblasts, and several immune cells as shown in the single RNA
sequencing data. This culture is thus a co-culture of multiple days (even if
cells don't attach) which can highly influence HSC activation status. (2)
After three days, cells are split and replated for an unclear amount of time
(3) It is unclear why HSCs were isolated from Fgf18 Tg mice which, as
shown by RNA seq data, influences their activation status. (4) why were
these cells culture with Y27632, a ROCK Inhibitor? In conclusion, these
experiments are not performed on freshly isolated quiescent HSCs which
makes further interpretation of results difficult and unnecessarily
complicated. Density gradient isolation or ultraviolet sorting methods to
culture pure quiescent HSCs from the start are suggested. In any way, SMA
staining and light microscopy are required to show the phenotype of the
HSCs. Staining for an HSC marker (e.g. Desmin) with quantification of the
amount of HSCs present within the cell culture at the start of the culture
should be provided.

RESPONSE: Thank you for pointing out the critical point. As suggested, we isolated
HSCs using the Nycodenz density gradient centrifugation. We then plated HSCs and
determined the percentages of HSCs following the depletion of contaminated
nonadherent cells before stimulation. HSCs and other adherent cells were determined
based on the signals for UV (Supplementary Fig. 13a). Moreover, we immunostained
adherent cells with anti-desmin antibody (Supplementary Fig. 13c). In both experiments,
more than 80% were UV⁺ or desmin⁺ HSCs. These cells were used for subsequent
experiments.

We then stimulated isolated HSCs with TGFβ or FGF18 for 24 hours. We found
that TGFβ increased the expression of *Colla2*, *Col3a1*, *Acta2*, and *Fgf18*. However,
FGF18 did not upregulate, but inhibited the TGFβ-induced upregulation of these genes.
Of note, FGF18 enhanced the proliferation of HSCs on days 3 and 5 after stimulation.
Consistent with the enhancement of proliferation, FGF18, but not TGFβ, induced
upregulation of *Ccnd1*, encoding cyclin D1, which is responsible for the transition from
G1 to G2/M phase. Together, these results suggest that FGF18 preferentially induces the
proliferation of HSCs, while keeping HSCs quiescent.

In contrast, these results appeared inconsistent with our previous results using
hiPSC-derived HSC-like cells. The difference may be caused by the maturation stages of
the hiPSC-derived HSC-like cells used in our study. To avoid confusion, we have deleted
the results using hiPSC-derived HSC-like cells, which will be further investigated in the

future studies. We have made a new Fig. 7 to include these results and mentioned them
in the Results and Discussion section (Lines 379-407, 478-510).

**Concerning iPSC HSC, light microscopy, SMA staining, and lipid droplet**
**content have to be shown to prove a quiescent to activated-like process.**

RESPONSE: As mentioned above, we have deleted the data on iPSC-derived HSCs in
the manuscript.

**Based on the results of this section, the reviewer cannot agree with the**
**statement that Fgf18-Tgfb constitutes a feed-forward loop: (1) in mouse**
**HSC: Fgf18 does not induce collagens, Fgf18 nor Tgfb. (2) in iPSC HSC,**
**Tgfb does not induce Fgf18.**

RESPONSE: As described above, we deleted the results using iPSC-derived HSC-like
cells. We found that TGF β 1 induced *Fgf18* expression in HSCs and hepatocytes; however,
FGF18 suppressed the expression of *Tgfb2* and *Tgfb3* and the TGF β 1-induced
upregulation of profibrotic genes. We have made the new Fig. 7f-i to include these results
and described them in the Results section (Lines 399-402).

**· Discussion**

**Line 330: Cf MAFLD comment above**

**Line 336: Cf Feedforward loop above**

**Line 368: the way the sentence is written alludes to the fact that HSCs**
**differentiate towards either myofibroblasts or portal fibroblasts. Generally,**
**portal fibroblasts are considered distinct from HSC with their fibrogenic**
**properties.**

RESPONSE: As suggested, we have changed the text to “Thus, FGF18 mainly induces
the expansion of weakly activated or quiescent *Lrat*⁺ HSCs and does not fully promote
their terminal differentiation toward *Acta2*⁺ myofibroblasts” (Lines 461-463).

**Line 377: single-cell data in this manuscript clearly shows Hgf expression**
**in HSCs, but not in portal fibroblasts or myofibroblasts (Fig6c). “may**
**produce” should be rephrased. Additionally, this could easily be tested in**
**vitro with correct HSC cultures as explained above**

RESPONSE: Thank you for the helpful advice. We isolated primary HSCs from wild-
type mice and tested whether FGF18 induced the upregulation of *Hgf*. FGF18 did not
upregulate the expression of *Hgf* in HSCs. In contrast, TGF β suppressed the expression
of *Hgf* (Supplementary Fig. 12c). Of note, the expression of *Hgf* was high in HSC
compared to hepatocyte (Supplementary Fig. 11b), and FGF18 induced the proliferation
of HSCs (Fig. 7b). These results suggest that FGF18 may induce the proliferation of
hepatocytes through an increase in the number of HSCs that express HGF at high levels.
We mentioned this in the text (Lines 369-377).

**· Materials & Methods**

**Line 441: If *Cflar* flox/flox mice were provided, please provide text on how**
**hepatocyte-specific *Cflar* KO mice were generated.**

RESPONSE: As suggested, we have added the reference and explained the information
on how to generate the hepatocyte-specific *Cflar*-deficient (*Cflar*^{LKO}) mice in the Results
and Materials and Methods section (Lines 131-134, 562-563).

**Line 441: Reference should be provided for the RIKEN database and initial**
**publication of the *Fgf18*^{FF} mice. A statement on ethical animal use is**
**missing in this section.**

RESPONSE: As suggested, we have mentioned the RIKEN database number of *Fgf18*^{FF}
mice along with the initial characterization of these mice ⁶ (lines 559-560). We also added
a statement on ethical animal use (Lines 568-570).

**Line 536: The reviewer cannot find these qPCR results**

RESPONSE: We apologize for not including the raw data from our qPCR analysis. As
suggested, we have included the results of the expression of transfected genes by HTVi.
We made Supplementary Fig. 3a and included it in the Results section (Lines 188-190).

**· MAFLD pathogenesis**

**If MAFLD pathogenic conclusions are to be made, the reviewer suggests**
**several further analyses: (1) RNA Scope for *Fgf18* in healthy vs MAFLD**
**patient livers, (2) analysis of single-cell RNA Seq of human livers (MAFLD**

**vs healthy – eg GSE136103): what cells express Fgf18, perhaps Cellchat for**
**Fgf18?**

**Since an Fgf18 hepatocyte overexpression model is used, it should be**
**shown that in chronic liver disease, hepatocytes have a higher expression**
**of Fgf18. Overall, a setup of hepatocyte-specific Fgf18 knock-out vs**
**hepatocyte-specific Fgf18 overexpression vs wild-type mice subjected to a**
**metabolic MAFLD model (e.g. high-fat diet + low dose CCl4 – Tsuchida T et**
**al journal of hepatology 2018 <http://dx.doi.org/10.1016/j.jhep.2018.03.011>)**
**vs standard diet (ie 6 conditions) seems the most obvious choice for**
**drawing conclusions on MAFLD pathogenesis.**

RESPONSE: We completely agree with the reviewer’s statement along with the other
reviewers’ comments that the CDE diet is not appropriate for inducing a NASH model in
mice. Thus, we have deleted the statement about NASH and have focused our study on
liver fibrosis associated with chronic liver injury (please see the Introduction section).

For human samples, we have examined the expression of human *FGF18* in liver
biopsy samples from various liver diseases by qPCR. Although the expression of *FGF18*
varied among diseases, the expression of *FGF18* was correlated with that of *COL1A1* and
*ACTA2*, which was consistent with the results of mice fed the normal diet or Western diet
treated with CCL₄ for 24 weeks (Supplementary Fig. 4i). Thus, we concluded that FGF18
may also be involved in liver fibrosis in human patients. To include these new results, we
have made Fig. 8 and described the findings in the Results section (Lines 409-416).

We really appreciate that the reviewer recommended us a good MAFLD model.
However, at this moment, we believe that the proposed experiments are beyond the scope
of the present study and should be performed in the future studies.

Regarding the reanalysis of GSE136103⁷, UMAP did not include clusters
representing *LRAT*⁺ cell from healthy and cirrhotic human livers (Rebuttal Fig. 4a). The
lack of HSC clusters even from healthy livers may be due to the difficulty in the
procedures to isolate nonparenchymal cells from human livers. Although the fact that
FGF18 was not expressed in healthy hepatocyte is consistent with our present results, the
reason for the lack of FGF18 expression in the cirrhotic livers is currently unknown
(Rebuttal Fig. 4b). Since we observed elevated expression of *FGF18* in liver biopsied
samples from liver diseases (Fig. 8), it is unclear why hepatocyte in cirrhotic livers did
not express *FGF18*. One plausible scenario would be that hepatocytes in severe fibrotic
areas may not be isolated from cirrhotic livers. Alternatively, elevation of FGF18 in
hepatocytes is not always observed in cirrhotic livers, but may depend on the cause of

cirrhosis, such as viral infection, alcohol, and NASH. Further study will be required to
 address this issue. We have only included the reanalysis data of GSE136103 in Rebuttal
 Fig. 4 and mentioned the data in the Rebuttal letters.

**Rebuttal Figure 3 | scRNA-seq data of healthy and cirrhotic human livers.**
 Reanalysis of scRNA-seq data (GSE136103). **a**, UMAP visualization of single cell
 clusters isolated from healthy ($n = 3$) and cirrhotic ($n = 3$) human livers based on gene
 expression profile similarity. Each color represents a distinct cell population. **b**, UMAP
 visualization of the expression of the indicated genes. Note that there is no cluster
 expressing *LRAT*.

**Reviewer #3 (Remarks to the Author):**

**The manuscript by Tsuchiya and colleagues presents an extensive set of**
**data that demonstrates the role of FGF 18 in the stimulation of hepatic**
**fibrosis. The study is well written and performed. However, a few areas of**
**concern exist.**

**1. The rationale for utilizing the CflarLKO mice is not clearly stated in the**
**manuscript. Also, the choice of CDE hepatotoxic diet rather than a**
**traditional HFD (western diet). This model seems to have an excessive**
**ductular reaction that is not present in NASH while the amount of fibrosis**
**is not that elevated.**

RESPONSE: We appreciate you bringing this critical issue in our study to our attention.
Our initial aim was to identify genes that promote the proliferation of hepatocytes,
CK19⁺ cells, or HSCs following cell death injury using *Cflar^{LKO}* mice, which are highly
susceptible to TNF-induced cell death (Piao et al., Sci Signaling 2012; Hepatology
2017). After feeding *Cflar^{LKO}* mice with various diets, including the high-fat, CDE, and
DDC diets, we found that the CDE diet dramatically exacerbated liver fibrosis in
*Cflar^{LKO}* mice compared to *Cflar^{FF}* mice. Therefore, we focused on the liver injury
induced by the CDE diet.

**2. Figure 2d,e the Ki67 increase in the animals treated with Fgf18 is**
**minimal. This should be discussed. Co-staining should be performed to**
**eliminate Ki67 expression in cholangiocytes.**

RESPONSE: As suggested, the numbers of Ki67⁺ cells did not dramatically increase in
FGF18-injected livers. We analyzed Ki67⁺ cells in the livers at one week after HTVi.
This may be one of the reasons there was no dramatic increase in Ki67⁺ cells.
Alternatively, we assume insufficient space existed for the proliferation of hepatocytes
and desmin⁺ HSCs in the normal liver.

As suggested, we performed co-staining of liver sections with anti-Ki67 and
anti-CK19 antibodies (a hallmark of cholangiocytes in the normal liver). The majority
of Ki67⁺ cells did not express CK19, excluding the possibility that Ki67⁺ cells were
cholangiocytes (Supplementary Fig. 3). Given that desmin⁺ cells were also slightly
increased in the livers injected with *Fgf18* (Fig. 2d, e), Ki67⁺ cells may also contain

desmin⁺ HSCs as well as hepatocytes. We have made Fig. 2d and e and Supplementary
Fig. c-e and mentioned these data in the Results section (Lines 190-195).

**3. Wild-type fed control diet and CDE should be included at least initially**
**for comparison to observed features in the CflarLKO mice.**

RESPONSE: As suggested, wild-type C57BL/6 mice were fed the CDE diet for 4
740 weeks. Liver fibrosis was induced in wild-type mice fed the CDE diet, although the
741 severity of liver fibrosis was not as dramatic as that in *Cflar*^{LKO} mice fed the CDE diet.
We have made a new Supplementary Fig. 2 to include these results and mentioned them
in the Results section (Lines 153-156).

**4. Only ALT levels are measured. AST and alkaline phosphatase levels**
**should also be included.**

RESPONSE: As suggested, we measured serum AST and ALP levels from the mice we
used in the experiments. However, we could not measure these enzyme levels in several
mice, because the sera had been used for other experiments. We have included the
results in Figs. 3d and 4h and Supplementary Figs. 2a, 4a and e, and 5c and mentioned
them in the Results section (Lines 242-244, 276).

**5. Rather than use HSCs from iPS cells, the authors should treat primary**
**HSCs or isolated HSCs with FGF18.**

RESPONSE: Thank you for identifying this critical point. As suggested, we extensively
analyzed the FGF18-induced responses using primary isolated HSCs. We found that
FGF18 mainly induced the proliferation of primary HSCs, but did not upregulate
profibrotic genes, such as *Col1a2*, *Col3a1*, or *Acta2*. Moreover, FGF18 suppressed
TGFβ1-induced upregulation of these profibrotic genes. In contrast, our previous
studies revealed that FGF18 upregulated profibrotic gene expression in hiPSC-derived
HSC-like cells, although the difference between the two results is currently unknown.
To avoid confusion, we have deleted the results using hiPSC-derived HSC-like cells.

Although FGF18 inhibited the expression of profibrotic genes, FGF18
transgenic mice spontaneously developed liver fibrosis. How do we reconcile
inconsistent results? One plausible scenario would be that FGF18 initially triggers the
proliferation of HSCs at relatively quiescent stages; then, profibrotic cytokines, such as

TGF β , transform qHSCs to aHSCs that highly express profibrotic genes. Whether
FGF18 promotes the attenuation of fibrosis may depend on the duration of HSC
exposure to FGF18. As reported previously¹, once liver fibrosis develops, the
administration of FGF18 may attenuate fibrosis, because HSCs cannot efficiently
proliferate under fibrotic liver conditions. In sharp contrast, when *Fgf18* is
overexpressed in the livers from embryonic stages, HSCs proliferate in response to
FGF18, and an increase in HSCs at relatively quiescent stages may be sufficient to
induce liver fibrosis in *Fgf18* Tg mice. Moreover, germline deletion of *Fgf18* in
hepatocytes diminished the proliferation of HSCs, thereby partially attenuating liver
fibrosis in *Cflar^{LKO};Fgf18^{LKO}* mice fed the CDE diet compared to *Cflar^{LKO}* mice fed the
CDE diet. We have made a new Fig. 7 to include these results and mentioned them in
the Results and Discussion sections (Lines 379-407, 478-496)

**6. Staining for HSCs should be performed in all of the animal treatment**
**groups.**

RESPONSE: Thank you for the thoughtful suggestion. Since desmin is a universal
marker for HSCs, we stained liver sections with anti-desmin antibody. Consistent with
the development of liver fibrosis, the desmin⁺ areas were increased in the livers of mice
investigated in our study. We have included the results in Figs. 1c and g, 2d and e, 3f
and h, and 5k and l, as well as Supplementary Figs. 2b and f, 5e and i. Moreover, we
have included *Des* in the heatmap and violin plots (Figs. 2b, 6c). We have mentioned
these findings in the Results section (Lines 168, 182, 191, 247, 319, 339).

**7. Interactions between HSCs and hepatocytes stimulated with TGFbeta**
**could be evaluated in co-culture experiments. In addition, Hepatocytes**
**should be stimulated directly with FGF18 and evaluated for factors that**
**are profibrotic.**

RESPONSE:

Thank you for your thoughtful suggestion. As suggested, we cocultured primary
hepatocytes and isolated HSCs at a ratio of 10:1, which is similar to the ratio of these
cells *in vivo*. While TGF β moderately elevated the expression of profibrotic genes, such
as *Colla2*, *Col3a1*, and *Acta2*, in the absence of HSCs, the expression levels were not
further increased in the presence of HSCs (Rebuttal Fig. 4). Moreover, under the same
experimental conditions, FGF18 did not upregulate profibrotic genes or *Ccnd1*

(Rebuttal Fig. 4). However, we do not think that the simple coculture method used in
 our study are suitable for mimicking the interaction between hepatocytes and HSCs *in*
 *vivo*. We assume that optimizing the conditions to improve coculture experiments using
 primary hepatocytes and HSCs seems beyond the present study. Because we hesitate to
 include these results in the manuscript, we have only included the results of coculture
 experiments in the Rebuttal letter. On the other hand, we have included the results using
 hepatocytes stimulated with TGF β or FGF18 in Fig. 7j and mentioned them in the
 Results section (Lines 402-403).

 **Rebuttal Fig. 4 | HSCs do not enhance or attenuate the expression of profibrotic**
 **genes and *Ccnd1*.** Primarily isolated hepatocytes and HSCs were cocultured and
 stimulated with TGF β 1 or FGF18 for 24 hours. The expression of the indicated genes
 was determined by qPCR. Results are mean \pm SD of triplicate samples. Results are
 representative of two independent experiments.

**Rebuttal letter references**

- 1. Tong, G., Chen, X., Lee, J., Fan, J., Li, S., Zhu, K., Hu, Z., Mei, L., Sui, Y., Dong, Y., et al.
(2022). Fibroblast growth factor 18 attenuates liver fibrosis and HSCs activation via the
SMO-LATS1-YAP pathway. *Pharmacol Res* 178, 106139. 10.1016/j.phrs.2022.106139.
- 2. Mullany, L.K., White, P., Hanse, E.A., Nelsen, C.J., Goggin, M.M., Mullany, J.E., Anttila,
C.K., Greenbaum, L.E., Kaestner, K.H., and Albrecht, J.H. (2008). Distinct proliferative
and transcriptional effects of the D-type cyclins in vivo. *Cell Cycle* 7, 2215-2224.
10.4161/cc.7.14.6274.
- 3. Piao, X., Komazawa-Sakon, S., Nishina, T., Koike, M., Piao, J.H., Ehlken, H., Kurihara,
H., Hara, M., Van Rooijen, N., Schutz, G., et al. (2012). c-FLIP maintains tissue
homeostasis by preventing apoptosis and programmed necrosis. *Sci Signal* 5, ra93.
10.1126/scisignal.2003558.
- 4. Tsuchida, T., Lee, Y.A., Fujiwara, N., Ybanez, M., Allen, B., Martins, S., Fiel, M.I.,
Goossens, N., Chou, H.I., Hoshida, Y., and Friedman, S.L. (2018). A simple diet- and
chemical-induced murine NASH model with rapid progression of steatohepatitis, fibrosis
and liver cancer. *J Hepatol* 69, 385-395. 10.1016/j.jhep.2018.03.011.
- 5. Tsuchiya, Y., Komazawa-Sakon, S., Tanaka, M., Kanakogi, T., Moriwaki, K., Akiba, H.,
Yagita, H., Okumura, K., Entzminger, K.C., Okumura, C.J., et al. (2023). A high-
sensitivity ELISA for detection of human FGF18 in culture supernatants from tumor cell
lines. *Biochem Biophys Res Commun* *in press*.
- 6. Kimura-Ueki, M., Oda, Y., Oki, J., Komi-Kuramochi, A., Honda, E., Asada, M., Suzuki,
843 M., and Imamura, T. (2012). Hair cycle resting phase is regulated by cyclic epithelial
FGF18 signaling. *J Invest Dermatol* 132, 1338-1345. 10.1038/jid.2011.490.
- 7. Ramachandran, P., Dobie, R., Wilson-Kanamori, J.R., Dora, E.F., Henderson, B.E.P., Luu,
846 N.T., Portman, J.R., Matchett, K.P., Brice, M., Marwick, J.A., et al. (2019). Resolving the
847 fibrotic niche of human liver cirrhosis at single-cell level. *Nature* 575, 512-518.
10.1038/s41586-019-1631-3.

REVIEWERS' COMMENTS

Reviewer #1 (Remarks to the Author):

The authors has addressed the main issues I raised and the manuscript has improved significantly, but there are still several issues that need to be addressed.

1, The authors are still unable to explain the contradictory issue of how FGF18 promotes HSC proliferation and enhances liver fibrosis by upregulating *Ccnd1* and promoting TGF β expression, while also improving fibrosis by inhibiting the expression of fibrosis-promoting genes induced by TGF β . Furthermore, it is important to compare germinal and shRNA-mediated deletion in adult mice in order to explore whether FGF18 could alleviate or worsen liver fibrosis.

2, Hepatic FGF18 expression directly regulated by *Cflar*? Are there any transcription factors involved in this process?

3, In Fig3, it is necessary to include the *Fgf18*LKO group, especially to compare the differences between the *Fgf18*LKO and *Cflar*LKO;*Fgf18*LKO groups.

4, Since FGF18 can regulate inflammation, does this mean that FGF18 directly affects the development of liver fibrosis through inflammation?

5, FGF18 activates the proliferation of HSCs through *Ccnd1*. Does the classic FGF-promoted proliferation pathway FGF/FGFR/Ras/ERK also play an important role in this process?

Further, how does FGF18 promote the upregulation of *Ccnd1* expression?

Reviewer #2 (Remarks to the Author):

Fibroblast growth factor 18 activates hepatic stellate cells that lead to liver fibrosis

- What are the noteworthy results?

These remain the same as after first review:

In their report, Tsuchiya and colleagues use a hepatocyte-specific Cflar knock-out model to show increased liver (hepatocyte) damage leading to several aspects of chronic liver disease (CLD) including ductular reaction and fibrosis. Upon usage of the CDE diet, these knock-out mice develop a more severe phenotype than their wild-type counterparts. Upon more deep investigation of these models, Fgf18 is identified as a potent (but partial) mediator of the Cflar knock-out phenotype. Hepatocyte-specific knock-out of Fgf18 in hepatocyte-specific Cflar knock-out mice alleviates the phenotype and hepatocyte-specific Fgf18 overexpression mimics the hepatocyte-specific Cflar knock-out mice phenotype. Moreover, hepatocyte-specific Fgf18 overexpression shows clear signs of spontaneous development of CLD: inflammation, ductular reaction, fibrosis, and angiogenesis, bypassing the need for hepatocyte damage (ie ALT levels normal). Next, the Fgf18 function is linked to hepatic stellate cell activation (HSC) activation including insightful single cell RNA Seq analysis. Fgf18 exhibits correlation with fibrogenic genes, suggesting its plausible relevance to human chronic liver disease.

- Will the work be of significance to the field and related fields? How does it compare to the established literature? If the work is not original, please provide relevant references.

These remain the same as during first review.

As mentioned by the authors, the findings presented here contrast that of a recent publication by Tong et al. However, the experimental setup is different: overexpression of Fgf18 in this manuscript is genetic whereas Tong et al use a vector-based method. Fgf18 knock-out in this manuscript consists of (combined Cflar-)Fgf18 hepatocyte-specific knock-out whereas Tong et al use an HSC-specific knock-out. Additionally, experimental models differ between manuscripts. More experiments in the future might elucidate these discrepancies.

If anything, both manuscripts underline a role for Fgf18 in CLD which is novel and thus provides a significant contribution to the field.

- Does the work support the conclusions and claims, or is additional evidence needed?

The authors have undertaken a significant revision of the manuscript. The evidence has been expanded where necessary, providing greater elaboration. The claims presented in this paper are substantiated by the data presented.

- Are there any flaws in the data analysis, interpretation, and conclusions? - Do these prohibit publication or require revision?

Numerous issues have been addressed, particularly those related to the experimental setup and cell culture. A few minor comments may still need attention. Overall, the manuscript has been adapted and has successfully addressed several critical concerns. After these minor comments have been discussed, this reviewer believes the manuscript is ready for publication.

- Is the methodology sound? Does the work meet the expected standards in your field?

The methodology differs from more conventional chronic liver disease (CLD) models, such as CCl₄, bile duct ligation, or metabolic MASLD, by utilizing the less conventional CDE model.

The quality of HSC culture has been highly improved.

The strength of the genetic mouse models deserves emphasis, particularly in investigating the role of Fgf18 in liver function. The successful overexpression or knock-out of indicated genes has been addressed.

- Is there enough detail provided in the methods for the work to be reproduced?

Yes.

Sex and Gender Equity in Research – SAGER – guidelines

DDC: male mice, other: female mice. The method section might further elaborate on the choice of sex.

Specific comments

- ***Language***

A revision of language has been done and been improved. In the abstract, line 59 might differently expressed.

- ***Title***

The title has been adjusted and fits better.

- ***NAFLD – MASLD - MASLD***

During the initial review, the nomenclature of NAFLD was updated to MAFLD. However, in the meantime, the nomenclature has once more evolved to MASLD/MASH. Kindly ensure consistency with this updated terminology across the manuscript. Reference: DOI: 10.1016/j.jhep.2023.06.003.

The authors have avoided making assertions about the pathogenesis of MASLD, considering that the CDE diet does not accurately represent this disease.

- ***Introduction***

MASLD comment: cf supra

The introduction has been expanded to encompass a section on hepatic stellate cells (HSCs). It would be beneficial to provide better clarification regarding the presence of quiescent HSCs in a healthy liver, as well as to emphasize that their activation occurs subsequent to liver injury. (lines 84 – 88)

- ***Results – Lines 130 - 174 (previous lines 119 – 148) + Figure 1***

Conclusion: Hepatocyte-specific Cflar knock-out mice spontaneously develop characteristics of CLD. When exposed to the CDE diet, they develop a more severe phenotype.

The experimental setup has been modified as requested during the initial review round. Comprehensive data is now accessible for all four conditions (WT & KO +/- CDE). Additionally, the successful knock-out/down of Cflar specifically in hepatocytes (both protein and gene expression) has been incorporated. These modifications effectively address the primary concerns in this section. Consequently, the conclusions are adequately supported by substantial evidence.

The authors mention the lack of statistical significance when comparing 4 groups instead of 2 groups. This is due to decreasing power as the amounts of groups to compare increase. This reviewer thinks that the absence of statistical significance in certain instances does not undermine the validity of the claims made. Moreover, supplementary figure 1 appropriately addresses this matter.

- **Results – Lines 176 - 220(previous 149 – 177) + Figure 2**

Conclusion: RNA Seq analysis and overexpression techniques are purposefully used to reveal Fgf18 as a potential mediator of the Cflar knock-out phenotype.

Considering the experimental setup of Figure 2c: Are these conditions the same as those in Figure 1? If so, please use consistent x-axis legends for both figures to enhance clarity. If not, kindly indicate the differences between the setups of Figure 1 and Figure 2c.

HTVi overexpression:

(1) As requested after the initial review, successful overexpression of gene expression is demonstrated, although not for protein.

(2) As suggested after the initial review, staining of the eGFP control is provided.

(3) While the methods section specifies that the HTVi should selectively overexpress the mentioned genes in hepatocytes, the authors were unable to demonstrate this specific overexpression in hepatocytes when compared to other liver cells. Nevertheless, the authors do not assert that the differences resulting from overexpression are specifically linked to hepatocytes.

The authors have responded to and substantiated their choice for a hepatocyte-specific approach targeting Fgf18, as evident in Figure 2 of the rebuttal.

As recommended, the authors have re-evaluated GSE99010, further reinforcing the evidence for Fgf18 expression across multiple models of liver fibrosis.

- **Results Line 221 - 256 (previous lines 178 – 199) + Figure 3**

Conclusion: Here, evidence shows that hepatocyte-specific knock-out of Fgf18 partially rescues the fibrogenic (but not ductular reaction) phenotype of the Cflar knock-out. This confirms hepatocyte derived Fgf18 as a partial mediator of the Cflar knock-out phenotype.

Following the request made in the first review round, successful knock-out/down of Cflar is demonstrated for both protein and gene expression, including a comparison between hepatocytes and other liver cells. However, for Fgf18, this knock-down is only illustrated for gene expression, with protein data lacking. This matter is addressed in Supplementary Figure 6, which appears to be linked to Fgf18 protein expression levels already being below the detection limit under normal conditions.

Lines 236 – 240 refer to HSCs, but this actually seems to be referring to NPC.

During the initial review round, numerous comments were provided regarding the selection of the four conditions (Cflar^{FF}, Cflar^{KO}, Cflar^{FF}Fgf18^{FF}, Cflar^{KO}Fgf18^{KO}). In their response to the reviewer's comments, the authors provide an explanation for their choices of these conditions. Supplementary Figure 5 has been added to illustrate that a knockout of Fgf18 (Fgf18^{LKO}) alone does not induce any alterations in

the phenotype of healthy mice or those subjected to a CDE diet. Due to technical limitations, generating Cflar^{FF}Fgf18^{LKO} mice is not feasible, and as a substitute, Fgf18^{LKO} mice have been used.

- **Results Line 257 - 296 (previous lines 200 – 225) + Figure 4**

Conclusion: Fgf18 overexpression bypasses the need for hepatocyte damage to induce characteristics of CLD: fibrosis, inflammation, and angiogenesis indicating its role in CLD.

As requested following the initial review round, successful overexpression of Fgf18 for both protein and gene expression, as well as a comparison between hepatocytes and other liver cells, has been demonstrated.

- **Results Line 297 – 328 (previous lines 226 – 258) + Figure 5**

Conclusion: The RNA Sequencing data nicely shows an overlap of Fgf18 overexpression and the Cflar knock-out further linking the fibrosis phenotype of Cflar knock-out to Fgf18. A deeper analysis of CD31-CD34+ cells identifies HSCs as one of the affected cells by Fgf18 overexpression.

No comments.

- **Results Lines 329-378 (previous lines 259 – 306) + Figure 6**

CellChat analysis nicely shows the impact of Fgf18 and provides evidence for direct hepatocyte-HSC crosstalk.

After first review round, some remarks were made, the authors have responded to these issues:

(1) Concerning the exclusion of hepatocytes, technical difficulties obstructed the goal of performing scRNA on hepatocytes.

(2) despite the selection of non-parenchymal cells, the single RNA seq results still show plenty of hepatocytes. The authors have replied to this issue.

Line 372: please add “in HSCs” after “upregulate the expression of Hgf” for clarity

- **Results Line 379 - 408 (Previous lines 307 – 328) + Figure 7**

Following the initial review round, significant concerns were raised regarding the culture of HSCs. The authors have addressed these issues by concentrating on a single culture method involving Nycodenz-based HSC isolation. The validity of this culture approach is demonstrated, and TGFbeta experiments indicate the susceptibility of these HSCs to fibrogenic stimulation.

Using these cultures, Fgf18 is shown to stimulate only the proliferation of HSCs, without inducing an increase in the expression of fibrogenic genes. It would be interesting to observe if the staining for αSMA in these three conditions (untreated, TGFb-stimulated, & FGF18-stimulated) reflects any alterations in the phenotype of these cells.

Based on these findings, TGFbeta and Fgf18 appear to play a role in an auto-regulatory mechanism.

A portion of figure 7a appears to be replicated in Fig7h, or were these experiments repeated with an additional fourth condition?

- **Results Line 409 - 416 + Figure 8**

The correlation between Fgf18 expression and Col1a1 & Acta2 expression is illustrated.

The authors have re-analysed GSE136103. However, their search for HSCs focused on LRAT+ cells, which yielded no results. It's important to note that the original paper's data analysis proceeded as follows: first, the identification of mesenchymal cells based on markers such as PDGFRB, ACTA2, COL1A1, COL1A2, COL3A1, DES, and DCN. Subsequently, the recognition of hepatic stellate cells (HSCs) using RGS5 & PDGFRA as a marker for myofibroblasts within the cirrhotic niche. It is strongly recommended to re-analyze this data using the same approach as the original manuscript, employing RGS5 versus PDGFRA as markers.

- ***Discussion***

Line 420: this line directly contradict the claims made in supplemental figure 5. Please address this.

Line 442: exploring how Fgf18 promotes HSC proliferation without enhancing fibrogenesis, and what triggers the activation of these proliferated HSCs, could enhance the discussion's depth.

- ***Materials & Methods***

The authors have responded adequately to all comments concerning the materials and methods.

- ***MASLD pathogenesis***

The authors have refrained from making conclusions on MASLD pathogenesis, considering the limitations of the CDE diet model.

- ***Summary Figure:***

A summary figure could provide added clarity and value to the manuscript.

Reviewer #3 (Remarks to the Author):

The authors have addressed my concerns.

REVIEWERS' COMMENTS

Reviewer #1 (Remarks to the Author):

The authors has addressed the main issues I raised and the manuscript has improved significantly, but there are still several issues that need to be addressed.

1, The authors are still unable to explain the contradictory issue of how FGF18 promotes HSC proliferation and enhances liver fibrosis by upregulating *Ccnd1* and promoting TGF β expression, while also improving fibrosis by inhibiting the expression of fibrosis-promoting genes induced by TGF β . Furthermore, it is important to compare germinal and shRNA-mediated deletion in adult mice in order to explore whether FGF18 could alleviate or worsen liver fibrosis.

RE: Thank you for highlighting the key points. Undoubtedly, understanding the intricate mechanisms governing FGF18-induced liver fibrosis is a complex endeavor, and the contribution of FGF18 to liver fibrosis may vary depending on the specific context. Notably, TGF β upregulated the expression of profibrotic genes in HSCs but did not trigger the proliferation of HSCs (Fig. 7a-c). Activation of HSCs, rather than their strong proliferation, seems sufficient to induce mild liver fibrosis, as observed in wild-type mice fed the CDE diet for 4 weeks, or mice fed the normal diet or the Western diet with CCl₄ for 12 weeks (Supplementary Figs. 2 and 4j).

As livers progress to advanced fibrosis, it is reasonable to speculate that the proliferation of HSCs may become necessary. When the strength of the stimuli exceeds a certain threshold, the expression of *Fgf18* is strongly induced (Fig. 2c). This situation was observed in the livers of *Cflar^{LKO}* mice on the CDE diet for 4 weeks, or mice on the normal diet or the Western diet treated with CCl₄ for 24 weeks (Fig. 1 and Supplementary Fig. 4j). Assuming that the expansion of HSCs requires some space, it is reasonable to speculate that FGF18 suppresses the production of extracellular matrix to make space for HSC proliferation (Fig. 7a, h). Furthermore, the expression levels of profibrotic genes in HSCs were relatively higher than those in hepatocytes, even if their expression levels declined in the presence of FGF18 (Fig. 7j and k). This finding indicates that the increased HSCs by FGF18 stimulation play an important role in the accumulation of collagens and extracellular matrix. Moreover, previous studies reported that scar-associated

macrophages differentiate from circulating monocytes, are recruited to the fibrotic niche, and interact with PDGFR α ⁺ collagen-producing mesenchymal cells, resulting in the activation of several profibrogenic pathways^{1 2}. Indeed, FGF18 induced the expression of inflammatory cytokines and chemokines, thereby recruiting immune cells, including scar-associated macrophages. Thus, proliferating HSCs induced by FGF18 further respond to profibrotic stimuli derived from scar-associated macrophages, such as platelet-derived growth factor (PDGF) and IL-1 β ¹, ultimately contributing to the development of liver fibrosis.

When researchers focus solely on the role of FGF18 in suppressing the expression of profibrotic genes, it appears that FGF18 may mitigate liver fibrosis under specific conditions, such as the transient expression of *Fgf18*, as noted by Tong et al³. However, in the long term, FGF18 seems to stimulate HSC proliferation, and subsequently, the proliferating HSCs become responsive to subsequent profibrotic stimuli, ultimately leading to liver fibrosis (Fig. 4).

From a therapeutic perspective, it would be intriguing to investigate the effect of shRNA-mediated deletion of *Fgf18* in adult mice with liver fibrosis. However, we believe that these requested experiments may extend beyond the scope of our current study and should be reserved for future research.

We have discussed these points in the Discussion (Lines 495-525).

2, Hepatic FGF18 expression directly regulated by Cflar? Are there any transcription factors involved in this process?

RE: Thank you for citing this interesting point. At this moment, we do not think that the expression of *Fgf18* in hepatocytes is directly regulated by *Cflar*. We hypothesize that deletion of *Cflar* in hepatocytes results in an increase in the number of apoptotic hepatocytes (Fig. 1c). Then, apoptotic hepatocytes are engulfed by macrophages, resulting in TGF β production. Then, TGF β may induce the expression of *Fgf18*. We have described this in the Discussion (Lines 443-451) and illustrate it in a new Figure 8b.

3, In Fig3, it is necessary to include the Fgf18LKO group, especially to compare the differences between the Fgf18LKO and CflarLKO;Fgf18LKO groups.

RE: Thank you for noting this critical issue. As suggested, to compare the extent of liver injury and fibrosis in the livers of *Fgf18^{LKO}*, *Cflar^{LKO}*, and *CflarFgf18^{LKO}* mice, we integrated the results of *Fgf18^{FF}* and *Fgf18^{LKO}* mice fed the CDE diet for 4 weeks in Supplementary Figure 5 with the results of *Cflar^{LKO}* and *CflarFgf18^{LKO}* mice in the old Figure 3. As shown in Supplementary Figure 5, deletion of *Fgf18* in hepatocytes did not attenuate or exacerbate liver injury or fibrosis. This result is partly because the induction levels of *Fgf18* were very low in hepatocytes under *Cflar*-sufficient conditions. In sharp contrast, under *Cflar*-deficient conditions in hepatocytes, liver injury and fibrosis were drastically exacerbated. Under these conditions, the expression of *Fgf18* was highly elevated compared to that in *Fgf18^{LKO}* mice. Therefore, deletion of *Fgf18* attenuated liver fibrosis in *CflarFgf18^{LKO}* mice compared to *Cflar^{LKO}* mice, suggesting that elevated FGF18 contributed to liver fibrosis. We have made a new Figure 3 to include these results and mentioned them in the Results (Lines 242-261).

4, Since FGF18 can regulate inflammation, does this mean that FGF18 directly affects the development of liver fibrosis through inflammation?

RE: Thank you for identifying this critical issue. Indeed, the expression of several inflammatory cytokines was elevated in the livers of *Fgf18* Tg mice (Supplementary Figure 8). Moreover, CellChat analysis revealed that HSCs recruit neutrophils, monocytes, and dendritic cells through the production of chemokines in the livers of *Fgf18* Tg mice (Figure 6e). Given that scar-associated macrophages play a crucial role in the development of liver fibrosis^{1,2}, it is also possible that FGF18 recruits scar-associated macrophages, which subsequently activate HSCs, leading to liver fibrosis. We have discussed this issue in the Discussion section (Lines 510-517).

5, FGF18 activates the proliferation of HSCs through *Ccnd1*. Does the classic FGF-promoted proliferation pathway FGF/FGFR/Ras/ERK also play an important role in this process? Further, how does FGF18 promote the upregulation of *Ccnd1* expression?

RE: It is important to elucidate the mechanism of how FGF18 stimulates the proliferation of HSCs. As the reviewer mentioned, a MEK inhibitor, U0126, completely abolished the FGF18-induced increase in *Ccnd1* (Fig. 7d), suggesting that the MEK/ERK pathway is essential for the induction of *Ccnd1* (Lines 399-401). We further investigated what transcription factor is involved in this process. A previous study reported that Paxillin,

Elk-1, and c-Fos are involved in the upregulation of *Ccnd1*⁴. We found that FGF18 elevated the expression of *Fos* but not *Paxillin* or *Elk1* in HSCs; moreover, the elevation of *Fos* was blocked in the presence of U0126 (Rebuttal Figure 1). c-Fos may be responsible for the induction of *Ccnd1*, but further study will be required to fully address this issue. Thus, we have only included the results in a rebuttal letter.

Rebuttal Figure 1. FGF18 upregulates Fos, and its expression is blocked in the presence of a MEK inhibitor. HSCs were isolated from wild-type female mice using Nycodenz as described in the Methods. HSCs were stimulated with FGF18 (100 ng/mL) in the absence or presence of U0126 (10 μ M) for 24 hours. The expression of the indicated genes was analyzed by qPCR. Results are mean \pm SD of triplicate samples. Statistical significance was determined by one-way ANOVA with Tukey's multiple comparisons. Results are representative of two independent experiments.

Reviewer #2 (Remarks to the Author):

Fibroblast growth factor 18 activates hepatic stellate cells that lead to liver fibrosis

- **What are the noteworthy results?**

These remain the same as after first review:

In their report, Tsuchiya and colleagues use a hepatocyte-specific Cflar knock-out model to show increased liver (hepatocyte) damage leading to several aspects of chronic liver disease (CLD) including ductular reaction and fibrosis. Upon usage of the CDE diet, these knock-out mice develop a more severe phenotype than their wild-type counterparts. Upon more deep investigation of these models, Fgf18 is identified as a potent (but partial) mediator of the Cflar knock-out phenotype. Hepatocyte-specific knock-out of Fgf18 in hepatocyte-specific Cflar knock-out mice alleviates the phenotype and hepatocyte-specific Fgf18 overexpression mimics the hepatocyte-specific Cflar knock-out mice phenotype. Moreover, hepatocyte-specific Fgf18 overexpression shows clear signs of spontaneous development of CLD: inflammation, ductular reaction, fibrosis, and angiogenesis, bypassing the need for hepatocyte damage (ie ALT levels normal). Next, the Fgf18 function is linked to hepatic stellate cell activation (HSC) activation including insightful single cell RNA Seq analysis. Fgf18 exhibits correlation with fibrogenic genes, suggesting its plausible relevance to human chronic liver disease.

- **Will the work be of significance to the field and related fields? How does it compare to the established literature? If the work is not original, please provide relevant references.**

These remain the same as during first review.

As mentioned by the authors, the findings presented here contrast that of a recent publication by Tong et al. However, the experimental setup is different: overexpression of Fgf18 in this manuscript is genetic whereas

Tong et al use a vector-based method. Fgf18 knock-out in this manuscript consists of (combined Cflar-)Fgf18 hepatocyte-specific knock-out whereas Tong et al use an HSC-specific knock-out. Additionally, experimental models differ between manuscripts. More experiments in the future might elucidate these discrepancies.

If anything, both manuscripts underline a role for Fgf18 in CLD which is novel and thus provides a significant contribution to the field.

- Does the work support the conclusions and claims, or is additional evidence needed?

The authors have undertaken a significant revision of the manuscript. The evidence has been expanded where necessary, providing greater elaboration. The claims presented in this paper are substantiated by the data presented.

RE: Thank you for your comments.

- Are there any flaws in the data analysis, interpretation, and conclusions? - Do these prohibit publication or require revision?

Numerous issues have been addressed, particularly those related to the experimental setup and cell culture. A few minor comments may still need attention. Overall, the manuscript has been adapted and has successfully addressed several critical concerns. After these minor comments have been discussed, this reviewer believes the manuscript is ready for publication.

RE: Thank you for your comments.

- Is the methodology sound? Does the work meet the expected standards in your field?

The methodology differs from more conventional chronic liver disease (CLD) models, such as CCl₄, bile duct ligation, or metabolic MASLD, by utilizing the less conventional CDE model.

The quality of HSC culture has been highly improved.

The strength of the genetic mouse models deserves emphasis, particularly in investigating the role of *Fgf18* in liver function. The successful overexpression or knock-out of indicated genes has been addressed.

RE: Thank you for your comments.

- **Is there enough detail provided in the methods for the work to be reproduced?**

Yes.

Sex and Gender Equity in Research – SAGER – guidelines

DDC: male mice, other: female mice. The method section might further elaborate on the choice of sex.

RE: As suggested, we have mentioned the gender of the mice used in the study in detail in the Methods (Lines 604-613).

Specific comments

- **Language**

A revision of language has been done and been improved. In the abstract, line 59 might differently expressed.

RE: As suggested, we have changed the text to “Deletion of *Fgf18* in hepatocytes attenuated liver fibrosis; conversely, overexpression of *Fgf18* promoted liver fibrosis.” (Lines 59-60).

- **Title**

The title has been adjusted and fits better.

RE: Thank you for your comments.

- **NAFLD – MASLD - MASLD**

During the initial review, the nomenclature of NAFLD was updated to MAFLD. However, in the meantime, the nomenclature has once more evolved to MASLD/MASH. Kindly ensure consistency with this updated terminology across the manuscript. Reference: DOI: 10.1016/j.jhep.2023.06.003.

RE: Thank you for the suggestion to update the nomenclature of NAFLD. As suggested, we have changed NAFLD and NASH to MASLD and MASH, respectively, in the Introduction and Results (Lines 74-79, 106, 108, 209).

The authors have avoided making assertions about the pathogenesis of MASLD, considering that the CDE diet does not accurately represent this disease.

- **Introduction**

MASLD comment: cf supra

The introduction has been expanded to encompass a section on hepatic stellate cells (HSCs). It would be beneficial to provide better clarification regarding the presence of quiescent HSCs in a healthy liver, as well as to emphasize that their activation occurs subsequent to liver injury. (lines 84 – 88)

RE: As suggested, to clarify the roles of HSCs under physiological and pathological conditions, we have added the sentences “Under physiological conditions, quiescent HSCs” and “In response to liver injury” to the Introduction (Lines 84 and 88).

- **Results – Lines 130 - 174 (previous lines 119 – 148) + Figure 1**

Conclusion: Hepatocyte-specific Cflar knock-out mice spontaneously develop characteristics of CLD. When exposed to the CDE diet, they

develop a more severe phenotype.

The experimental setup has been modified as requested during the initial review round. Comprehensive data is now accessible for all four conditions (WT & KO +/- CDE). Additionally, the successful knock-out/down of Cflar specifically in hepatocytes (both protein and gene expression) has been incorporated. These modifications effectively address the primary concerns in this section. Consequently, the conclusions are adequately supported by substantial evidence.

RE: Thank you for your comments.

The authors mention the lack of statistical significance when comparing 4 groups instead of 2 groups. This is due to decreasing power as the amounts of groups to compare increase. This reviewer thinks that the absence of statistical significance in certain instances does not undermine the validity of the claims made. Moreover, supplementary figure 1 appropriately addresses this matter.

RE: Thank you for your comments.

- **Results – Lines 176 - 220(previous 149 – 177) + Figure 2**

Conclusion: RNA Seq analysis and overexpression techniques are purposefully used to reveal Ffg18 as a potential mediator of the Cflar knock-out phenotype.

Considering the experimental setup of Figure 2c: Are these conditions the same as those in Figure 1? If so, please use consistent x-axis legends for both figures to enhance clarity. If not, kindly indicate the differences between the setups of Figure 1 and Figure 2c.

RE: Thank you for noting this ambiguous expression. We compared the expression of the indicated genes in the livers of 8-week-old mice before and after CDE diet feeding for 4 weeks. Precisely, eight-week-old mice that were initially fed the normal diet were

either sacrificed or switched to the CDE diet for an additional 4 weeks. Consequently, the experimental conditions in Figure 2c differed slightly from those in Figure 1. As suggested, we have changed the x-axis to (-) and 4 weeks (Lines 184-185, 1252-1253).

HTVi overexpression:

(1) As requested after the initial review, successful overexpression of gene expression is demonstrated, although not for protein.

RE: Thank you for your comments.

(2) As suggested after the initial review, staining of the eGFP control is provided.

RE: Thank you for your comments.

(3) While the methods section specifies that the HTVi should selectively overexpress the mentioned genes in hepatocytes, the authors were unable to demonstrate this specific overexpression in hepatocytes when compared to other liver cells. Nevertheless, the authors do not assert that the differences resulting from overexpression are specifically linked to hepatocytes.

The authors have responded to and substantiated their choice for a hepatocyte-specific approach targeting Fgf18, as evident in Figure 2 of the rebuttal.

As recommended, the authors have re-evaluated GSE99010, further reinforcing the evidence for Fgf18 expression across multiple models of liver fibrosis.

RE: Thank you for your comments.

- **Results Line 221 - 256 (previous lines 178 – 199) + Figure 3**

Conclusion: Here, evidence shows that hepatocyte-specific knock-out of Fgf18 partially rescues the fibrogenic (but not ductular reaction) phenotype of the Cflar knock-out. This confirms hepatocyte derived Fgf18 as a partial mediator of the Cflar knock-out phenotype.

RE: Thank you for your comments.

Following the request made in the first review round, successful knock-out/down of Cflar is demonstrated for both protein and gene expression, including a comparison between hepatocytes and other liver cells. However, for Fgf18, this knock-down is only illustrated for gene expression, with protein data lacking. This matter is addressed in Supplementary Figure 6, which appears to be linked to Fgf18 protein expression levels already being below the detection limit under normal conditions.

RE: Thank you for your comments.

Lines 236 – 240 refer to HSCs, but this actually seems to be referring to NPC.

RE: As suggested, we have changed “HSCs” to “NPCs” (Lines 237-240).

During the initial review round, numerous comments were provided regarding the selection of the four conditions (CflarFF, CflarKO, CflarFFFgf18FF, CflarKOFgf18KO). In their response to the reviewer's comments, the authors provide an explanation for their choices of these conditions. Supplementary Figure 5 has been added to illustrate that a knockout of Fgf18 (Fgf18LKO) alone does not induce any alterations in the phenotype of healthy mice or those subjected to a CDE diet. Due to technical limitations, generating CflarFFFgf18LKO mice is not feasible, and as a substitute, Fgf18LKO mice have been used.

RE: Thank you for your comments.

- **Results Line 257 - 296 (previous lines 200 – 225) + Figure 4**

Conclusion: Fgf18 overexpression bypasses the need for hepatocyte damage to induce characteristics of CLD: fibrosis, inflammation, and angiogenesis indicating its role in CLD.

As requested following the initial review round, successful overexpression of Fgf18 for both protein and gene expression, as well as a comparison between hepatocytes and other liver cells, has been demonstrated.

RE: Thank you for your comments.

- **Results Line 297 – 328 (previous lines 226 – 258) + Figure 5**

Conclusion: The RNA Sequencing data nicely shows an overlap of Fgf18 overexpression and the Cflar knock-out further linking the fibrosis phenotype of Cflar knock-out to Fgf18. A deeper analysis of CD31- CD34+ cells identifies HSCs as one of the affected cells by Fgf18 overexpression.

No comments.

- **Results Lines 329-378 (previous lines 259 – 306) + Figure 6**

CellChat analysis nicely shows the impact of Fgf18 and provides evidence for direct hepatocyte-HSC crosstalk.

After first review round, some remarks were made, the authors have responded to these issues:

(1) Concerning the exclusion of hepatocytes, technical difficulties obstructed the goal of performing scRNA on hepatocytes.

(2) despite the selection of non-parenchymal cells, the single RNA seq results still show plenty of hepatocytes. The authors have replied to this issue.

RE: Thank you for your comments.

Line 372: please add “in HSCs” after “upregulate the expression of Hgf” for clarity

RE: As suggested, we have changed the sentence to “Given that FGF18 did not upregulate the expression of *Hgf* in HSCs” (Line 379).

- **Results Line 379 - 408 (Previous lines 307 – 328) + Figure 7**

Following the initial review round, significant concerns were raised regarding the culture of HSCs. The authors have addressed these issues by concentrating on a single culture method involving Nycodenz- based HSC isolation. The validity of this culture approach is demonstrated, and TGFbeta experiments indicate the susceptibility of these HSCs to fibrogenic stimulation.

Using these cultures, Fgf18 is shown to stimulate only the proliferation of HSCs, without inducing an increase in the expression of fibrogenic genes. It would be interesting to observe if the staining for α SMA in these three conditions (untreated, TGFb-stimulated, & FGF18-stimulated) reflects any alterations in the phenotype of these cells.

RE: As suggested, we performed immunostaining with anti- α -SMA antibody. As expected, TGF stimulation induced striking morphological changes in HSCs, such as loss of a star-shaped appearance and elongated and spindle-shaped morphology. In sharp contrast, HSCs still showed a star-shaped appearance upon stimulation with FGF18. We have described these results in the Results and made Supplementary Figure 14 to include the results (Lines 393-397).

Based on these findings, TGFbeta and Fgf18 appear to play a role in an auto-regulatory mechanism.

A portion of figure 7a appears to be replicated in Fig7h, or were these

experiments repeated with an additional fourth condition?

RE: Yes. We first examined the expression of *Fgf18* under three conditions, as shown in Fig. 7a. We then further repeated similar experiments, including TGF β plus FGF18 treatment.

- **Results Line 409 - 416 + Figure 8**

The correlation between Fgf18 expression and Col1a1 & Acta2 expression is illustrated.

RE: As suggested, I have included a model to illustrate how FGF18 promotes liver fibrosis (Figure 8b).

The authors have re-analysed GSE136103. However, their search for HSCs focused on LRAT+ cells, which yielded no results. It's important to note that the original paper's data analysis proceeded as follows: first, the identification of mesenchymal cells based on markers such as PDGFRB, ACTA2, COL1A1, COL1A2, COL3A1, DES, and DCN. Subsequently, the recognition of hepatic stellate cells (HSCs) using RGS5 & PDGFRA as a marker for myofibroblasts within the cirrhotic niche. It is strongly recommended to re-analyze this data using the same approach as the original manuscript, employing RGS5 versus PDGFRA as markers.

RE: Thank you very much for your thoughtful suggestion. As suggested, we reanalyzed GSE136103 and found clusters of HSC based on the expression of *RGS5* but not *LRAT*. A few cells expressed *FGF18* in the myofibroblast cluster in healthy livers and the HSC cluster in cirrhotic human livers. Given that the number of cells in HSC clusters from the cirrhotic livers was lower than those from healthy livers, more HSCs would express *FGF18* in cirrhotic human livers if sufficient numbers of HSCs were obtained from fibrotic livers. Since these results did not convincingly support our conclusion, we have only included the results in a rebuttal letter.

Rebuttal Figure 2 | scRNA-seq data of healthy and cirrhotic human livers.

Reanalysis of scRNA-seq data (GSE136103). **a**, UMAP visualization of single-cell clusters isolated from healthy ($n = 3$) and cirrhotic ($n = 3$) human livers based on gene expression profile similarity focusing on mesenchymal cells. Each color represents a distinct cell population. **b**, UMAP visualization of the expression of the indicated genes.

- Discussion**

Line 420: this line directly contradict the claims made in supplemental figure 5. Please address this.

RE: As suggested, we have changed the sentence to “Deletion of *Fgf18* in hepatocytes attenuated CDE-induced fibrosis in *Cflar^{LKO}* mice” (Line 438).

Line 442: exploring how *Fgf18* promotes HSC proliferation without enhancing fibrogenesis, and what triggers the activation of these proliferated HSCs, could enhance the discussion's depth.

RE: Thank you for noting these critical issues. We assume that the expansion of HSCs requires some space; therefore, it is reasonable that FGF18 suppresses the production of extracellular matrix to make space for HSC proliferation. Moreover, the expression levels of profibrotic genes in HSCs were relatively higher than those in hepatocytes, even if their expression levels declined in the presence of FGF18. This finding indicates that the increased HSCs by FGF18 stimulation play an important role in the accumulation of collagens and extracellular matrix (Lines 503-510).

Regarding the activation of proliferating HSCs, we hypothesize that HSCs further respond to profibrotic stimuli derived from scar-associated macrophages, such as platelet-derived growth factor (PDGF) or IL-1 β ¹, ultimately contributing to the development of liver fibrosis. We have mentioned these points in the Discussion (Lines 510-517).

- **Materials & Methods**

The authors have responded adequately to all comments concerning the materials and methods.

RE: Thank you for your comments.

- **MASLD pathogenesis**

The authors have refrained from making conclusions on MASLD pathogenesis, considering the limitations of the CDE diet model.

RE: Thank you for your comments.

- **Summary Figure:**

A summary figure could provide added clarity and value to the manuscript.

RE: As suggested, we have included the summary figure as Fig. 8b to explain the whole picture of our study for readers (Lines 427-433).

Reviewer #3 (Remarks to the Author):

The authors have addressed my concerns.

RE: Thank you for your comments.

Rebuttal letter references

1. Ramachandran P, *et al.* Resolving the fibrotic niche of human liver cirrhosis at single-cell level. *Nature* **575**, 512-518 (2019).
2. Fabre T, *et al.* Identification of a broadly fibrogenic macrophage subset induced by type 3 inflammation. *Sci Immunol* **8**, eadd8945 (2023).
3. Tong G, *et al.* Fibroblast growth factor 18 attenuates liver fibrosis and HSCs activation via the SMO-LATS1-YAP pathway. *Pharmacol Res* **178**, 106139 (2022).
4. Sen A, *et al.* Paxillin mediates extranuclear and intranuclear signaling in prostate cancer proliferation. *J Clin Invest* **122**, 2469-2481 (2012).